# DC-LA: Difference-of-Convex Langevin Algorithm

**Hoang Phuc Hau Luu** [1]  **Zhongjian Wang** [1]

## Abstract

We study a sampling problem whose target distribution is $\pi \propto \exp(-f - r)$ where the data fidelity term $f$ is Lipschitz smooth while the regularizer term $r = r_1 - r_2$ is a non-smooth difference-of-convex (DC) function, i.e., $r_1, r_2$ are convex. By leveraging the DC structure of $r$, we can smooth out $r$ by applying Moreau envelopes to $r_1$ and $r_2$ separately. In line with DC programming, we then redistribute the concave part of the regularizer to the data fidelity and study its corresponding proximal Langevin algorithm (termed DC-LA). We establish convergence of DC-LA to the target distribution $\pi$, up to discretization and smoothing errors, in the $q$-Wasserstein distance for all $q \in \mathbb{N}^*$, under the assumption that $V$ is distant dissipative. Our results improve previous work on non-log-concave sampling in terms of a more general framework and assumptions. Numerical experiments show that DC-LA produces accurate distributions in synthetic settings and provides qualitatively reasonable uncertainty quantification in a real-world Computed Tomography application.

## 1. Introduction

The sampling problem from a Gibbs distribution in $\mathbb{R}^d$ of the form $\pi \propto e^{-V}$ is fundamental in machine learning and Bayesian inference (Sanz-Alonso et al., 2023). A conventional approach is via the Langevin dynamics, under the assumption that $V$ is differentiable:

$$dX_t = -\nabla V(X_t)dt + \sqrt{2}dB_t \tag{1}$$

where $B_t$ denotes the standard Brownian motion in $\mathbb{R}^d$. Under suitable conditions on $V$, this stochastic differential

equation admits $\pi$ as a unique invariant distribution, and the law of $X_t$ converges to $\pi$ as $t \to \infty$ (Roberts & Tweedie, 1996). In practice, the stochastic dynamics (1) must be discretized for implementation. A simple Euler-Maruyama scheme results in the so-called Unadjusted Langevin Algorithm (ULA)

$$X_{k+1} = X_k - \gamma \nabla V(X_k) + \sqrt{2\gamma}Z_{k+1}$$

where $\gamma > 0$ and $\{Z_k\}_k$ follows i.i.d. normal distribution.

When $V$ is (strongly) convex, the sampling problem falls within the class of log-concave sampling, and the properties of these dynamics are well studied; see, e.g., the recent note (Chewi, 2023). In contrast, in many practical applications (Weinan et al., 2002; Cui et al., 2023), the target distribution may exhibit multi-peak, non-log-concave, non-log-differentiable behaviors. Beyond convexity, distant dissipativity (also known as *strong convexity at infinity*) provides a key structural condition under which convergence of these dynamics can still be established (Eberle, 2016; De Bortoli & Durmus, 2019; Johnston et al., 2025). While the ULA is simple and easy to implement, it is not universally well-suited to all problems, particularly when the potential $V$ is non-differentiable. For composite potentials with a nonsmooth component, Moreau (or Moreau-Yosida) smoothing leads to MYULA-type algorithms, which replace the nonsmooth term by its Moreau envelope (Durmus et al., 2018); more recently, Habring et al. (2026) proposed a successive Moreau-envelope Langevin scheme that samples from a sequence of smoothed distributions approaching the target, inspired by diffusion models. However, when the nonsmooth component is non-weakly-convex, directly applying a Moreau envelope to the whole term can be difficult to analyze or ill-behaved. In this work, we consider nonconvex and nonsmooth potentials $V$ that satisfy a distant dissipativity condition and admit the decomposition

$$V = f + r = f + (r_1 - r_2) \tag{2}$$

where $f$ is $L$-smooth and $r_1, r_2$ are real-valued convex functions. The function $r$ is called a difference-of-convex (DC) function, with $r_1$ and $r_2$ as its DC components.

The class of DC functions forms a broad and expressive subclass of nonconvex, nonsmooth functions (Pham Dinh & Le Thi, 1997; Le Thi et al., 2022; Le Thi & Pham Dinh, 2005;

[1]Division of Mathematical Sciences, School of Physical and Mathematical Sciences, Nanyang Technological University, Singapore. Correspondence to: Zhongjian Wang <zhongjian.wang@ntu.edu.sg>.

*Proceedings of the 43rd International Conference on Machine Learning*, Seoul, South Korea. PMLR 306, 2026. Copyright 2026 by the author(s).

Nouiehed et al., 2019; Cui et al., 2018; Ahn et al., 2017). In the context of Bayesian inference, $f$ typically arises from the likelihood and is often $L$-smooth, such as $\|Ax - b\|^2$ under Gaussian noise, and the regularizer $r$, which can be nondifferentiable, comes from the prior[1]. There are two main classes of priors: hand-crafted priors (e.g., LASSO, total variation), which are designed using domain knowledge to promote desired structural properties in the solution, and data-driven priors learned from training data using deep neural networks (e.g., U-Net (Ronneberger et al., 2015)), which are capable of capturing more complex data characteristics. In the former paradigm, nonconvex DC priors offer greater flexibility for encoding expert knowledge and have been shown to outperform convex priors in compressed sensing tasks (Yin et al., 2015; Liu & Pong, 2017; Yao & Kwok, 2018). Hand-crafted DC priors (with *explicit* decompositions) include Laplace, log-sum penalty, Smoothly Clipped Absolute Deviation, Minimax Concave Penalty, Capped-$\ell_1$, PiL, $\ell_p^+$ $(0 < p < 1)$, $\ell_p^-$ $(p < 0)$, $\ell_1 - \ell_2$, and $\ell_1 - \ell_{\sigma_p}$ (Le Thi et al., 2015; Yin et al., 2015; Luo et al., 2013; Yao & Kwok, 2018). In the latter paradigm, data-driven DC regularizers based on the difference of two input-convex neural networks (ICNNs), named DICNNs, have been shown to generalize better than input–weakly convex neural networks (Zhang & Leong, 2025), while remaining more analyzable than general neural priors.

In this work, we introduce and study a practical sampler named Difference-of-convex Langevin algorithm (DC-LA) that aims to sample from $\pi \propto e^{-V}$ where $V$ is given in (2). DC-LA is essentially a forward-backward sampler whose forward-backward splitting is based on the principle of DC programming and DC algorithm (Pham Dinh & Le Thi, 1997). We also incorporate Moreau smoothing techniques tailored to DC regularizers (Sun & Sun, 2023; Hu et al., 2024) to facilitate the analysis (detailed in Section 3). Let $\lambda > 0$ be a smoothing parameter, since $r_1$ and $r_2$ are convex, their Moreau envelopes, denoted $r_1^\lambda, r_2^\lambda$, are well-defined and smooth (see Subsection 2.2). DC-LA reads as

$$
X_{k+1} = \text{Prox}_{\gamma r_1^\lambda} \circ \\
\left( X_k - \gamma \nabla f(X_k) + \gamma \nabla r_2^\lambda(X_k) + \sqrt{2\gamma} Z_{k+1} \right),
\tag{3}
$$

where $\gamma > 0$ is the step size and $\{Z_k\}_k$ follows i.i.d. normal distribution. As detailed in Section 3, DC-LA can be decomposed into *elementary operators*: gradient of $f$ and proximal operators of $r_1$ and $r_2$. Our main theoretical results are as follows: if $V$ is distant dissipative (a.k.a. strongly convex at

---

[1] The prior $p_{\text{prior}}(x) \propto e^{-r(x)}$ may be improper if the regularizer $r(x)$ does not grow sufficiently fast at infinity, which is typically the case for DC regularizers. Throughout the theoretical analysis, we assume that its product with the likelihood defines a proper posterior (with augmentation when needed).

| Regularizer $r = r_1 - r_2$ | $r_1$ | $r_2$ | Weakly convex? |
|---|---|---|---|
| $\ell_1 - \ell_2$ | L | L | No |
| $\ell_1 - \ell_{\sigma_q}$ | L | L | No |
| Capped-$\ell_1$ | L | L | No |
| PiL | L | L | No |
| $\ell_1 - \ell_2^p, 1 < p < 2$ | L | H | No |
| Geman penalty | L | S | Yes |
| Log-sum penalty | L | S | Yes |
| Laplace penalty | L | S | Yes |
| MCP | L | S | Yes |
| SCAD | L | S | Yes |
| DICNNs leaky ReLU | L* | L* | No, in general |

*Table 1.* Examples of DC regularizers and their regularities: L = Lipschitz continuous, H = $\kappa$-Hölder continuous gradient with $\kappa \in (0, 1)$, S = smooth. (*) Lipschitzness is encouraged by the training framework. See Appendices D and F for further details.

infinity), $r_1$ is Lipschitz continuous, and $r_2$ is *either*

(a) Lipschitz continuous;

(b) differentiable whose gradient is $\kappa$-Hölder continuous for $0 < \kappa < 1$ (class $C^{1,\kappa}$);

(c) smooth (class $C^{1,1}$) – in this case, we directly use $\nabla r_2$ in (3);

then for any $q \in \mathbb{N}^*$, the $q$-Wasserstein distance between $p_{X_{k+1}}$ (law of $X_{k+1}$) and $\pi$ is upper bounded by $O(\rho^{k\gamma}) + O(\gamma^{\frac{1}{2q}}) + O(\lambda^{\frac{1}{q}})$ where $\rho \in (0, 1)$. Here, the hidden constants in the first and second Big O's and $\rho$ may depend on $\lambda$, and all constants depend on $q$. In the above result, the assumptions on $r_2$ form a continuum of regularity of $r_2$: case (b) interpolates between the nonsmooth and smooth regimes as $\kappa$ increases from 0 to 1, with case (a) corresponding to $\kappa = 0$ and case (c) corresponding to $\kappa = 1$. We also note that case (c) recovers the weakly convex setting, whereas cases (a) and (b) allow DC regularizers that need not be weakly convex. Table 1 shows the regularity conditions of some DC regularizers. Our theoretical results leverage recent advances in forward–backward sampling algorithms (Renaud et al., 2025a), which we tailor and revise for the DC setting. See the first paragraph in the **Related Work** section for a detailed discussion. To our knowledge, DC-LA handles non-weakly convex DC regularizers (e.g., $\ell_1 - \ell_2$ and DICNNs), for which no prior *Langevin-type* convergence guarantees existed.

**Contributions** We study the sampling problem with the nonsmooth DC potential $V$, see (2). We propose a forward-backward style algorithm termed DC-LA and establish its convergence to $\pi$ in terms of the $q$-Wasserstein distance– up to discretization and smoothing errors–for all $q \in \mathbb{N}^*$ under a distant dissipativity condition on $V$. Numerical experiments demonstrate the merit of DC-LA, showing that

it produces faithful empirical distributions in synthetic experiments and provides qualitatively reasonable uncertainty quantification under the assumed model for a real-world computed tomography application.

### Related work

**Euclidean forward-backward samplers**[2]  For the composite structure of $V = f + r$, the following forward-backward scheme, named Proximal Stochastic Gradient Langevin Algorithm (PSGLA)

$$X_{k+1} = \text{Prox}_{\gamma r}\left(X_k - \gamma \nabla f(X_k) + \sqrt{2\gamma} Z_{k+1}\right) \quad (4)$$

is a natural choice. When both $f$ and $r$ are convex, convergence of PSGLA and its variants was studied thoroughly, for example, in (Durmus et al., 2019; Salim & Richtarik, 2020; Salim et al., 2019; Ehrhardt et al., 2024). Recently, (Renaud et al., 2025a) studied the case where $f$ is $L$-smooth and $r$ is *weakly convex* [3], and established for the first time convergence in $q$-Wasserstein distance ($q \in \mathbb{N}^*$) of PSGLA to $\pi$ (up to discretization error), assuming convex at infinity (distant dissipativity). To be specific, they derived $W_q(p_{X_k}, \nu_\gamma) = O(\rho^{k\gamma} + \gamma^{\frac{1}{2q}})$ where $\rho \in (0,1)$ and $\nu_\gamma$ satisfies $\lim_{\gamma \to 0} W_q(\nu_\gamma, \pi) = 0$ and $W_q(\nu_\gamma, \pi) = O(\gamma^{\frac{1}{q}})$ if $r$ is further assumed to be Lipschitz continuous. We note that a weakly convex function $r$ is a DC function whose second DC component is $r_2 = \frac{\eta}{2}\|\cdot\|^2$ for some $\eta > 0$. The converse, however, does not hold in general, for instance, several DC regularizers used in compressed sensing—such as $\ell_1 - \ell_2$ (Yin et al., 2015; Lou & Yan, 2018), $\ell_1 - \ell_{\sigma_q}$ (Luo et al., 2013), Capped $\ell_1$ (Zhang, 2010b), and DCINNs with leaky ReLU activations (Zhang & Leong, 2025)—are *not* weakly convex (see Appendix F), yet are widely used in image and signal reconstruction because they better model (gradient) sparsity and yield higher reconstruction quality than convex counterparts. Furthermore, while the assumption of convexity at infinity is quite a minimal assumption if one seeks convergence to $\pi$, their analysis relies on a strict convexity condition on $r$ at infinity. Loosely speaking, Assumption 3 in (Renaud et al., 2025a) requires the convex modulus at infinity of $r$ (formally its Moreau envelope $r^\gamma$) to dominate four times of the Lipschitz smoothness constant (defined over some proximal region) of $r$. This requirement is very strong, since the latter usually dominates the former instead. Indeed, in Appendix E we argue that this assumption is not possible to hold. That said, their work provides important stability results (Subsection 2.4) and lays the foundation for our work. In another work tackling nonconvex, nonsmooth potentials, Luu et al. (2021) also studied forward-

backward schemes leveraging the Moreau envelope. However, their results only established consistent guarantees for the schemes without quantifying convergence to the target distribution. When $f$ and $r$ are both nonsmooth but convex, (Habring et al., 2024) established the convergence of the scheme: $X_{k+1} = \text{Prox}_{\gamma r}(X_k - \gamma \partial f(X_k)) + \sqrt{2\gamma} Z_{k+1}$. Since the proximal operator is nonlinear, applying it before adding Gaussian noise makes this scheme fundamentally different from (4), where the proximal step acts on a noise-perturbed point.

**Wasserstein forward-backward samplers**  On a different front, where forward-backward schemes are designed directly in the Wasserstein space, Salim et al. (2020) studied a Wasserstein proximal algorithm in the convex setting, and Luu et al. (2024) extended the scheme to DC settings. In contrast to our approach, where (Euclidean) proximal operators are often available in closed form, the Wasserstein proximal operators lack closed-form expressions and must instead be approximated, e.g., by neural networks (Mokrov et al., 2021; Luu et al., 2024). When restricting to Gaussian distributions, the Wasserstein proximal operator admits a closed-form expression, whereas computing the Bures–Wasserstein gradient in the forward step relies on Monte Carlo estimation or variance-reduction techniques (Diao et al., 2023; Luu et al., 2025).

**Gibbs samplers for composite structures**  Composite structures can also be handled by Gibbs samplers, which decompose a complicated sampling problem into simpler sampling subproblems (Sorensen et al., 1995; Geman & Geman, 1984). Recently, Sun et al. (2024) established first-order stationary convergence guarantees (in terms of Fisher information) for Gibbs sampling in the nonconvex setting. Complementarily, Kuric et al. (2025) proposed the Gaussian latent machine, which introduces auxiliary Gaussian variables for product-of-experts models and yields an efficient two-block Gibbs sampler, highlighting the practical usefulness of Gibbs augmentation for composite structures.

## 2. Preliminary

### 2.1. Lipschitz and distant dissipativity

A function $f$ is called Lipschitz continuous (or $M$-Lipschitz for some $M > 0$) if $|f(x) - f(y)| \leq M\|x - y\|$ for all $x, y \in \mathbb{R}^d$. $f$ is called Lipschitz-smooth (or $L_f$-smooth for some $L_f > 0$) if

$$\|\nabla f(x) - \nabla f(y)\| \leq L_f \|x - y\|, \ \forall x, y \in \mathbb{R}^d.$$

while it is said to have $(\theta, M)$-Hölder continuous gradient if

$$\|\nabla f(x) - \nabla f(y)\| \leq M\|x - y\|^\theta, \ \forall x, y \in \mathbb{R}^d.$$

Given a drift $b : \mathbb{R}^d \to \mathbb{R}^d$, we say that $b$ is distant dissipative (or weak dissipative) (Mou et al., 2022; Debussche

---

[2]Here, "Euclidean" refers to the algorithmic design rather than the geometric nature of the underlying flow.

[3]A function $r$ is called $\eta$ weakly convex ($\eta > 0$) if $r + \frac{\eta}{2}\|\cdot\|^2$ is convex.

et al., 2011) if there exist $R \geq 0$ and $m > 0$ such that $\forall x, y \in \mathbb{R}^d$ satisfying $\|x - y\| \geq R$,

$$\langle b(x) - b(y), x - y \rangle \geq m\|x - y\|^2. \qquad (5)$$

$b$ is also called $(m, R)$-distant dissipative. If $b = \nabla V$ for some $V$, by abuse of terminology, we also say that $V$ is distant dissipative. In our work, as $V$ is not differentiable, we may extend the concept to some generalized notion of gradients (Section 3). Note that a strongly convex function is dissipative, i.e., (5) holds for all $x, y \in \mathbb{R}^d$.

## 2.2. Moreau envelope and proximal operator

Given a convex function $g$ and $\lambda > 0$, we denote the Moreau envelope $g^\lambda$ of $g$ as

$$g^\lambda(x) = \inf_y \left\{ g(y) + \frac{1}{2\lambda}\|x - y\|^2 \right\}$$

and the proximal operator of $g$ as

$$\text{Prox}_{\lambda g}(x) = \text{argmin}_y \left\{ g(y) + \frac{1}{2\lambda}\|x - y\|^2 \right\}.$$

Note that, in this paper, we only work with Moreau envelopes and proximal operators of convex functions. In the following lemma, we summarize some known important properties of the Moreau envelope and the proximal operator (Bauschke & Combettes, 2020) that are used throughout this paper.

**Lemma 2.1.** *Let $g$ be a convex function and $\lambda > 0$. The following properties hold*

(i) $g^\lambda$ *is convex, $\frac{1}{\lambda}$-smooth, and is a lower bound of $g$, i.e., $g^\lambda(x) \leq g(x)$, $\forall x \in \mathbb{R}^d$. Furthermore, if $g$ is $G$-Lipschitz, $g(x) \leq g^\lambda(x) + \frac{G^2\lambda}{2}$, $\forall x \in \mathbb{R}^d$.*

(ii) $\nabla g^\lambda(x) = \frac{1}{\lambda}(x - \text{Prox}_{\lambda g}(x))$.

(iii) $\text{Prox}_{\lambda g}$ *is nonexpansive, i.e., for all $x, y \in \mathbb{R}^d$, $\|\text{Prox}_{\lambda g}(x) - \text{Prox}_{\lambda g}(y)\| \leq \|x - y\|$.*

(iv) $\nabla g^\lambda(x) \in \partial g(\text{Prox}_{\lambda g}(x))$ *where $\partial g$ denotes the usual convex subdifferential of $g$.*

## 2.3. $q$-Wasserstein distance

Given a measurable map $\varphi : \mathbb{R}^d \to \mathbb{R}^d$ and a probability measure $\mu$, the pushforward measure $\varphi_{\#}\mu$ is defined by $(\varphi_{\#}\mu)(A) = \mu(\varphi^{-1}(A))$ for all measurable sets $A$.

For $q \geq 1$, let $\mathcal{P}_q(\mathbb{R}^d)$ denote the set of probability distributions over $\mathbb{R}^d$ that have finite $q$-th moment: $\mu \in \mathcal{P}_q(\mathbb{R}^d)$ iff $\int \|x\|^q d\mu(x) < +\infty$. Let $\mu, \nu \in \mathcal{P}_q(\mathbb{R}^d)$, the $q$-Wasserstein distance between $\mu$ and $\nu$ is (Villani et al., 2008;

Ambrosio et al., 2005)

$$W_q(\mu, \nu) = \left( \min_{\xi \in \Gamma(\mu, \nu)} \int_{\mathbb{R}^d \times \mathbb{R}^d} \|x - y\|^q d\xi(x, y) \right)^{\frac{1}{q}}$$

where $\Gamma(\mu, \nu)$ is the set of probability distributions over $\mathbb{R}^d \times \mathbb{R}^d$ whose marginals are $\mu$ and $\nu$.

## 2.4. Langevin dynamics with general drifts

In the definition of the ULA, one may, in practice, replace the exact gradient $\nabla V$ with an approximate drift $b$, provided that $b$ remains close to $\nabla V$ (Rásonyi & Tikosi, 2022),

$$X_{k+1} = X_k - \gamma b(X_k) + \sqrt{2\gamma} Z_{k+1}. \qquad (6)$$

Stability results are to establish (quantitative) bounds on the distance (e.g. Wasserstein) between the invariant distribution of ULA and the invariant distribution of (6) based on the distance between $b$ and $\nabla V$ (e.g., some $L_2$ norm). More generally, given two general drifts $b^1$ and $b^2$, it is desirable to bound the discrepancy between the invariant distributions of two processes by $\|b^1 - b^2\|_{L_2}$. Note that the step size $\gamma$ also influences the invariant distribution, as different values of $\gamma$ lead to different limiting laws. Under suitable conditions, let $\pi^1_\gamma$ and $\pi^2_\gamma$ denote the invariant distributions of the processes with drifts $b^1$ and $b^2$, respectively. It is established in (Renaud et al., 2023) that, under certain conditions, $W_1(\pi^1_\gamma, \pi^2_\gamma) = O(\|b^1 - b^2\|_{L_2(\pi^1_\gamma)}^{\frac{1}{2}}) + O(\gamma^{\frac{1}{8}})$ and later refined and generalized in (Renaud et al., 2025a) to $W_q(\pi^1_\gamma, \pi^2_\gamma) = O(\|b^1 - b^2\|_{L_2(\pi^1_\gamma)}^{\frac{1}{q}})$ for $q \in \mathbb{N}^*$. The latter result shows that discretization errors do not accumulate along the processes. Similar ergodic results were also established earlier in compact domains, see (Wang et al., 2021; Ferré & Stoltz, 2019).

## 3. Proximal Langevin algorithm with DC regularizers

Recall the target distribution $\pi \propto e^{-V}$ and $V = f + r$ where $f$ is Lipschitz smooth and $r$ is DC.

**Assumption 1.** $f$ is $L_f$-smooth and $r = r_1 - r_2$ where $r_1, r_2 : \mathbb{R}^d \to \mathbb{R}$ are convex functions.

We next impose a *distant dissipativity* condition on $V$. Unlike strong convexity, this assumption requires the gradient field of $V$ to be dissipative only when $x$ and $y$ are sufficiently far apart (Eberle, 2016; Eberle & Majka, 2019).

**Assumption 2** (Distant dissipative $V$). There exist $R_0 \geq 0, \mu > 0$ such that $\forall x, y \in \mathbb{R}^d$ with $\|x - y\| \geq R_0$,

$$\langle \nabla f(x) + u_1 - u_2 - \nabla f(y) - v_1 + v_2, x - y \rangle \geq \mu\|x - y\|^2, \qquad (7)$$

$\forall u_1 \in \partial r_1(x), u_2 \in \partial r_2(x), v_1 \in \partial r_1(y), v_2 \in \partial r_2(y)$.

Distant dissipativity (7) implies that $V$ exhibits quadratic growth and consequently $\pi \propto e^{-V}$ has sub-Gaussian tails (Appendix A).

**Difference of Moreau envelopes** Now we want to apply a forward-backward Langevin-type algorithm to $V$. Thanks to the DC structure of $r$, the principle of DC programming and DC algorithm (Pham Dinh & Le Thi, 1997) suggests that the forward step should apply to $f - r_2$ while the backward step should apply to $r_1$ for tractability and robustness. To retain a degree of smoothness in the forward step, we replace $r_2$ with its Moreau envelope $r_2^\lambda$. This modification is necessary, as the concavity of $-r_2$ and its associated tangent inequality are hardly sufficient to guarantee the stability results and to enable two-sided comparisons of general drifts (Subsection 4.1). On the other hand, the analysis in (Renaud et al., 2025a) suggests that, if the backward step is applied to $r_1$, $r_1$ should be smooth on the set $\operatorname{Prox}_{\gamma r_1}(\mathbb{R}^d) := \{\operatorname{Prox}_{\gamma r_1}(x) : x \in \mathbb{R}^d\}$, where $\gamma$ is the step size. This requirement arises in order to prevent the Lipschitz smoothness constant of $r_1^\gamma$ to explode as $\gamma \to 0$. In our setting, $\operatorname{Prox}_{\gamma r_1}(\mathbb{R}^d) = \mathbb{R}^d$ (Hiriart-Urruty, 2024, Fact 2), and $r_1$ remains nonsmooth on $\mathbb{R}^d$, so this requirement cannot be satisfied. To circumvent this issue, we further replace $r_1$ with its Moreau envelope $r_1^\lambda$. These considerations motivate the following augmented potential and distribution

$$V_\lambda = \left(f - r_2^\lambda\right) + r_1^\lambda, \quad \pi_\lambda \propto e^{-V_\lambda}, \qquad (8)$$

which results in the DC-LA (3).

*Remark* 3.1. The difference-of-convex structure of $r = r_1 - r_2$ allows this Moreau envelope approximation, rather than applying one Moreau envelope to the entire $r$ which is generally an ill-posed object (e.g., multi-valued and discontinuous, see Appendix G). This kind of difference-of-Moreau-envelope approximation has been used in the optimization literature (Sun & Sun, 2023; Hu et al., 2024).

*Remark* 3.2. In practice, there is also a class of sparsity promoting DC regularizers where $r_2$ is Lipschitz smooth (Yao & Kwok, 2018). In such a case, we do not need to smooth out $r_2$. We consider this case in Subsection 4.2.

We impose the following assumption on $r_1$.

**Assumption 3.** There exists $G_1 > 0$ such that $\forall x \in \mathbb{R}^d$, $\forall z \in \partial r_1(x)$, it holds $\|z\| \le G_1$.

Note that Assumption 3 is equivalent to $r_1$ being $G_1$-Lipschitz continuous. See Table 1 for regularizers that satisfy this assumption.

**Unrolled DC-LA** For a fixed $\lambda > 0$, starting from some initial distribution $X_0 \sim p_0$, DC-LA (3) with step size

$\gamma > 0$ can be rewritten as follows

$$Y_{k+1} = X_k - \gamma \nabla f(X_k) + \gamma \nabla r_2^\lambda(X_k) + \sqrt{2\gamma} Z_{k+1}$$
$$X_{k+1} = \operatorname{Prox}_{\gamma r_1^\lambda}(Y_{k+1}).$$
$$\text{(DC-LA)}$$

Since $\nabla r_2^\lambda(x) = (1/\lambda)(x - \operatorname{Prox}_{\lambda r_2}(x))$, $Y_{k+1}$ becomes

$$Y_{k+1} = \frac{\lambda + \gamma}{\lambda} X_k - \gamma \nabla f(X_k) - \frac{\gamma}{\lambda} \operatorname{Prox}_{\lambda r_2}(X_k) + \sqrt{2\gamma} Z_{k+1}. \quad (9)$$

To compute $X_{k+1}$, we use the identity (Lemma H.1 in the appendix) connecting the proximal operator of the Moreau envelope to the proximal operator of $r_1$:

$$X_{k+1} = \frac{1}{\gamma + \lambda} \left(\gamma \operatorname{Prox}_{(\gamma+\lambda)r_1}(Y_{k+1}) + \lambda Y_{k+1}\right). \quad (10)$$

Now DC-LA has been decomposed into elementary operators (also see Appendix C for a complete unroll). For the analysis of DC-LA in the next section, we work directly with $\nabla r_2^\lambda$ and $\operatorname{Prox}_{\gamma r_1^\lambda}(x)$ for convenience.

*Remark* 3.3. The backward step of DC-LA can be also written as $X_{k+1} = Y_{k+1} - \gamma \nabla r_1^{\lambda+\gamma}(Y_{k+1})$. Thus, DC-LA can be viewed as two gradient steps on Moreau envelopes: noise is injected in the first step, while the second step is deterministic. Moreover, the second step is stabilizing, since its step size $\gamma$ is always smaller than the Moreau-envelope parameter $\lambda + \gamma$.

## 4. Convergence Analysis of DC-LA

We begin with a convergence analysis in the general setting where both $r_1$ and $r_2$ are nonsmooth in Section 4.1. We then specialize to the case where $r_2$ is smooth and no longer needs to be regularized by Moreau envelope in Section 4.2.

### 4.1. Nonsmooth $r_2$

We impose the following assumption on $r_2$.

**Assumption 4.** $r_2$ satisfies one of the followings:

(i) There exists $G_2 > 0$ such that $\forall x \in \mathbb{R}^d, \forall z \in \partial r_2(x)$, it holds $\|z\| \le G_2$.

(ii) $r_2$ is differentiable whose gradient is $(\kappa, M)$-Hölder continuous with $\kappa \in (0, 1)$ and $M > 0$.

See Table 1 for regularizers that satisfy this assumption.

The following lemma (proved in Appendix I.1) provides a more concrete characterization of the distant dissipativity of $V$ under Assumption 4.

**Lemma 4.1.** *Under Assumptions 1, 3 and 4, $V$ is distant dissipative (7) iff $f$ is distant dissipative.*

We first analyze the sequence $\{Y_k\}_k$ of DC-LA. Similar to (Renaud et al., 2025a), we have the following lemma whose proof is in Appendix I.2.

**Lemma 4.2.** $\{Y_k\}_k$ *is an instance of the general ULA,*

$$Y_{k+1} = Y_k - \gamma b_\lambda^\gamma(Y_k) + \sqrt{2\gamma} Z_{k+1} \qquad (11)$$

*where the drift is* $b_\lambda^\gamma(y) := \nabla r_1^\lambda(\mathrm{Prox}_{\gamma r_1^\lambda}(y)) + \nabla f(\mathrm{Prox}_{\gamma r_1^\lambda}(y)) - \nabla r_2^\lambda(\mathrm{Prox}_{\gamma r_1^\lambda}(y))$. *Furthermore,* $b_\lambda^\gamma$ *is* $(\frac{2}{\lambda} + L_f)$-*Lipschitz.*

In the following lemma whose proof is in Appendix I.3, we show that $b_\lambda^\gamma$ is distant dissipative.

**Lemma 4.3.** *Let* $\gamma_0 > 0$. *Under Assumptions 1, 2, 3, 4, for all* $\gamma \in (0, \gamma_0]$, *and* $x, y$ *satisfying the distance condition:* $\|x - y\| \geq \max\left\{R_0, \frac{8G_1 + 4L_f\gamma_0 G_1 + 4G_2}{\mu}\right\}$ *if Assumption 4(i);* $\|x - y\| \geq \max\left\{R_0, \frac{16G_1 + 8L_f\gamma_0 G_1}{\mu}, \left(\frac{4M}{\mu}\right)^{\frac{1}{1-\kappa}}\right\}$ *if Assumption 4(ii), it holds*

$$\langle b_\lambda^\gamma(x) - b_\lambda^\gamma(y), x - y \rangle \geq \frac{\mu}{2}\|x - y\|^2$$

*where* $G_1$ *is given in Assumption 3,* $R_0, \mu$ *are given in Assumption 2, and* $G_2, M, \kappa$ *are given in Assumption 4 .*

We next define another drift

$$\bar{b}_\lambda^\gamma(y) = \nabla r_1^\lambda(\mathrm{Prox}_{\gamma r_1^\lambda}(y)) + \nabla f(y) - \nabla r_2^\lambda(y), \quad (12)$$

and the corresponding unadjusted Langevin algorithm:

$$\bar{Y}_{k+1} = \bar{Y}_k - \gamma \bar{b}_\lambda^\gamma(\bar{Y}_k) + \sqrt{2\gamma} \bar{Z}_{k+1}. \qquad (13)$$

with $\bar{Y}_1 \sim Y_1$. We also denote

$$\pi_{\lambda,\gamma}(x) \propto \exp(-f(x) - (r_1^\lambda)^\gamma(x) + r_2^\lambda(x)) \qquad (14)$$

and

$$\nu_{\lambda,\gamma} = (\mathrm{Prox}_{\gamma r_1^\lambda})_{\#}\pi_{\lambda,\gamma}. \qquad (15)$$

Note that $-\nabla \log \pi_{\lambda,\gamma} = \bar{b}_\lambda^\gamma$. In a similar manner, we can show that: for all $\gamma > 0$, $\bar{b}_\lambda^\gamma$ is $(\frac{2}{\lambda} + L_f)$-Lipschitz; and for all $\lambda > 0$, for all $\gamma > 0$, $\langle \bar{b}_\lambda^\gamma(x) - \bar{b}_\lambda^\gamma(y), x - y \rangle \geq \frac{\mu}{2}\|x - y\|^2$ whenever $\|x - y\| \geq \max\left\{R_0, \frac{8G_1 + 4G_2}{\mu}\right\}$ if Assumption 4(i); $\|x - y\| \geq \max\left\{R_0, \frac{16G_1}{\mu}, \left(\frac{4M}{\mu}\right)^{\frac{1}{1-\kappa}}\right\}$ if Assumption 4(ii). For completeness, we provide proof in Appendix I.4.

Based on the Lipschitz smoothness and distant dissipativity of these general drifts, the stability results of (Renaud et al., 2025a) apply. Leveraging these results, we derive Theorem 4.4 which quantifies the Wasserstein distance between the distributions of iterates $\{Y_k\}_k$ (resp. $\{X_k\}_k$) of DC-LA and $\pi_{\lambda,\gamma}$ (resp. $\nu_{\lambda,\gamma}$). Detailed proof is in Appendix I.5.

**Theorem 4.4.** *Let* $q \in \mathbb{N}^*$ *and* $\lambda > 0$, *under Assumptions 1, 2, 3, 4, for* $\{(X_k, Y_k)\}_k$ *being DC-LA's sequence starting from* $X_0$ *with* $\mathbb{E}\|X_0\| < +\infty$ *if* $q = 1$ *and* $\mathbb{E}\|X_0\|^{2q} < +\infty$ *if* $q \geq 2$. *Let* $\gamma$ *satisfy:* $\gamma \leq \frac{\mu\lambda^2}{2(2+\lambda L_f)^2}$ *if* $q = 1$ *and* $\gamma \leq \min\left\{\frac{\mu\lambda^2}{(2+\lambda L_f)^2 2^{2q+3}(2q-1)}, \frac{\lambda}{4(2+\lambda L_f)}\right\}$ *if* $q \geq 2$. *There exist* $A_\lambda, B_\lambda > 0$ *and* $\rho_\lambda \in (0, 1)$ *such that*

$$W_q(p_{X_{k+1}}, \nu_{\lambda,\gamma}) \leq W_q(p_{Y_{k+1}}, \pi_{\lambda,\gamma}) \leq A_\lambda \rho_\lambda^{k\gamma} + B_\lambda \gamma^{\frac{1}{2q}}$$

*where* $\pi_{\lambda,\gamma}$ *and* $\nu_{\lambda,\gamma}$ *are defined in (14) and (15).*

The dependence of the bound on the key parameters $\lambda, L_f, \mu, R, G_1, G_2, M$ is discussed in Appendix I.7 for case $q = 1$ for simplicity. In terms of the step size condition, it comes mainly from the Wasserstein contraction of the Markov transition kernels (De Bortoli & Durmus, 2019): $\gamma \leq \bar{\gamma} := \frac{m^+}{L^2}$, where $m^+$ is the distant dissipativity constant of the drift and $L$ is its Lipschitz constant. In our setting, $m^+ = \mu/2$ and $L = 2/\lambda + L_f$. On the other hand, when $q \geq 2$, an additional step size restriction is needed to ensure that all relevant distributions remain in the Wasserstein-$q$ space.

*Remark* 4.5. The side process $\{\bar{Y}_k\}$ in (13) also defines a valid sampler with quantifiable behaviors. It corresponds to applying the forward step simultaneously to two Moreau envelopes, rather than performing two separate steps as in DC-LA. In this work, however, we use it only as an interim sequence to facilitate the analysis.

Finally, we quantify $W_q(\pi_{\lambda,\gamma}, \pi_\lambda)$ and $W_q(\pi_\lambda, \pi)$. Since $\nabla r_1^\lambda(x) \in \partial r_1(\mathrm{Prox}_{\lambda r_1}(x))$, under the assumption 3, $\|\nabla r_1^\lambda(x)\| \leq G_1$ for all $x$, which means that $r_1^\lambda$ is $G_1$-Lipschitz continuous. Applying (Renaud et al., 2025a, Proposition 1), $W_q(\pi_{\lambda,\gamma}, \pi_\lambda) = O(\gamma^{1/q})$ and $W_q(\nu_{\lambda,\gamma}, \pi_\lambda) = O(\gamma^{1/q})$[4]. Under the Assumption 4, we show in Appendix I.6 that: $W_q(\pi_\lambda, \pi) = O(\lambda^{1/q})$. Putting these bounds together, we derive the following bound.

**Theorem 4.6.** *Let* $q \in \mathbb{N}^*$, *under Assumptions 1, 2, 3, 4, for* $0 < \lambda \leq \lambda_0$ *for some* $\lambda_0 > 0$, *let* $\{X_k\}$ *be the DC-LA's sequence starting from* $X_0$ *with* $\mathbb{E}\|X_0\| < +\infty$ *if* $q = 1$ *and* $\mathbb{E}\|X_0\|^{2q} < +\infty$ *if* $q \geq 2$. *For* $\gamma$ *satisfying the condition as in Theorem 4.4, there exist* $A_\lambda, B_\lambda', C > 0$ *and* $\rho_\lambda \in (0, 1)$,

$$W_q(p_{X_{k+1}}, \pi) \leq A_\lambda \rho_\lambda^{k\gamma} + B_\lambda' \gamma^{\frac{1}{2q}} + C\lambda^{\frac{1}{q}}.$$

$A_\lambda$ is the same as in Theorem 4.4 and $B_\lambda' = B_\lambda + O(1)$. See Appendix I.7 for the dependence of this bound on the key parameters. In Theorem 4.6, the bound converges to zero in a stage-wise fashion. The third term scales as $O(\lambda^{1/q})$. With $\lambda$ fixed, the second term scales as $O(\gamma^{1/(2q)})$. When both $\lambda$ and $\gamma$ are fixed, the first term vanishes with $k$.

---

[4]We remove the condition $\gamma \leq 2/G_1^2$ in (Renaud et al., 2025a, Proposition 1) for simplicity.

## 4.2. Smooth $r_2$

Yao & Kwok (2018) identified a structured family of DC regularizers with a Lipschitz continuous first component and a smooth second component, consisting of the Geman penalty (Geman & Yang, 1995), log-sum penalty (Candes et al., 2008), Laplace (Trzasko & Manduca, 2008), Minimax Concave Penalty (Zhang, 2010a), and Smoothly Clipped Absolute Deviation (Fan & Li, 2001), see Table 1. When $r_2$ has some smoothness, we only need to approximate $r_1$ by its Moreau envelope, resulting in the augmented potential $V_\lambda = f - r_2 + r_1^\lambda$ and the corresponding scheme named DC-LA-S(implified)

$$Y_{k+1} = X_k - \gamma \nabla f(X_k) + \gamma \nabla r_2(X_k) + \sqrt{2\gamma} Z_{k+1}$$
$$X_{k+1} = \mathrm{Prox}_{\gamma r_1^\lambda}(Y_{k+1}). \tag{16}$$

Applying a similar analysis in the general case, we obtain Theorem 4.7. Proof is given in Appendix I.8.

**Theorem 4.7.** *Let $q \in \mathbb{N}^*$. Under Assumptions 1, 2, 3, and $r_2$ being $L_{r_2}$-smooth, for $\lambda > 0$, let $\{X_k\}$ be DC-LA-S's sequence starting from $X_0$ with $\mathbb{E}\|X_0\| < +\infty$ if $q = 1$ and $\mathbb{E}\|X_0\|^{2q} < +\infty$ if $q \geq 2$. Let $\gamma$ satisfy: $\gamma \leq \frac{\mu\lambda^2}{2(1+\lambda L_f + \lambda L_{r_2})^2}$ if $q = 1$ and $\gamma \leq \min\left\{\frac{\mu\lambda^2}{(1+\lambda L_f + \lambda L_{r_2})^2 2^{2q+3}(2q-1)}, \frac{\lambda}{4(1+\lambda L_f + \lambda L_{r_2})}\right\}$ if $q \geq 2$. There exist $A''_\lambda, B''_\lambda, C'' > 0$ and $\rho_\lambda \in (0,1)$ such that*

$$W_q(p_{X_{k+1}}, \pi) \leq A''_\lambda \rho_\lambda^{k\gamma} + B''_\lambda \gamma^{\frac{1}{2q}} + C'' \lambda^{\frac{1}{q}}.$$

# 5. Experiments

We study sampling problems using an $\ell_1 - \ell_2$ prior (Yin et al., 2015) on synthetic data (Gaussian likelihoods with varying means and covariance matrices) in a 2D setting to enable density visualization in Subsection 5.1, and a DCINNs prior (Zhang & Leong, 2025) on real Computed Tomography data of size $512 \times 512$ in Subsection 5.2. See Appendix J for additional experiments and details.[5]

## 5.1. $\ell_1 - \ell_2$ prior

We consider the potential of the form $V(x) = f(x) + \tau(\|x\|_1 - \|x\|_2)$ where $f(x) = \frac{1}{2}(x - \mu)^\top \Sigma (x - \mu)$ for some $\Sigma \succ 0$ and $\tau > 0$ controlling the sparsity level. The proximal operators of $\|\cdot\|_1$ and $\|\cdot\|_2$ are available in closed-form, known as the soft thresholding operator and the block soft thresholding operator, respectively (Appendix J). Furthermore, both $\|\cdot\|_1$ and $\|\cdot\|_2$ are Lipschitz and Lemma 4.1 applies, implying that $V$ is distant dissipative.

**Baselines** Since $r$ is not weakly convex, PSGLA does not come with convergence guaranties. Nevertheless, because

the proximal operator of $r$ admits a closed-form expression (Lou & Yan, 2018), we include a comparison of our scheme against PSGLA. For reference, we also run ULA on both the nonsmooth potential $V$ and its smoothed surrogate $V_\lambda$, the latter referred to as Moreau ULA.

*Remark 5.1.* Each DC-LA iteration requires one gradient evaluation of $f$ and one proximal evaluation each for $r_1$ and $r_2$, while ULA requires evaluations of $\nabla f$, $\partial r_1$, and $\partial r_2$. In this experiment, the $\ell_1$ and $\ell_2$ proximal maps are available in closed form, so DC-LA has a per-iteration complexity comparable to ULA. Without closed-form proximal maps, DC-LA may require convex solvers, and thus incur higher costs. DC-LA and Moreau ULA have the same per-iteration complexity.

**Setups** For the target distribution, we set $\tau = 10$, $\Sigma = \begin{bmatrix} 1 & 0.8 \\ 0.8 & 1 \end{bmatrix}$, $\mu \in \{[0,0]^\top, [1,1]^\top, [2,2]^\top\}$. For the augmented potential $V_\lambda$, we set $\lambda = 0.01$. Although our theoretical analysis provides a sufficient condition on the step size $\gamma$ for convergence of DC-LA, in this experiment, we fix a common step size $\gamma = 0.005$ for all methods across all datasets. This value yields stable behavior and reasonable mixing for three algorithms and is used as a neutral default rather than a performance-optimized choice. For each algorithm, we run 5000 chains of length 1000 and keep the last samples. In Appendix J.2, we conduct an ablation study on $(\lambda, \gamma)$, suggesting DC-LA is relatively robust.

**Results** Figure 1 shows the target density (the normalizing constant is computed by `dblquad` from `scipy`) and histograms of samples produced by the sampling algorithms. The effect of $\ell_1 - \ell_2$ prior can be seen clearly as it highlights coordinate axes, promoting sparsity. DC-LA faithfully captures the target density. On the other hand, ULA and Moreau ULA are blurry, while PSGLA is the sharpest but appears to overemphasize the coordinates. We compute the binned KL divergence [6] between the histograms produced by each sampling method and the binned target distribution using identical bins, and Figure 2 shows that DC-LA achieves the lowest KL divergence across bin resolutions.

## 5.2. Computed Tomography with DICNNs priors

We consider a Computed Tomography (CT) experiment using the Mayo Clinic's human abdominal CT scans (Moen et al., 2021), each of size $512 \times 512$. Given a ground truth CT scan $x^*$, the forward model is $y = Ax^* + \epsilon$ where $\epsilon \sim \mathcal{N}(0, \sigma^2 I)$ and $A$ is the data acquisition using a 2D parallel-beam ray transform implemented in ODL (Adler et al., 2017) with 350 uniform projections collected from $0°$ to $120°$. We use the data-driven DC prior based on the difference of two input-convex

---

[5]Our code is at https://github.com/MCS-hub/DC-LA2026.

[6]In this scenario, the Wasserstein distance is difficult to evaluate due to the absence of true samples from the target distribution.

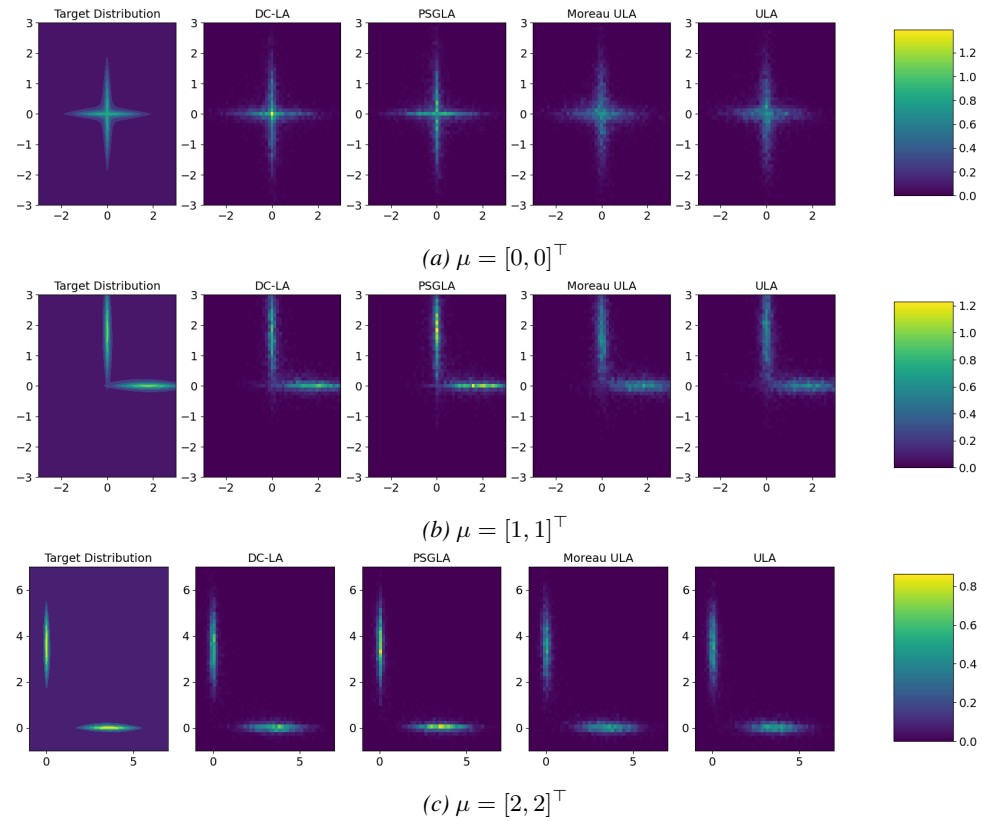

*(a)* $\mu = [0,0]^\top$

*(b)* $\mu = [1,1]^\top$

*(c)* $\mu = [2,2]^\top$

*Figure 1.* Target densities and histograms of samples produced by DC-LA, PSGLA, Moreau ULA, and ULA

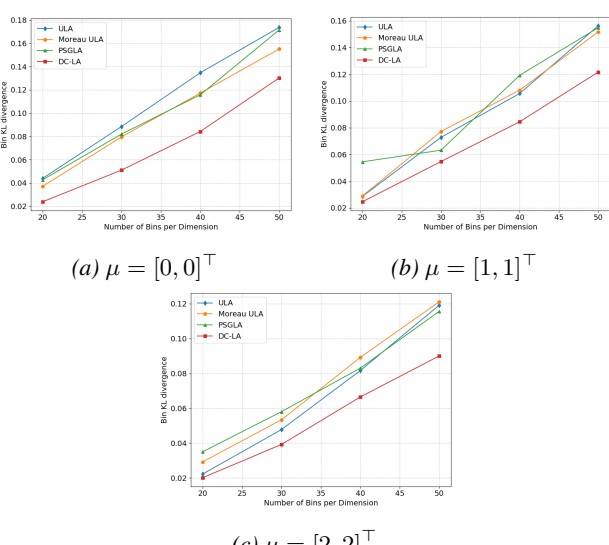

*(a)* $\mu = [0,0]^\top$

*(b)* $\mu = [1,1]^\top$

*(c)* $\mu = [2,2]^\top$

*Figure 2.* Binned KL divergences between samples from {ULA, Moreau ULA, PSGLA, DC-LA} and the target distributions

neural networks (ICNNs) proposed in (Zhang & Leong, 2025). The target distribution is of the form $\pi(x) \propto \exp\left(-\frac{1}{2\sigma^2}\|y - Ax\|^2 - \tau(\mathrm{NN}_1(\theta_1^*, x) - \mathrm{NN}_2(\theta_2^*, x))\right)$ where $\mathrm{NN}_1, \mathrm{NN}_2$ are two ICNNs with pretrained parameters $\theta_1^*, \theta_2^*$. The experiment is conditioned on $\sigma$, i.e., treating $\sigma$ as a known parameter. The DC regularizer $r(\theta, x) = \mathrm{NN}_1(\theta_1, x) - \mathrm{NN}_2(\theta_2, x)$ was trained using the adversarial regularization framework (Lunz et al., 2018) encouraging it to be Lipschitz. Therefore, we largely assume that each of its components is also Lipschitz. On the other hand, as $A$ is in general ill-conditioned, Lemma 4.1 does not apply directly. We can slightly augment $f$ to confine it at infinity. For example, we can add to $f$ a radial part $\phi(\|x\|)$, where $\phi(r) \approx 0$ for $r \leq R$ and $\phi(r) \approx (r - R)^2$ for $r > R$ with $R$ sufficiently large to preserve the main posterior mass. Nevertheless, in this experiment, we leave the posterior as-is.

**Setups** We set $\sigma = 0.2$, $\tau = \hat{\lambda}/\sigma^2$ where $\hat{\lambda} = \|A^*(Ax_{val} - y_{val})\|$ with one validation data point $(x_{val}, y_{val})$. For DC-LA, the proximal operators of $\mathrm{NN}_1$ and $\mathrm{NN}_2$ are not in closed-form. For a given $\eta > 0$, let $v^* = \mathrm{Prox}_{\eta \, \mathrm{NN}_i}(x)$, $v^*$ solves the fixed-point equation: $v^* \in x - \eta \partial \mathrm{NN}_i(v^*)$, we apply several fixed-point iterations: $v := x - \eta \partial \mathrm{NN}_i(v)$. In practice, we found that 1

iteration is sufficient, and we use it for both proximal operators. Note that Ehrhardt et al. (2024) suggested that bounded approximation errors in computing proximal operators lead to controlled bias; extending our analysis to this setting is a promising direction for future work. The step size of DC-LA is $\gamma = 10^{-5}\sigma^2$ and the smoothing parameter $\lambda = 10^{-4}$. We simulate 100 Markov chains, each of length 2000, initialized from a black image and retaining only the final sample from each chain. We also compute the MAP estimate by iterating $x_{k+1} = \text{Prox}_{\gamma r_1}(x_k - \gamma\nabla f(x_k) + \eta\partial r_2(x_k))$ (called PSM – Proximal Subgradient Method (Zhang & Leong, 2025)), which is the optimization counterpart of DC-LA. The optimizer runs for up to $20,000$ iterations.

**Results** In Figure 3 (more results are given in Appendix J.3), panel (a) shows the ground-truth image, panel (b) presents the posterior mean produced by DC-LA, panel (c) shows the estimated pixel-wise posterior variance produced by DC-LA, while panels (d), (e), (f) show the MAP reconstructions produced by PSM at iterations $2000, 10000, 20000$ iterations.

The posterior mean from DC-LA (with 100 parallel chains) successfully recovers the overall anatomical structures. It achieves *higher* SSIM (Structural Similarity Index Measure) and PSNR (Peak Signal-to-Noise Ratio) [7] than the MAP estimate obtained by PSM at 2000 iterations, which matches the length of the DC-LA chains. At 10000 iterations, PSM roughly matches the performance of DC-LA. We run the PSM until the $20,000$ iterations, when it slightly surpasses DC-LA. The PSM result at 20000 iterations improves over that at 10000 iterations, although there are signs of overfitting by 20000 iterations (Figure 4). The posterior variance produced by DC-LA highlights regions of higher uncertainty, primarily around boundaries and fine textures, indicating where the reconstruction is less confident. In contrast, large homogeneous regions exhibit small variance, implying agreement between the likelihood and prior in these regions. However, the samples miss a small structure in the lower-left region, and the posterior variance does not highlight this area, suggesting either that the samples are not sufficiently representative of the posterior or that the posterior itself is overconfident in this region.

## 6. Conclusion

We propose a mathematically justified sampler DC-LA for nonsmooth DC regularizers not necessarily weakly convex, extending beyond log-concave settings and ensuring convergence under distant dissipativity. DC-LA is able to produce

---

[7]SSIM and PSNR are reported as rough indicators of reconstruction quality; higher values are better. They should not be interpreted as metrics of sampling accuracy, and their values are strongly influenced by the choice of prior.

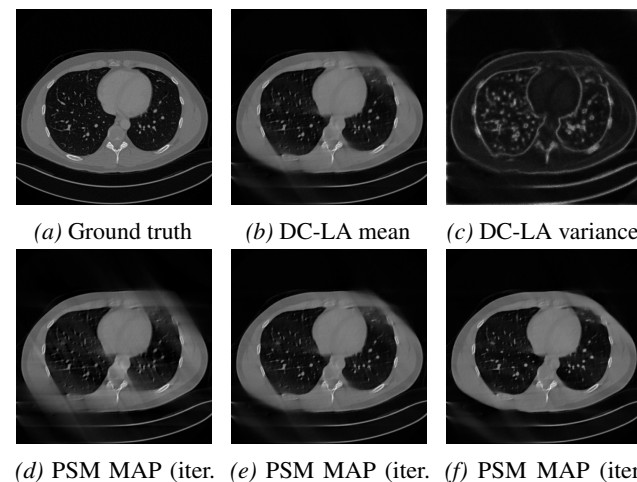

*(a)* Ground truth    *(b)* DC-LA mean    *(c)* DC-LA variance

*(d)* PSM MAP (iter. 2k)    *(e)* PSM MAP (iter. 10k)    *(f)* PSM MAP (iter. 20k)

*Figure 3.* CT reconstruction results. DC-LA mean achieves SSIM 0.8493 and PSNR 26.2303. PSM MAP with iterations: 2k (SSIM 0.7586, PSNR 24.2803), 10k (SSIM 0.8488, PSNR 26.2498), 20k (SSIM 0.8501, PSNR 26.5246).

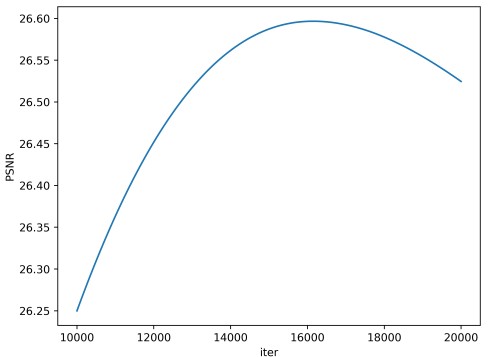

*Figure 4.* PSNR curve of PSM

sensible variance images in a computed tomography application with DICNN priors, complementing MAP estimation in this setting. Future work will explore the practical use of these variance images in downstream applications, e.g., (Jun et al., 2025). For the analysis, a natural extension is to relax the Lipschitz continuity of $r_1$ to similar conditions on $r_2$. The current analysis does not cover this case. However, for mixed-growth $r_1$, such as the elastic net, one may move the smooth quadratic part to the forward step and reuse the present framework. We leave this for future work.

## Acknowledgments

The authors thank Marien Renaud for the helpful discussion, Yasi Zhang for the pretrained parameters of DICNNs, and the funding support from NTU-SUG and Singapore Ministry of Education (MOE) AcRF Tier 1 RG17/24. We thank the anonymous reviewers for their constructive feedback.

## Impact Statement

This paper presents work whose goal is to advance the field of machine learning. There are many potential societal consequences of our work, none of which we feel must be specifically highlighted here.

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

## A. Sub-Gaussian tails

Let $V$ satisfy Assumptions 1 and 2, we show that $\pi \propto e^{-V}$ has sub-Gaussian tails. Let $g_1(x) \in \partial r_1(x)$ and $g_2(x) \in \partial r_2(x)$ be some measurable selections and let $g(x) := \nabla f(x) + g_1(x) - g_2(x)$, under Assumption 2: there exist $R_0 \geq 0, \mu > 0$ such that

$$\langle g(x) - g(y), x - y \rangle \geq \mu \|x - y\|^2$$

for all $\|x - y\| \geq R_0$. Now fix $x$ such that $\|x\| > R_0$, we have

$$\langle g(tx) - g(0), x \rangle \geq \mu t \|x\|^2$$

whenever $t \geq R_0/\|x\|$. Let $t_0 := R_0/\|x\| < 1$, by applying Mean Value Theorem (MVT) to $f, r_1, r_2$ (see (Hiriart-Urruty & Lemaréchal, 2004, Theorem 2.3.4) for MTV for convex functions), we get

$$
\begin{aligned}
V(x) - V(0) &= \int_0^1 \langle g(tx), x \rangle dt \\
&= \int_0^{t_0} \langle g(tx), x \rangle dt + \int_{t_0}^1 \langle g(tx), x \rangle dt \\
&\geq \int_0^{t_0} \langle g(tx), x \rangle dt + \int_{t_0}^1 (\mu t \|x\|^2 + \langle g(0), x \rangle) dt \\
&= \int_0^{t_0} \langle g(tx), x \rangle dt + \frac{\mu}{2}(1 - t_0^2)\|x\|^2 + (1 - t_0)\langle g(0), x \rangle \\
&\geq \int_0^{t_0} \langle g(tx), x \rangle dt + \frac{\mu}{2}\|x\|^2 - \frac{\mu}{2}R_0^2 - \|g(0)\|\|x\|.
\end{aligned}
$$

On the other hand, $\|g(tx)\|$ is bounded for all $t \in [0, t_0]$. Indeed, since $f$ is $L_f$-smooth,

$$
\begin{aligned}
\|\nabla f(tx)\| &\leq L_f t\|x\| + \|\nabla f(0)\| \\
&\leq L_f R_0 + \|\nabla f(0)\|.
\end{aligned}
$$

Since $r_i$ is convex, its subgradients are bounded locally: $\|g_i(tx)\| \leq M_i$ for all $t \in [0, t_0]$.

Therefore, there exists $C, D > 0$ such that

$$
\begin{aligned}
V(x) - V(0) &\geq \frac{\mu}{2}\|x\|^2 - C\|x\| - \frac{\mu}{2}R_0^2 \\
&\geq \frac{\mu}{4}\|x\|^2 - D,
\end{aligned}
$$

implying that $\pi$ has sub-Gaussian tails. This further implies $\pi^\lambda$ (8) and $\pi^{\lambda,\gamma}$ (14) have sub-Gaussian tails. Indeed, it holds $V_\lambda(x) \geq V(x) - (\lambda G_1^2)/2$ for all $x$. On the other hand, $\nabla r_1^\lambda(x) \in \partial r_1(\text{Prox}_{\lambda r_1}(x))$, so $\|\nabla r_1^\lambda(x)\| \leq G_1$ for all $x$. Therefore,

$$f(x) + (r_1^\lambda)^\gamma(x) - r_2^\lambda(x) \geq f(x) + r_1^\lambda(x) - r_2^\lambda(x) - \frac{\gamma G_1^2}{2}. \tag{17}$$

## B. Finite moments of random variables

We first show that: if $X_0$ has finite $p'$ moment ($p' \geq 1$), $\{X_k\}, \{Y_k\}$ generated by DC-LA have finite $p'$-th moment.

By induction, suppose that $\mathbb{E}\|X_k\|^{p'} < +\infty$, we show $\mathbb{E}\|Y_{k+1}\|^{p'} < +\infty$ and $\mathbb{E}\|X_{k+1}\|^{p'} < +\infty$. Indeed, by applying Minkowski inequality,

$$
\begin{aligned}
(\mathbb{E}\|Y_{k+1}\|^{p'})^{\frac{1}{p'}} &\leq (\mathbb{E}\|X_k\|^{p'})^{\frac{1}{p'}} + \gamma(\mathbb{E}\|\nabla f(X_k)\|^{p'})^{\frac{1}{p'}} + \gamma(\mathbb{E}\|\nabla r_2^\lambda(X_k)\|^{p'})^{\frac{1}{p'}} + \sqrt{2\gamma}(\mathbb{E}\|Z_{k+1}\|^{p'})^{\frac{1}{p'}} \\
&\leq (\mathbb{E}\|X_k\|^{p'})^{\frac{1}{p'}} + \gamma(\|\nabla f(0)\| + L_f(\mathbb{E}\|X_k\|^{p'})^{\frac{1}{p'}}) \\
&\quad + \gamma\left(\|\nabla r_2^\lambda(0)\| + \frac{1}{\lambda}(\mathbb{E}\|X_k\|^{p'})^{\frac{1}{p'}}\right) + \sqrt{2\gamma}(\mathbb{E}\|Z_{k+1}\|^{p'})^{\frac{1}{p'}} < +\infty.
\end{aligned}
$$

Then, using the non-expansiveness of the proximal operator,

$$\left(\mathbb{E}\|X_{k+1}\|^{p'}\right)^{\frac{1}{p'}} = \left(\mathbb{E}\|\operatorname{Prox}_{\gamma r_1^{\lambda}}(Y_{k+1})\|^{p'}\right)^{\frac{1}{p'}} \leq \|\operatorname{Prox}_{\gamma r_1^{\lambda}}(0)\| + \left(\mathbb{E}\|Y_{k+1}\|^{p'}\right)^{\frac{1}{p'}} < +\infty.$$

Similarly, $\mathbb{E}\|\bar{Y}_k\|^{p'} < +\infty$ where $\{\bar{Y}_k\}$ is define in (13) if $\mathbb{E}\|\bar{Y}_1\|^{p'} < +\infty$.

## C. Unrolled DC-LA

The unrolled form of DC-LA is

$$
\begin{aligned}
X_{k+1} =& X_k - \frac{\gamma\lambda}{\gamma+\lambda}\nabla f(X_k) - \frac{\gamma}{\gamma+\lambda}\operatorname{Prox}_{\lambda r_2}(X_k) + \frac{\lambda\sqrt{2\gamma}}{\gamma+\lambda}Z_{k+1} \\
&+ \frac{\gamma}{\gamma+\lambda}\operatorname{Prox}_{(\lambda+\gamma)r_1}\left(\frac{\lambda+\gamma}{\lambda}X_k - \gamma\nabla f(X_k) - \frac{\gamma}{\lambda}\operatorname{Prox}_{\lambda r_2}(X_k) + \sqrt{2\gamma}Z_{k+1}\right).
\end{aligned}
$$

## D. DC regularizers

We first recall some non-convex regularizers and their DC decompositions. We then show that the first DC component of each regularizer is Lipschitz continuous, while the second is either Lipschitz continuous, or is differentiable with Hölder or Lipschitz continuous gradient. Hence, Theorems 4.6 and 4.7 can be invoked accordingly, provided the distant dissipative assumptions are satisfied.

### D.1. DC regularizers with two nonsmooth components

(1) $\ell_1 - \ell_2$ **regularizer** (Yin et al., 2015; Lou & Yan, 2018)

$$r(x) = \|x\|_1 - \|x\|_2.$$

(2) $\ell_1 - \ell_{\sigma_q}$ **regularizer** (Luo et al., 2013) We define $\|x\|_{\sigma_q} = \sum_{i=1}^{q}|x_{[i]}|$ where $x_{[i]}$ represents the i-th element of $x$ in descending order of magnitude. Let

$$r(x) = \|x\|_1 - \|x\|_{\sigma_q}.$$

(3) **Capped $\ell_1$** (Zhang, 2010b; Le Thi et al., 2015) Let $\theta > 0$ be a parameter. We define

$$r(x) = \sum_{i=1}^{d}\min\{1, \theta|x_i|\} = \theta\|x\|_1 - \sum_{i=1}^{d}\left(\theta|x_i| - \min\{1, \theta|x_i|\}\right).$$

(4) **PiL** (Le Thi et al., 2015) Let $\theta > 0, a > 1$ be parameters, we define

$$
\begin{aligned}
r(x) =& \sum_{i=1}^{d}\min\left\{1, \max\left\{0, \frac{\theta|x_i|-1}{a-1}\right\}\right\} \\
=& \frac{\theta}{a-1}\sum_{i=1}^{d}\max\left\{\frac{1}{\theta}, |x_i|\right\} - \sum_{i=1}^{d}\left(\frac{\theta}{a-1}\max\left\{\frac{1}{\theta}, |x_i|\right\} - \min\left\{1, \max\left\{0, \frac{\theta|x_i|-1}{a-1}\right\}\right\}\right).
\end{aligned}
$$

(5) $\ell_1 - \ell_2^p$ **regularizer** $(1 < p < 2)$

$$r(x) = \|x\|_1 - \|x\|_2^p.$$

The regularizers (1), (2), (3), (4) have both non-differentiable DC components, while the regularizer (5) has non-differentiable $r_1$ and differentiable $r_2$ whose gradient is Hölder continuous.

Since $\ell_1$ is $\sqrt{d}$-Lipschitz continuous ($d$ is the dimension), the first DC components of $\ell_1 - \ell_2$, $\ell_1 - \ell_{\sigma_q}$, $\ell_1 - \ell_2^p$ are $\sqrt{d}$-Lipschitz continuous and that of Capped $\ell_1$ is $\theta\sqrt{d}$-Lipschitz continuous. Regarding the first DC component of PiL, it is straightforward to verify that

$$\left| \max\left\{\frac{1}{\theta}, |u|\right\} - \max\left\{\frac{1}{\theta}, |v|\right\} \right| \le |u - v|, \quad \forall u, v \in \mathbb{R}.$$

Therefore, the first DC component of PiL is $\frac{\theta}{a-1}\sqrt{d}$-Lipschitz continuous.

Now the second DC component of the $\ell_1 - \ell_2$ regularizer is 1-Lipschitz continuous, while the second DC component of the $\ell_1 - \ell_{\sigma_q}$ regularizer is $\sqrt{d}$-Lipschitz continuous. The latter is due to

$$|\|x\|_{\sigma_q} - \|y\|_{\sigma_q}| \le \|x - y\|_{\sigma_q} \le \|x - y\|_1 \le \sqrt{d}\|x - y\|_2.$$

It is direct to verify that $|\min\{1, \theta|u|\} - \min\{1, \theta|v|\}| \le \theta|u - v|$, therefore the second DC component of Capped $\ell_1$ is $2\theta\sqrt{d}$-Lipschitz continuous. We then have

$$\begin{aligned}
&\left| \min\left\{1, \max\left\{0, \frac{\theta|u| - 1}{a - 1}\right\}\right\} - \min\left\{1, \max\left\{0, \frac{\theta|v| - 1}{a - 1}\right\}\right\} \right| \\
&\le \left| \max\left\{0, \frac{\theta|u| - 1}{a - 1}\right\} - \max\left\{0, \frac{\theta|v| - 1}{a - 1}\right\} \right| \\
&\le \frac{\theta}{a - 1}|u - v|.
\end{aligned}$$

Therefore, the second component of PiL is $2\frac{\theta}{a-1}\sqrt{d}$-Lipschitz continuous.

Lastly, $r_2(x) := \|x\|^p$ has $(p - 1, C)$-Hölder continuous gradient for some $C > 0$. Indeed, the gradient is given by $\|x\|^{p-2}x$ if $x \ne 0$ and 0 if $x = 0$. Indeed, as a standard result, there exists $C > 0$ such that

$$|\|x\|^{p-2}x - \|y\|^{p-2}y| \le C\|x - y\|^{p-1}, \quad \forall x, y \ne 0.$$

For completeness, we give a proof. We consider two cases: (a) $\|x - y\| \ge \frac{1}{2}\max\{\|x\|, \|y\|\}$ and (b) $\|x - y\| < \frac{1}{2}\max\{\|x\|, \|y\|\}$.

Under case (a),

$$|\|x\|^{p-2}x - \|y\|^{p-2}y| \le \|x\|^{p-1} + \|y\|^{p-1} \le 2^p\|x - y\|^{p-1}.$$

Under case (b), without loss of generality, assume that $\|x\| \ge \|y\|$. Note that $[x, y]$ cannot contain the origin. Applying Mean Value Theorem, there exists $z \in [x, y]$:

$$\begin{aligned}
|\|x\|^{p-2}x - \|y\|^{p-2}y| &\le \|\nabla^2 r_2(z)\|\|x - y\| \\
&= \left\|p\|z\|^{p-2}I + p(p - 2)\|z\|^{p-4}zz^\top\right\|\|x - y\| \\
&\le C'\|z\|^{p-2}\|x - y\|.
\end{aligned}$$

On the other hand

$$\|z\| \ge \|x\| - \|x - z\| \ge \|x\| - \|x - y\| > \frac{1}{2}\|x\|.$$

Therefore,

$$\|z\|^{p-2}\|x - y\| \le C''\|x\|^{p-2}\|x - y\| \le C\|x - y\|^{p-1}.$$

## D.2. DC regularizers with nonsmooth first DC component and smooth second DC component

Yao & Kwok (2018) identified a family of DC regularizers whose first component is nonsmooth and second is smooth. This family of regularizers can be written as

$$r(x) = \sum_{i=1}^{K} \mu_i \kappa(\|A_i x\|_2)$$

where $\mu_i \geq 0$ for all $i = \overline{1, K}$ and $A_i$ are matrices (to induce structured sparsity like group LASSO, fused LASSO or graphical LASSO (Yao & Kwok, 2018)), $\kappa : [0, \infty) \to [0, \infty)$ such that $\kappa$ is concave, non-decreasing, $\rho$-smooth for some $\rho > 0$ with $\kappa'$ non-differentiable at finite many points, $\kappa(0) = 0$. As shown in Table 1 in (Yao & Kwok, 2018), this structure covers German penalty (GP), log-sum penalty (LSP), Laplace, Minimax Concave Penalty (MCP), and Smoothly Clipped Absolute Deviation (SCAD).

Let $\kappa_0 := \kappa'(0)$, $r$ has the following DC decomposition

$$r(x) = \kappa_0 \sum_{i=1}^{K} \mu_i \|A_i x\|_2 - \sum_{i=1}^{K} (\kappa_0 \mu_i \|A_i x\|_2 - \mu_i \kappa(\|A_i x\|_2)).$$

The first DC component of $r$ is nonsmooth, while the second can be shown to be smooth (Yao & Kwok, 2018). We further see that

$$|\|A_i x\|_2 - \|A_i y\|_2| \leq \|A_i x - A_i y\|_2 \leq \|A_i\|_F \|x - y\|_2$$

where $\|A_i\|_F$ is the Frobenius norm of $A_i$. Therefore the first DC component of $r$ is $\kappa_0(\sum \mu_i \|A_i\|_F)$-Lipschitz continuous.

# E. On Assumption 3 in (Renaud et al., 2025a)

Let $\rho > 0$, suppose that $r$ is $\rho$-weakly convex, i.e., $r + \frac{\rho}{2}\|\cdot\|^2$ is convex.

We recall the assumption as follows.

*Assumption.* (i) $\forall \gamma \in (0, \frac{1}{\rho})$, $r$ is $L_r$-smooth on $\mathrm{Prox}_{\gamma r}(\mathbb{R}^d)$.

(ii) $r^\gamma$ is $\mu$-strongly convex at infinity with $\mu \geq 8L_f + 4L_r$, i.e, there exists $\gamma_1 > 0$ and $R_0 \geq 0$ such that $\forall \gamma \in (0, \gamma_1]$, $\nabla^2 r^\gamma \succeq \mu I$ on $\mathbb{R}^d \setminus B(0, R_0)$.

We show that this assumption does not hold. Let $\gamma \in (0, \frac{1}{\rho})$. For $x \in \mathbb{R}^d$, recall that

$$\mathrm{Prox}_{\gamma r}(x) = \mathrm{argmin}_y \left\{ r(y) + \frac{1}{2\gamma}\|y - x\|^2 \right\}.$$

By the first-order optimality condition

$$0 \in \partial \left( r + \frac{1}{2\gamma}\|\cdot - x\|^2 \right)(\mathrm{Prox}_{\gamma r}(x)). \tag{18}$$

By Assumption (i), $r$ is differentiable at $\mathrm{Prox}_{\gamma r}(x)$, (18) becomes

$$0 = \nabla r(\mathrm{Prox}_{\gamma r}(x)) + \frac{1}{\gamma}(\mathrm{Prox}_{\gamma r}(x) - x)$$

On the other hand, since $\gamma \rho < 1$, it follows from (Renaud et al., 2025a, Lemma 12),

$$\nabla r^\gamma(x) = \frac{1}{\gamma}(x - \mathrm{Prox}_{\gamma r}(x))$$

so $\nabla r^\gamma(x) = \nabla r(\mathrm{Prox}_{\gamma r}(x))$ for all $x$. Now that

$$\|\nabla r^\gamma(x) - \nabla r^\gamma(y)\| = \|\nabla r(\mathrm{Prox}_{\gamma r}(x)) - \nabla r(\mathrm{Prox}_{\gamma r}(y))\|$$

$$\leq L_r \|\mathrm{Prox}_{\gamma r}(x) - \mathrm{Prox}_{\gamma r}(y)\| \leq \frac{L_r}{1 - \gamma\rho}\|x - y\|$$

since $\mathrm{Prox}_{\gamma r}$ is $\frac{1}{1-\gamma\rho}$-Lipschitz (Renaud et al., 2025b, Lemma 11). Therefore $r^\gamma$ is $\frac{L_r}{1-\gamma\rho}$-smooth in $\mathbb{R}^d$.

From Assumption (ii), let $x, y$ be two points far away from the origin so that the segment $[x, y]$ does not intersect $B(0, R_0)$. Let $\gamma < \min\{\frac{3}{4\rho}, \gamma_1\}$,

$$\nabla r^\gamma(x) - \nabla r^\gamma(y) = \int_0^1 \nabla^2 r^\gamma(tx + (1-t)y)(x - y)dt.$$

Therefore

$$\langle x - y, \nabla r^\gamma(x) - \nabla r^\gamma(y)\rangle = \int_0^1 (x-y)^\top \nabla^2 r^\gamma(tx + (1-t)y)(x-y)dt \geq \mu\|x - y\|^2$$

On the other hand,

$$\langle x - y, \nabla r^\gamma(x) - \nabla r^\gamma(y)\rangle \leq \|x - y\|\|\nabla r^\gamma(x) - \nabla r^\gamma(y)\| \leq \frac{L_r}{1 - \gamma\rho}\|x - y\|^2.$$

These inequalities imply $\mu \leq \frac{L_r}{1-\gamma\rho}$. Therefore, the condition $\mu \geq 4L_r + 8L_f$ implies $\gamma \geq \frac{3}{4\rho}$ and this is a contradiction.

# F. Non-weakly-convex DC regularizers

We show the following DC regularizers are not weakly convex.

## F.1. $\ell_1 - \ell_2$

Let $r(x) = \|x\|_1 - \|x\|_2$. Suppose that there exists $\alpha > 0$ such that $u(x) := r(x) + \alpha\|x\|_2^2$ is convex.

Let's pick $x^*$ with positive entries such that $\|x^*\|_2 < \frac{1}{2\alpha}$. We have $\nabla^2\|\cdot\|_1(x^*) = 0$ and

$$\nabla^2\|\cdot\|_2(x^*) = \frac{1}{\|x^*\|_2}I - \frac{1}{\|x^*\|_2^3}x^*x^{*\top}.$$

Therefore,

$$\nabla^2 u(x^*) = 2\alpha I - \frac{1}{\|x^*\|_2}I + \frac{1}{\|x^*\|_2^3}x^*x^{*\top}.$$

Let $d \in \mathbb{R}^d, \|d\|_2 = 1$ such that $d$ is orthogonal to $x^*$, it follows that

$$d^\top \nabla^2 u(x^*)d = 2\alpha - \frac{1}{\|x^*\|_2} < 0.$$

So $u$ is not convex. This is a contradiction, we conclude that $r$ is not weakly convex.

## F.2. $\ell_1 - \ell_{\sigma_q}$

We define $\|x\|_{\sigma_q} = \sum_{i=1}^q |x_{[i]}|$ where $x_{[i]}$ represents the i-th element of $x$ ordered by magnitude. Note that when $q = 1$, $\|x\|_{\sigma_q} = \|x\|_\infty$.

Now let $r(x) = \|x\|_1 - \|x\|_{\sigma_q}$ and suppose that there exists $\alpha > 0$ such that $u(x) = r(x) + \alpha\|x\|_2^2$ is convex.

Let $A > 1$ and let $x$ such that

$$x_1 = x_2 = \ldots = x_{q-1} = A,$$
$$|x_q| < 1, \; |x_{q+1}| < 1,$$
$$x_{q+2} = \ldots = x_d = 0.$$

Hence $\|x\|_{\sigma_q} = (q-1)A + \max\{|x_q|, |x_{q+1}|\}$. Therefore

$$r(x) = |x_q| + |x_{q+1}| - \max\{|x_q|, |x_{q+1}|\} = \min\{|x_q|, |x_{q+1}|\}.$$

Consider the 2D slice, it would follow that the function $\bar{u}(x_q, x_{q+1}) = \min\{|x_q|, |x_{q+1}|\} + \alpha(x_q^2 + x_{q+1}^2)$ is convex in $(-1, 1) \times (-1, 1)$. Let $0 < \epsilon < 1$,

$$\bar{u}(\epsilon, 0) + \bar{u}(0, \epsilon) \geq 2\bar{u}\left(\frac{\epsilon}{2}, \frac{\epsilon}{2}\right)$$

reducing to

$$2\alpha\epsilon^2 \geq 2\left(\frac{\epsilon}{2} + \alpha\frac{\epsilon^2}{2}\right)$$

or $\alpha\epsilon \geq 1$. This cannot hold for small $\epsilon$. We get a contradiction.

### F.3. Capped-$\ell_1$

Let $\theta > 0$, we define

$$r(x) = \sum_{i=1}^{d} \min\{1, \theta|x_i|\}.$$

Since $\mathrm{Cap}\ell_1$ is separable, we only need to show it is not weakly convex in 1D.

Now in 1D, by contradiction, suppose there is $\alpha > 0$ such that $u(x) = r(x) + \alpha x^2$ is convex.

Let $\epsilon \in \left(0, \frac{1}{2\theta}\right)$ and we denote $x^+ = \frac{1}{\theta} + \epsilon$, $x^- = \frac{1}{\theta} - \epsilon > 0$. Mid-point inequality reads

$$2u\left(\frac{1}{2}(x^+ + x^-)\right) \leq u(x^+) + u(x^-)$$

or

$$2\left(1 + \alpha\frac{1}{\theta^2}\right) \leq 1 + \alpha\left(\frac{1}{\theta} + \epsilon\right)^2 + (1 - \theta\epsilon) + \alpha\left(\frac{1}{\theta} - \epsilon\right)^2.$$

which reduces to $2\alpha\epsilon \geq \theta$. This cannot hold for small $\epsilon$ and is a contradiction.

### F.4. PiL

Let $a > 1$ and $\theta > 0$, we define

$$r(x) = \sum_{i=1}^{d} \min\left\{1, \max\left\{0, \frac{\theta|x_i| - 1}{a - 1}\right\}\right\}$$

Since $r$ is separable, we only need to show that $r$ is not weakly convex in 1D.

Now in 1D, suppose there exists $\alpha > 0$ such that $u(x) := r(x) + \alpha x^2$ is convex. Let $x^+ = \frac{a}{\theta} + \epsilon$ and $x^- = \frac{a}{\theta} - \epsilon$ for $\epsilon \in (0, \frac{a-1}{\theta})$. We compute

$$u(x^+) = \alpha\left(\frac{a}{\theta} + \epsilon\right)^2 + 1$$

$$u(x^-) = \alpha\left(\frac{a}{\theta} - \epsilon\right)^2 + \frac{a - \theta\epsilon - 1}{a - 1}$$

$$u\left(\frac{x^+ + x^-}{2}\right) = \alpha\frac{a^2}{\theta^2} + 1.$$

The midpoint inequality $2u(\frac{x^+ + x^-}{2}) \leq u(x^+) + u(x^-)$ implies

$$2\alpha\frac{a^2}{\theta^2} + 2 \leq \alpha\left(\frac{a}{\theta} + \epsilon\right)^2 + 1 + \alpha\left(\frac{a}{\theta} - \epsilon\right)^2 + \frac{a - \theta\epsilon - 1}{a - 1}$$

which reduces to $\frac{\theta}{a-1} \leq 2\alpha\epsilon$. This cannot hold for small $\epsilon$ and is a contradiction.

**F.5.** $\ell_1 - \ell_2^p, p \in (1, 2)$

Restrict $r$ in $x = te_1$ for $t > 0$: $r(te_1) = \|te_1\|_1 - \|te_1\|_2^p = t - t^p$. Consider $\phi(t) = t - t^p + \alpha t^2$, then

$$\phi''(t) = -p(p-1)t^{p-2} + 2\alpha.$$

We see that for any fixed $\alpha > 0$, $\phi''(t) \to -\infty$ as $t \to 0^+$, so $\phi$ cannot be convex.

### F.6. DICNNs with leaky ReLU activations

Let $r$ be given by $r(x) = r_1(x) - r_2(x)$ where $r_1$ and $r_2$ are two ICNNs with leaky ReLU activations. $r_1$ and $r_2$ are then two convex, piecewise linear functions with respect to $x$. In general, $r$ is not weakly convex. To see this, take a 1D slice, $-r_2$ restricted to this slice is a 1D piecewise concave function that is not a linear function in general. If $r_1$ restricted to the same slice cannot exactly cancel out the concavity of $-r_2$, no amount of quadratic $\alpha\|x\|^2$ can.

## G. Example on discontinuous proximal operator of a DC function

Consider the following 1D example, $r(x) = -|x|$. Let $\gamma > 0$ and let $|v| < \gamma$, we consider

$$\text{Prox}_{\gamma r}(v) = \text{argmin}_x \left\{ \frac{1}{2\gamma}(x - v)^2 - |x| \right\}$$

Let $u(x) = \frac{1}{2}(x - v)^2 - \gamma|x|$. For $x > 0$, $u$ is minimized at $x^+ = v + \gamma > 0$; For $x < 0$, $u$ is minimized at $x^- = v - \gamma$. Comparing $f(x^+)$, $f(x^-)$ and $f(0)$ we conclude: if $v = 0$, both $x^+, x^-$ minimize $u$, if $v > 0$, $x^+$ is the unique minimizer, and if $v < 0$, $x^-$ is the unique minimizer. Therefore,

$$\text{Prox}_{\gamma r}(v) = \begin{cases} v + \gamma & \text{if } v > 0 \\ \{\gamma, -\gamma\} & \text{if } v = 0, \\ v - \gamma & \text{if } v < 0. \end{cases}$$

Hence, $\text{Prox}_{\gamma r}$ is multivalued and discontinuous at 0, as reflected in Figure 5.

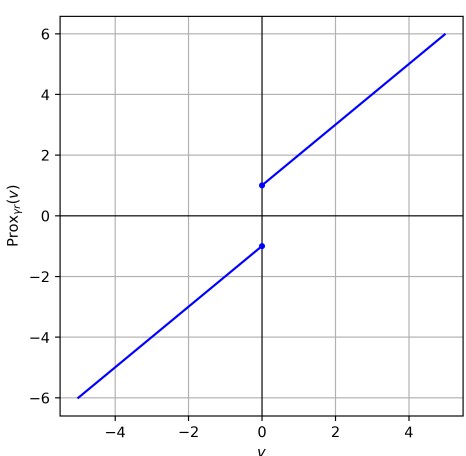

*Figure 5.* Plot of $\text{Prox}_{-|\cdot|}$

## H. Lemmas

**Lemma H.1.** *Let $g$ be a convex function and $\lambda, \gamma > 0$, it holds*

$$\text{Prox}_{\gamma g^\lambda}(x) = \frac{1}{\gamma + \lambda} \left( \gamma \text{Prox}_{(\gamma + \lambda)g}(x) + \lambda x \right).$$

*Proof.* This is a standard and known result. For completeness, we give a proof as follows. $\text{Prox}_{\gamma g^\lambda}(x)$ is the solution of

$$\min_{y} \left\{ g^\lambda(y) + \frac{1}{2\gamma} \|x - y\|^2 \right\} \tag{19}$$

By using the definition of Moreau envelope, the problem (19) can be cast to

$$\min_{(y,z)} \left\{ g(z) + \frac{1}{2\lambda} \|y - z\|^2 + \frac{1}{2\gamma} \|x - y\|^2 \right\}. \tag{20}$$

Now this problem is jointly convex w.r.t. $(y, z)$, by first-order optimality condition, we need to solve

$$\begin{cases} 0 \in \partial g(z) + \dfrac{1}{\lambda}(z - y) \\ \dfrac{1}{\lambda}(y - z) + \dfrac{1}{\gamma}(y - x) = 0. \end{cases}$$

Substituting $y = \frac{\gamma}{\gamma+\lambda}z + \frac{\lambda}{\gamma+\lambda}x$ from the second equation to the first equation, $z = \text{Prox}_{(\gamma+\lambda)g}(x)$. $\qquad\square$

**Lemma H.2.** *Let $q \in \mathbb{N}^*$ and $q \geq 2$. There exists $C_q \in \mathbb{R}$ such that for all $x, z \in \mathbb{R}^d$*

$$\|x + z\|^q \leq \|x\|^q + q\|x\|^{q-2}\langle x, z \rangle + C_q(\|x\|^{q-2}\|z\|^2 + \|z\|^q).$$

*Proof.* Let $\phi(u) = \|u\|^q$ and $g(t) = \phi(x + tz) = \|x + tz\|^q$. By Taylor expansion,

$$g(1) = g(0) + g'(0) + \int_0^1 g''(t)(1 - t)dt.$$

Now that

$$\nabla\phi(u) = q\|u\|^{q-2}u$$
$$\nabla^2\phi(u) = q\|u\|^{q-2}I + q(q-2)\|u\|^{q-4}uu^\top, \quad u \neq 0,$$

and $\nabla^2\phi(0) = 2I$ if $q = 2$ and $\nabla^2\phi(0) = 0$ if $q > 2$. It follows

$$g'(0) = \langle \nabla\phi(x), z \rangle = q\|x\|^{q-2}\langle x, z \rangle,$$
$$\begin{aligned} g''(t) &= z^\top \nabla^2\phi(x + tz)z \\ &= z^\top \left( q\|x + tz\|^{q-2}I + q(q-2)\|x + tz\|^{q-4}(x + tz)(x + tz)^\top \right) z \\ &= q\|x + tz\|^{q-2}\|z\|^2 + q(q-2)\|x + tz\|^{q-4}\langle z, x + tz \rangle^2. \end{aligned}$$

We evaluate

$$\begin{aligned} g''(t) &\leq q\|x + tz\|^{q-2}\|z\|^2 + q(q-2)\|x + tz\|^{q-2}\|z\|^2 \\ &= q(q-1)\|x + tz\|^{q-2}\|z\|^2. \end{aligned}$$

We have $\|x + tz\| \leq \|x\| + \|z\|$ for $t \in [0, 1]$ and

$$(\|x\| + \|z\|)^{q-2} \leq 2^{q-3}(\|x\|^{q-2} + \|z\|^{q-2})$$

where the inequality is an equality for $q \in \{2, 3\}$ and follows from Hölder inequality for $q \geq 4$. Therefore,

$$g''(t) \leq q(q-1)2^{q-3}(\|x\|^{q-2}\|z\|^2 + \|z\|^q),$$

it follows

$$\int_0^1 (1 - s)g''(s)ds \leq q(q-1)2^{q-4}(\|x\|^{q-2}\|z\|^2 + \|z\|^q).$$

Putting together,

$$\|x + z\|^q \leq \|x\|^q + q\|x\|^{q-2}\langle x, z \rangle + q(q-1)2^{q-4}(\|x\|^{q-2}\|z\|^2 + \|z\|^q).$$

$\qquad\square$

**Lemma H.3.** *Let $\{Y_k\}_k$ be given by*

$$Y_{k+1} = Y_k - \gamma b(Y_k) + \sqrt{2\gamma} Z_{k+1}$$

*where $\{Z_k\}_k$ follows i.i.d. normal distribution and $b$ is $L$-Lipschitz and $(m, R)$-distant dissipative. Suppose that $\gamma \leq \frac{m}{L^2}$. Let $Q$ be the transition Markov kernel of $\{Y_k\}_k$, then $Q$ has an invariant distribution $\pi_Y \in \mathcal{P}_1(\mathbb{R}^d)$. Furthermore, for $p' \geq 2, p' \in \mathbb{N}^*$, if $\gamma$ is sufficiently small, i.e.,*

$$\gamma \leq \min\left\{\frac{1}{4L}, \frac{m}{2^{p'+2}L^2(p'-1)}\right\} \tag{21}$$

*it holds $\pi_Y \in \mathcal{P}_{p'}(\mathbb{R}^d)$.*

*Proof.* Since $b$ is Lipschitz, similar to Subsection B, once the chain starts in $\mathcal{P}_{p'}(\mathbb{R}^d)$, it stays in $\mathcal{P}_{p'}(\mathbb{R}^d)$. Theorem 5 in (Renaud et al., 2025a) guarantees that: there exist $D > 0, \rho \in (0, 1)$:

$$W_1(\mu Q^k, \nu Q^k) \leq D\rho^{k\gamma} W_1(\mu, \nu) \tag{22}$$

for all $\mu, \nu \in \mathcal{P}_1(\mathbb{R})$. Therefore, for large enough $k$, $Q^k$ is contractive in $(\mathcal{P}_1(\mathbb{R}^d), W_1)$. Applying Picard fixed point theorem, for some fixed large $k$, there exists unique $\pi_Y \in \mathcal{P}_1(\mathbb{R}^d)$ such that $\pi_Y Q^k = \pi_Y$. It follows that $(\pi_Y Q)Q^k = \pi_Y Q$ and by uniqueness $\pi_Y Q = \pi_Y$. If $p' = 1$, the proof is complete. For $p' \geq 2$, we need to further show $\pi_Y \in \mathcal{P}_{p'}(\mathbb{R}^d)$ when $\gamma$ satisfies (21). Since $b$ is $L$-Lipschitz,

$$\|b(x)\| \leq L\|x\| + \|b(0)\| := L\|x\| + a, \quad \forall x.$$

$b$ being $(m, R)$-distant dissipative implies

$$\langle b(x), x \rangle \geq \frac{m}{2}\|x\|^2 - \frac{1}{2m}\|b(0)\|^2 \quad \forall x : \|x\| \geq R.$$

So there exists $c > 0$:

$$\langle b(x), x \rangle \geq \frac{m}{2}\|x\|^2 - c, \quad \forall x \in \mathbb{R}^d. \tag{23}$$

We have

$$\mathbb{E}(\|Y_{k+1}\|^{p'}|Y_k = y) = \mathbb{E}\|y - \gamma b(y) + \sqrt{2\gamma}Z\|^{p'},$$

where $Z \sim \mathcal{N}(0, I)$. Applying Lemma H.2, there exists $C_{p'}$ such that

$$\|x + z\|^{p'} \leq \|x\|^{p'} + p'\|x\|^{p'-2}\langle x, z \rangle + C_{p'}(\|x\|^{p'-2}\|z\|^2 + \|z\|^{p'}).$$

Therefore,

$$\begin{aligned}
\|y - \gamma b(y) + \sqrt{2\gamma}Z\|^{p'} \leq &\|y\|^{p'} + p'\|y\|^{p'-2}\langle y, -\gamma b(y) + \sqrt{2\gamma}Z \rangle \\
&+ C_{p'}\left(\|y\|^{p'-2}\| - \gamma b(y) + \sqrt{2\gamma}Z\|^2 + \| - \gamma b(y) + \sqrt{2\gamma}Z\|^{p'}\right)
\end{aligned}$$

so

$$\begin{aligned}
\mathbb{E}\|y - \gamma b(y) + \sqrt{2\gamma}Z\|^{p'} \leq &\|y\|^{p'} - \gamma p'\|y\|^{p'-2}\langle y, b(y) \rangle \\
&+ C_{p'}\|y\|^{p'-2}\mathbb{E}\| - \gamma b(y) + \sqrt{2\gamma}Z\|^2 + C_{p'}\mathbb{E}\| - \gamma b(y) + \sqrt{2\gamma}Z\|^{p'}.
\end{aligned}$$

Now use (23),

$$\begin{aligned}
&\mathbb{E}\|y - \gamma b(y) + \sqrt{2\gamma}Z\|^{p'} \\
&\leq \|y\|^{p'} + \gamma p'\|y\|^{p'-2}\left(-\frac{m}{2}\|y\|^2 + c\right) + 2C_{p'}\|y\|^{p'-2}(\gamma^2\|b(y)\|^2 + 2\gamma d) \\
&\quad + C_{p'}\mathbb{E}\left(\gamma\|b(y)\| + \sqrt{2\gamma}\|Z\|\right)^{p'}.
\end{aligned}$$

Applying Hölder inequality

$$\left(\gamma\|b(y)\| + \sqrt{2\gamma}\|Z\|\right)^{p'} \leq 2^{p'-1}\left(\gamma^{p'}\|b(y)\|^{p'} + (2\gamma)^{\frac{p'}{2}}\|Z\|^{p'}\right),$$

we get

$$\mathbb{E}\|y - \gamma b(y) + \sqrt{2\gamma}Z\|^{p'}$$
$$\leq \|y\|^{p'} + \gamma p'\|y\|^{p'-2}\left(-\frac{m}{2}\|y\|^2 + c\right) + 2C_{p'}\|y\|^{p'-2}(\gamma^2\|b(y)\|^2 + 2\gamma d)$$
$$+ C_{p'}\gamma^{p'}2^{p'-1}\|b(y)\|^{p'} + 2^{\frac{3p'}{2}-1}C_{p'}\gamma^{\frac{p'}{2}}\mathbb{E}\|Z\|^{p'}$$
$$= \|y\|^{p'} - \frac{m\gamma p'}{2}\|y\|^{p'} + 2C_{p'}\gamma^2\|y\|^{p'-2}\|b(y)\|^2 + 4C_{p'}\gamma d\|y\|^{p'-2} + c\gamma p'\|y\|^{p'-2}$$
$$+ C_{p'}\gamma^{p'}2^{p'-1}\|b(y)\|^{p'} + 2^{\frac{3p'}{2}-1}C_{p'}\gamma^{\frac{p'}{2}}\mathbb{E}\|Z\|^{p'}.$$

Now use $\|b(y)\| \leq L\|y\| + a$, it holds

$$\|b(y)\|^2 \leq 2(L^2\|y\|^2 + a^2)$$
$$\|b(y)\|^{p'} \leq 2^{p'-1}(L^{p'}\|y\|^{p'} + a^{p'}).$$

Then

$$\mathbb{E}\|y - \gamma b(y) + \sqrt{2\gamma}Z\|^{p'} \leq \left(1 + 4C_{p'}\gamma^2 L^2 + C_{p'}\gamma^{p'}2^{2p'-2}L^{p'} - \frac{m\gamma p'}{2}\right)\|y\|^{p'}$$
$$+ (4C_{p'}\gamma d + 4C_{p'}\gamma^2 a^2 + c\gamma p')\|y\|^{p'-2} + C_{p'}\gamma^{p'}2^{2p'-2}a^{p'} + 2^{\frac{3p'}{2}-1}C_{p'}\gamma^{\frac{p'}{2}}\mathbb{E}\|Z\|^{p'}. \tag{24}$$

Taking expectation over $y \sim p_{Y_k}$, we get

$$\mathbb{E}\|Y_{k+1}\|^{p'} \leq \left(1 + 4C_{p'}\gamma^2 L^2 + C_{p'}\gamma^{p'}2^{2p'-2}L^{p'} - \frac{m\gamma p'}{2}\right)\mathbb{E}\|Y_k\|^{p'}$$
$$+ (4C_{p'}\gamma d + 4C_{p'}\gamma^2 a^2 + c\gamma p')\mathbb{E}\|Y_k\|^{p'-2} + C_{p'}\gamma^{p'}2^{2p'-2}a^{p'} + 2^{\frac{3p'}{2}-1}C_{p'}\gamma^{\frac{p'}{2}}\mathbb{E}\|Z\|^{p'}. \tag{25}$$

Let $\bar{R} > 0$ so that

$$\bar{R}^2 \geq 4\frac{4C_{p'}\gamma d + 4C_{p'}\gamma^2 a^2 + c\gamma p'}{m\gamma p'}. \tag{26}$$

We have

$$\|y\|^{p'-2} \leq \frac{\|y\|^{p'}}{\bar{R}^2} \quad \text{for } \|y\| \geq \bar{R}.$$

So

$$\mathbb{E}\|Y_k\|^{p'-2} = \int_{\|y\|\geq\bar{R}} \|y\|^{p'-2}p_{Y_k}(y)dy + \int_{\|y\|\leq\bar{R}} \|y\|^{p'-2}p_{Y_k}(y)dy$$
$$\leq \frac{1}{\bar{R}^2}\int_{\|y\|\geq\bar{R}} \|y\|^{p'}p_{Y_k}(y)dy + \bar{R}^{p'-2}$$
$$\leq \frac{1}{\bar{R}^2}\mathbb{E}\|Y_k\|^{p'} + \bar{R}^{p'-2}.$$

Apply this inequality to (25),

$$
\begin{aligned}
\mathbb{E}\|Y_{k+1}\|^{p'} &\leq \left(1 + 4C_{p'}\gamma^2 L^2 + C_{p'}\gamma^{p'} 2^{2p'-2} L^{p'} - \frac{m\gamma p'}{2} + \frac{4C_{p'}\gamma d + 4C_{p'}\gamma^2 a^2 + c\gamma p'}{\bar{R}^2}\right) \mathbb{E}\|Y_k\|^{p'} \\
&\quad + (4C_{p'}\gamma d + 4C_{p'}\gamma^2 a^2 + c\gamma p')\bar{R}^{p'-2} + C_{p'}\gamma^{p'} 2^{2p'-2} a^{p'} + 2^{\frac{3p'}{2}-1} C_{p'}\gamma^{\frac{p'}{2}} \mathbb{E}\|Z\|^{p'} \\
&\leq \left(1 + 4C_{p'}\gamma^2 L^2 + C_{p'}\gamma^{p'} 2^{2p'-2} L^{p'} - \frac{m\gamma p'}{4}\right) \mathbb{E}\|Y_k\|^{p'} \\
&\quad + (4C_{p'}\gamma d + 4C_{p'}\gamma^2 a^2 + c\gamma p')\bar{R}^{p'-2} + C_{p'}\gamma^{p'} 2^{2p'-2} a^{p'} + 2^{\frac{3p'}{2}-1} C_{p'}\gamma^{\frac{p'}{2}} \mathbb{E}\|Z\|^{p'}
\end{aligned}
\tag{27}
$$

thanks to (26). Now for $\gamma$ small enough as in (21), it holds

$$
1 + 4C_{p'}\gamma^2 L^2 + C_{p'}\gamma^{p'} 2^{2p'-2} L^{p'} - \frac{m\gamma p'}{4} < 1.
\tag{28}
$$

By choosing $Y_1 \in \mathcal{P}_{p'}(\mathbb{R}^d)$, the recursion (27) and (28) imply

$$
\sup_k \mathbb{E}\|Y_k\|^{p'} < +\infty.
$$

From (22), setting $\nu = \pi_Y$ we get $p_{Y_k}$ converging to $\pi_Y$ in 1-Wasserstein, which further implies *narrow convergence*, i.e., for any $\varphi$ bounded and continuous, it holds

$$
\int \varphi(y) p_{Y_k}(y) dy \to \int \varphi(y) \pi_Y(y) dy.
$$

Let $\varphi_M(y) := \min\{\|y\|^{p'}, M\}$,

$$
\int \varphi_M(y) \pi_Y(y) dy = \lim_{k\to\infty} \int \varphi_M(y) p_{Y_k}(y) dy \leq \sup_k \int \|y\|^{p'} p_{Y_k}(y) dy = \sup_k \mathbb{E}\|Y_k\|^{p'} < +\infty.
$$

By letting $M \to \infty$ and applying the monotone convergence theorem, we obtain

$$
\int \|y\|^{p'} \pi_Y(y) dy < +\infty.
$$

$\square$

**Lemma H.4.** *Let $b : \mathbb{R}^d \to \mathbb{R}^d$ be a drift. Assume that $b$ is continuous and $(\mu, R)$-distant dissipative. Then there exists $x^* \in \mathbb{R}^d$ such that $b(x^*) = 0$.*

*Proof.* If $b(0) = 0$, it is done. Now consider $b(0) \neq 0$, we have

$$
\langle b(x) - b(y), x - y \rangle \geq \mu\|x - y\|^2, \quad \forall x, y : \|x - y\| \geq R.
$$

Set $y = 0$, for $\|x\| \geq R$,

$$
\begin{aligned}
\langle b(x), x \rangle &\geq \mu\|x\|^2 + \langle b(0), x \rangle \\
&\geq \frac{\mu}{2}\|x\|^2 - \frac{1}{2\mu}\|b(0)\|^2 \\
&\geq \frac{1}{2\mu}\|b(0)\|^2 \quad \text{if } \|x\| \geq (\sqrt{2}/\mu)\|b(0)\|.
\end{aligned}
$$

So for $R_1 := \max\{R, (\sqrt{2}/\mu)\|b(0)\|\}$, it holds $\langle b(x), x \rangle > 0$ for all $x : \|x\| \geq R_1$. According to Teschl (2005, Thm. 2.13), $b$ vanishes somewhere inside $B(0, R_1)$. $\square$

# I. Proofs

## I.1. Proof of Lemma 4.1

1) Suppose that for all $\|x - y\| \geq R$,

$$\langle \nabla f(x) + \partial r_1(x) - \partial r_2(x) - \nabla f(y) - \partial r_1(y) + \partial r_2(y), x - y \rangle \geq \mu \|x - y\|^2.$$

Here, by abuse of notation we still use the set notation $\partial$ in the above inequality.

It follows that

$$
\begin{aligned}
\langle \nabla f(x) - \nabla f(y), x - y \rangle &\geq \mu \|x - y\|^2 - \langle \partial r_1(x) - \partial r_1(y), x - y \rangle + \langle \partial r_2(x) - \partial r_2(y), x - y \rangle \\
&\geq \mu \|x - y\|^2 - \langle \partial r_1(x) - \partial r_1(y), x - y \rangle \\
&\geq \mu \|x - y\|^2 - 2G_1 \|x - y\|.
\end{aligned}
$$

Therefore, $f$ is $(\frac{\mu}{2}, \bar{R})$ distant dissipative where $\bar{R} = \max\{R, 4G_1/\mu\}$.

2) Suppose that for all $\|x - y\| \geq R$,

$$\langle \nabla f(x) - \nabla f(y), x - y \rangle \geq \mu \|x - y\|^2.$$

Then

$$
\begin{aligned}
&\langle \nabla f(x) + \partial r_1(x) - \partial r_2(x) - \nabla f(y) - \partial r_1(y) + \partial r_2(y), x - y \rangle \\
&\geq \mu \|x - y\|^2 - \langle \partial r_2(x) - \partial r_2(y), x - y \rangle \\
&\geq
\begin{cases}
\mu \|x - y\|^2 - 2G_2 \|x - y\| & \text{if Assumption 4(i)} \\
\mu \|x - y\|^2 - M \|x - y\|^{\kappa + 1} & \text{if Assumption 4(ii).}
\end{cases}
\end{aligned}
$$

Therefore, $V$ is $(\frac{\mu}{2}, \bar{R})$-distant dissipative where $\bar{R} = \max\{R, 4G_2/\mu\}$ if Assumption 4(i), $\max\{R, (2M/\mu)^{\frac{1}{1-\kappa}}\}$ if Assumption 4(ii).

## I.2. Proof of Lemma 4.2

We write

$$Y_{k+1} = \text{Prox}_{\gamma r_1^\lambda}(Y_k) - \gamma \nabla f(\text{Prox}_{\gamma r_1^\lambda}(Y_k)) + \gamma \nabla r_2^\lambda(\text{Prox}_{\gamma r_1^\lambda}(Y_k)) + \sqrt{2\gamma} Z_{k+1}. \tag{29}$$

Now using $\text{Prox}_{\gamma r_1^\lambda}(Y_k) = Y_k - \gamma \nabla (r_1^\lambda)^\gamma(Y_k)$, we get

$$Y_{k+1} = Y_k - \gamma \nabla (r_1^\lambda)^\gamma(Y_k) - \gamma \nabla f(\text{Prox}_{\gamma r_1^\lambda}(Y_k)) + \gamma \nabla r_2^\lambda(\text{Prox}_{\gamma r_1^\lambda}(Y_k)) + \sqrt{2\gamma} Z_{k+1}. \tag{30}$$

Since $\nabla (r_1^\lambda)^\gamma(Y_k) = \nabla r_1^\lambda(\text{Prox}_{\gamma r_1^\lambda}(Y_k))$,

$$Y_{k+1} = Y_k - \gamma \nabla r_1^\lambda(\text{Prox}_{\gamma r_1^\lambda}(Y_k)) - \gamma \nabla f(\text{Prox}_{\gamma r_1^\lambda}(Y_k)) + \gamma \nabla r_2^\lambda(\text{Prox}_{\gamma r_1^\lambda}(Y_k)) + \sqrt{2\gamma} Z_{k+1}.$$

Now since $r_i^\lambda$ is $\frac{1}{\lambda}$-smooth for $i \in \{1, 2\}$, it holds

$$\|\nabla r_i^\lambda(\text{Prox}_{\gamma r_1^\lambda}(x)) - \nabla r_i^\lambda(\text{Prox}_{\gamma r_1^\lambda}(y))\| \leq \frac{1}{\lambda} \|\text{Prox}_{\gamma r_1^\lambda}(x) - \text{Prox}_{\gamma r_1^\lambda}(y)\| \leq \frac{1}{\lambda} \|x - y\|$$

where the last inequality is thanks to the nonexpansiveness of the proximal operator. Therefore, for all $\gamma > 0$, $\nabla r_i^\lambda(\text{Prox}_{\gamma r_1^\lambda}(x))$ is $\frac{1}{\lambda}$-Lipschitz. As $f$ is $L_f$-smooth, the drift $b_\lambda^\gamma$ is $(\frac{2}{\lambda} + L_f)$-Lipschitz.

## I.3. Proof of Lemma 4.3

$$
\begin{aligned}
&\langle b_\lambda^\gamma(x) - b_\lambda^\gamma(y), x - y \rangle \\
&= \langle \nabla r_1^\lambda(\mathrm{Prox}_{\gamma r_1^\lambda}(x)) + \nabla f(\mathrm{Prox}_{\gamma r_1^\lambda}(x)) - \nabla r_2^\lambda(\mathrm{Prox}_{\gamma r_1^\lambda}(x)) \\
&\quad - \nabla r_1^\lambda(\mathrm{Prox}_{\gamma r_1^\lambda}(y)) - \nabla f(\mathrm{Prox}_{\gamma r_1^\lambda}(y)) + \nabla r_2^\lambda(\mathrm{Prox}_{\gamma r_1^\lambda}(y)), x - y \rangle \\
&= \underbrace{\langle \nabla r_1^\lambda(\mathrm{Prox}_{\gamma r_1^\lambda}(x)) - \partial r_1(x) - \nabla r_1^\lambda(\mathrm{Prox}_{\gamma r_1^\lambda}(y)) + \partial r_1(y), x - y \rangle}_{:=I} + \\
&\quad + \underbrace{\langle \nabla f(\mathrm{Prox}_{\gamma r_1^\lambda}(x)) - \nabla f(x) - \nabla f(\mathrm{Prox}_{\gamma r_1^\lambda}(y)) + \nabla f(y), x - y \rangle}_{:=II} \\
&\quad + \underbrace{\langle -\nabla r_2^\lambda(\mathrm{Prox}_{\gamma r_1^\lambda}(x)) + \partial r_2(x) + \nabla r_2^\lambda(\mathrm{Prox}_{\gamma r_1^\lambda}(y)) - \partial r_2(y), x - y \rangle}_{:=III} \\
&\quad + \underbrace{\langle \partial r_1(x) - \partial r_1(y) + \nabla f(x) - \nabla f(y) - \partial r_2(x) + \partial r_2(y), x - y \rangle}_{:=IV}
\end{aligned}
$$

The first term:

$$
I \geq - \left( \|\nabla r_1^\lambda(\mathrm{Prox}_{\gamma r_1^\lambda}(x))\| + \|\partial r_1(x)\| + \|\nabla r_1^\lambda(\mathrm{Prox}_{\gamma r_1^\lambda}(y))\| + \|\partial r_1(y)\| \right) \|x - y\|.
$$

Since $\nabla r_1^\lambda(\mathrm{Prox}_{\gamma r_1^\lambda}(x)) \in \partial r_1 \left( \mathrm{Prox}_{\lambda r_1}(\mathrm{Prox}_{\gamma r_1^\lambda}(x)) \right)$, under Assumption 3, it holds

$$
I \geq -4G_1 \|x - y\|.
$$

The second term,

$$
\begin{aligned}
II &\geq -\|\nabla f(\mathrm{Prox}_{\gamma r_1^\lambda}(x)) - \nabla f(x) - \nabla f(\mathrm{Prox}_{\gamma r_1^\lambda}(y)) + \nabla f(y)\| \|x - y\| \\
&\geq -L_f \left( \|\mathrm{Prox}_{\gamma r_1^\lambda}(x) - x\| + \|\mathrm{Prox}_{\gamma r_1^\lambda}(y) - y\| \right) \|x - y\|.
\end{aligned}
$$

Now that

$$
\begin{aligned}
x - \mathrm{Prox}_{\gamma r_1^\lambda}(x) &= \gamma \nabla (r_1^\lambda)^\gamma(x) \\
&= \gamma \nabla r_1^\lambda(\mathrm{Prox}_{\gamma r_1^\lambda}(x)) \\
&\in \gamma \partial r_1 \left( \mathrm{Prox}_{\lambda r_1}(\mathrm{Prox}_{\gamma r_1^\lambda}(x)) \right).
\end{aligned}
\tag{31}
$$

Therefore,

$$
II \geq -2L_f \gamma G_1 \|x - y\|.
$$

The third term:

$$
III \geq \langle -\nabla r_2^\lambda(\mathrm{Prox}_{\gamma r_1^\lambda}(x)) + \nabla r_2^\lambda(\mathrm{Prox}_{\gamma r_1^\lambda}(y)), x - y \rangle.
$$

Under Assumption 4, if case (i) happens, we have

$$
III \geq -2G_2 \|x - y\|,
$$

and if case (ii) happens

$$
\begin{aligned}
III &\geq -\|\nabla r_2(\mathrm{Prox}_{\lambda r_2}(\mathrm{Prox}_{\gamma r_1^\lambda}(x))) - \nabla r_2(\mathrm{Prox}_{\lambda r_2}(\mathrm{Prox}_{\gamma r_1^\lambda}(y)))\| \|x - y\| \\
&\geq -M \|\mathrm{Prox}_{\lambda r_2}(\mathrm{Prox}_{\gamma r_1^\lambda}(x)) - \mathrm{Prox}_{\lambda r_2}(\mathrm{Prox}_{\gamma r_1^\lambda}(y))\|^\kappa \|x - y\| \\
&\geq -M \|\mathrm{Prox}_{\gamma r_1^\lambda}(x) - \mathrm{Prox}_{\gamma r_1^\lambda}(y)\|^\kappa \|x - y\| \\
&\geq -M \|x - y\|^{\kappa+1}.
\end{aligned}
$$

The last term: if $\|x - y\| \geq R_0$

$$IV \geq \mu \|x - y\|^2$$

thanks to the Assumption 2. From these evaluations, we get

$$
\begin{aligned}
&\langle b_\lambda^\gamma(x) - b_\lambda^\gamma(y), x - y \rangle \\
&\geq \begin{cases} \mu \|x - y\|^2 - (4G_1 + 2L_f \gamma G_1 + 2G_2)\|x - y\| & \text{if Assumption 4(i)} \\ \mu \|x - y\|^2 - (4G_1 + 2L_f \gamma G_1)\|x - y\| - M\|x - y\|^{\kappa + 1} & \text{if Assumption 4(ii).} \end{cases}
\end{aligned}
$$

For $\|x - y\|$ being large, the second-order term dominates the lower-order terms (note that $\kappa < 1$), we get:

$$\langle b_\lambda^\gamma(x) - b_\lambda^\gamma(y), x - y \rangle \geq \frac{\mu}{2}\|x - y\|^2 \tag{32}$$

whenever

$$
\|x - y\| \geq \begin{cases} \max\left\{ R_0, \dfrac{8G_1 + 4L_f \gamma_0 G_1 + 4G_2}{\mu} \right\} & \text{if Assumption 4(i)} \\[3ex] \max\left\{ R_0, \dfrac{16G_1 + 8L_f \gamma_0 G_1}{\mu}, \left(\dfrac{4M}{\mu}\right)^{\frac{1}{1-\kappa}} \right\} & \text{if Assumption 4(ii).} \end{cases} \tag{33}
$$

## I.4. Proof of the distant dissipativity of $\bar{b}_\lambda^\gamma$

For $\|x - y\| \geq R_0$,

$$
\begin{aligned}
&\langle \bar{b}_\lambda^\gamma(x) - \bar{b}_\lambda^\gamma(y), x - y \rangle \\
&= \langle \nabla r_1^\lambda(\mathrm{Prox}_{\gamma r_1^\lambda}(x)) + \nabla f(x) - \nabla r_2^\lambda(x) - \nabla r_1^\lambda(\mathrm{Prox}_{\gamma r_1^\lambda}(y)) - \nabla f(y) + \nabla r_2^\lambda(y), x - y \rangle \\
&= \langle \nabla r_1^\lambda(\mathrm{Prox}_{\gamma r_1^\lambda}(x)) - \partial r_1(x) - \nabla r_1^\lambda(\mathrm{Prox}_{\gamma r_1^\lambda}(y)) + \partial r_1(y), x - y \rangle \\
&\quad + \langle -\nabla r_2^\lambda(x) + \partial r_2(x) + \nabla r_2^\lambda(y) - \partial r_2(y), x - y \rangle \\
&\quad + \langle \partial r_1(x) + \nabla f(x) - \partial r_2(x) - \partial r_1(y) - \nabla f(y) + \partial r_2(y), x - y \rangle \\
&\geq -4G_1 \|x - y\| + \langle -\nabla r_2^\lambda(x) + \nabla r_2^\lambda(y), x - y \rangle + \mu \|x - y\|^2.
\end{aligned}
$$

Therefore,

$$
\langle \bar{b}_\lambda^\gamma(x) - \bar{b}_\lambda^\gamma(y), x - y \rangle \geq \begin{cases} \mu \|x - y\|^2 - (4G_1 + 2G_2)\|x - y\| & \text{if Assumption 4(i)} \\ \mu \|x - y\|^2 - 4G_1 \|x - y\| - M\|x - y\|^{\kappa + 1} & \text{if Assumption 4(ii).} \end{cases}
$$

where the last inequality follows the same arguments as in the proof of Lemma 4.3 (Appendix I.3).

Therefore,

$$\langle \bar{b}_\lambda^\gamma(x) - \bar{b}_\lambda^\gamma(y), x - y \rangle \geq \frac{\mu}{2}\|x - y\|^2$$

whenever

$$
\|x - y\| \geq \begin{cases} \max\left\{ R_0, \dfrac{8G_1 + 4G_2}{\mu} \right\} & \text{if Assumption 4(i)} \\[3ex] \max\left\{ R_0, \dfrac{16G_1}{\mu}, \left(\dfrac{4M}{\mu}\right)^{\frac{1}{1-\kappa}} \right\} & \text{if Assumption 4(ii).} \end{cases}
$$

## I.5. Proof of Theorem 4.4

Let $Q, \bar{Q}$ be transition kernels of $\{Y_k\}_k$ and $\{\bar{Y}_k\}_k$, respectively. Recall that

$$Y_1 = X_0 - \gamma \nabla f(X_0) + \gamma \nabla r_2^\lambda(X_0) + \sqrt{2\gamma} Z_1.$$

In Appendix B, we show that: if $X_0 \in \mathcal{P}_{p'}(\mathbb{R}^d)$ for some $p' \geq 1$, then $Y_1 \in \mathcal{P}_{p'}(\mathbb{R}^d)$. By assumption, $X_0 \in \mathcal{P}_1(\mathbb{R}^d)$ if $q = 1$ and $X_0 \in \mathcal{P}_{2q}(\mathbb{R}^d)$ if $q \geq 2$. So $Y_1 \in \mathcal{P}_1(\mathbb{R}^d)$ if $q = 1$ and $Y_1 \in \mathcal{P}_{2q}(\mathbb{R}^d)$ if $q \geq 2$. Furthermore, applying Lemma H.3 (for $p' = 2q$), there exist invariant distributions of $Q$ and $\bar{Q}$ named $p_{\lambda,\gamma}, \bar{p}_{\lambda,\gamma}$ in $\mathcal{P}_1(\mathbb{R}^d)$ and further in $\mathcal{P}_{2q}(\mathbb{R}^d)$ if $q \geq 2$.

We now consider two cases $q = 1$ and $q \geq 2$ separately since they require slightly different conditions.

Case 1: $q = 1$

Renaud et al. (2025a) showed in the proof of Theorem 5 that: there exist $D(\lambda) > 0$ and $\rho_\lambda \in (0, 1)$ so that

$$W_1(\nu_1 Q^k, \nu_2 Q^k) \leq D(\lambda)\rho_\lambda^{k\gamma} W_1(\nu_1, \nu_2) \tag{34}$$

where $\nu_1, \nu_2 \in \mathcal{P}_1(\mathbb{R}^d)$ are two initial distributions.

Plugging $\nu_1 := p_{\lambda,\gamma}$ the invariant distribution of $Q$ and $\nu_2 := p_{Y_1}$ the initial distribution of the chain, we get

$$W_1(p_{\lambda,\gamma}, p_{Y_{k+1}}) \leq D(\lambda)\rho_\lambda^{k\gamma} W_1(p_{\lambda,\gamma}, p_{Y_1}).$$

We will show that: $W_1(p_{\lambda,\gamma}, p_{Y_1})$ is uniformly bounded by a constant that does not depend on $\gamma$, i.e., there exist $C_1(\lambda) > 0, \rho_\lambda \in (0, 1)$,

$$W_1(p_{\lambda,\gamma}, p_{Y_{k+1}}) \leq C_1(\lambda)\rho_\lambda^{k\gamma}, \quad \forall k \in \mathbb{N}. \tag{35}$$

We defer the proof of (35) to a later part of the text.

Now (Renaud et al., 2025a, Theorem 1) states that: there exists $C_2(\lambda)$ such that

$$W_1(p_{\lambda,\gamma}, \bar{p}_{\lambda,\gamma}) \leq C_2(\lambda) \left( \mathbb{E}_{X \sim p_{\lambda,\gamma}} \| b_\lambda^\gamma(X) - \bar{b}_\lambda^\gamma(X) \|^2 \right)^{\frac{1}{2}}. \tag{36}$$

On the other hand,

$$\| b_\lambda^\gamma(y) - \bar{b}_\lambda^\gamma(y) \| \leq \left( L_f + \frac{1}{\lambda} \right) \| \mathrm{Prox}_{\gamma r_1^\lambda}(y) - y \| \leq \left( L_f + \frac{1}{\lambda} \right) G_1 \gamma \tag{37}$$

where the last inequality is thanks to (31). Therefore,

$$W_1(p_{\lambda,\gamma}, \bar{p}_{\lambda,\gamma}) \leq C_2(\lambda) \left( L_f + \frac{1}{\lambda} \right) G_1 \gamma. \tag{38}$$

On the other hand, to decouple the step size and the drift, consider the following auxiliary Markov chain:

$$\bar{Y}_{k+1} = \bar{Y}_k - \epsilon \bar{b}_\lambda^\gamma(\bar{Y}_k) + \sqrt{2\epsilon} \bar{Z}_{k+1}, \tag{39}$$

where the step size $\epsilon \leq \frac{\mu \lambda^2}{2(2 + \lambda L_f)^2}$. Applying (Renaud et al., 2025a, Theorem 2), this chain has an invariant distribution $\bar{p}_{\lambda,\gamma,\epsilon}$ and there exists $C_3(\lambda) > 0$ independent to both $\epsilon$ and $\gamma$ (note that $C_3(\lambda)$ depends on the drift $\bar{b}_\lambda^\gamma$ only via its smoothness parameter and distant dissipativity parameters and these parameters are independent to $\gamma$):

$$W_1(\bar{p}_{\lambda,\gamma,\epsilon}, \pi_{\lambda,\gamma}) \leq C_3(\lambda)\epsilon^{\frac{1}{2}} \tag{40}$$

where $\pi_{\lambda,\gamma}$ is defined in (14). By setting $\epsilon = \gamma$ and $\bar{p}_{\lambda,\gamma} := \bar{p}_{\lambda,\gamma,\gamma}$, we get

$$W_1(\bar{p}_{\lambda,\gamma}, \pi_{\lambda,\gamma}) \leq C_3(\lambda)\gamma^{\frac{1}{2}}. \tag{41}$$

From (35), (38), and (41),

$$W_1(p_{Y_{k+1}}, \pi_{\lambda,\gamma}) \leq C_1(\lambda)\rho_\lambda^{k\gamma} + C_2(\lambda)\left(L_f + \frac{1}{\lambda}\right)G_1\gamma + C_3(\lambda)\gamma^{\frac{1}{2}}. \tag{42}$$

Here we can denote $A_\lambda = C_1(\lambda), B_\lambda = C_2(\lambda)\left(L_f + \frac{1}{\lambda}\right)G_1\gamma_0^{\frac{1}{2}} + C_3(\lambda)$ for some $\gamma_0$ being an upper bound of $\gamma$.

The above result gives a convergence guarantee to $\pi_{\lambda,\gamma}$ for the law of the sequence $\{Y_k\}$. As promised, we prove (35) by uniformly bounding $W_1(p_{\lambda,\gamma}, p_{Y_1})$ as follows. We have

$$W_1(p_{\lambda,\gamma}, p_{Y_1}) \leq W_1(p_{\lambda,\gamma}, \bar{p}_{\lambda,\gamma}) + W_1(\bar{p}_{\lambda,\gamma}, \pi_{\lambda,\gamma}) + W_1(\pi_{\lambda,\gamma}, p_{Y_1})$$

$$\leq C_2(\lambda)\left(L_f + \frac{1}{\lambda}\right)G_1\gamma_0 + C_3(\lambda)\gamma_0^{\frac{1}{2}} + W_1(\pi_{\lambda,\gamma}, p_{Y_1})$$

where $\gamma_0$ denotes the upper bound of $\gamma$ (the step size condition in Theorem 4.4). On the other hand, from (17), it holds

$$\pi_{\lambda,\gamma}(x) \leq e^{\frac{G_1^2\gamma_0}{2}}\pi_\lambda(x). \tag{43}$$

So,

$$W_1(\pi_{\lambda,\gamma}, p_{Y_1}) \leq \int \|x\|\pi_{\lambda,\gamma}(x)dx + \int \|x\|p_{Y_1}(x)dx$$

$$\leq e^{\frac{G_1^2\gamma_0}{2}}\int \|x\|\pi_\lambda(x)dx + \int \|x\|p_{Y_1}(x)dx.$$

This is a uniform bound.

Now we turn to the original sequence $X_{k+1} = \text{Prox}_{\gamma r_1^\lambda}(Y_{k+1})$. The distribution of $X_{k+1}$ is $p_{X_{k+1}} = (\text{Prox}_{\gamma r_1^\lambda})_\# p_{Y_{k+1}}$. Let $\nu_{\lambda,\gamma} = (\text{Prox}_{\gamma r_1^\lambda})_\# \pi_{\lambda,\gamma}$ and $(X, Y)$ be the optimal coupling of $(\pi_{\lambda,\gamma}, p_{Y_{k+1}})$,

$$W_1(p_{X_{k+1}}, \nu_{\lambda,\gamma}) \leq \mathbb{E}\|\text{Prox}_{\gamma r_1^\lambda}(X) - \text{Prox}_{\gamma r_1^\lambda}(Y)\|$$

$$\leq \mathbb{E}\|X - Y\|$$

$$= W_1(\pi_{\lambda,\gamma}, p_{Y_{k+1}}).$$

Case 2: $q \geq 2$

According to the proof of Theorem 5 in (Renaud et al., 2025a) and Lemma 18 therein: there exist $D_q(\lambda) > 0, \rho_\lambda \in (0, 1)$:

$$W_q(\mu Q^k, \nu Q^k)^q \leq 2^{q-1}\|\mu Q^k - \nu Q^k\|_{V_q} \leq D_q(\lambda)\rho_\lambda^{k\gamma}(\mu(V_q^2) + \nu(V_q^2)).$$

where $V_q = 1 + \|\cdot\|^q$. In particular, with $\nu = p_{\lambda,\gamma}$, it holds

$$W_q(\mu Q^k, p_{\lambda,\gamma})^q \leq D_q(\lambda)\rho_\lambda^{k\gamma}(\mu(V_q^2) + p_{\lambda,\gamma}(V_q^2)). \tag{44}$$

Thanks to Lemma H.3, $p_{\lambda,\gamma}(V_q^2) < +\infty$, the convergence is exponentially fast given that $\mu \in \mathcal{P}_{2q}(\mathbb{R}^d)$. From (44), we will further show the following uniform bound in the later part of the text: there exist $C_{1,q}(\lambda) > 0$:

$$W_q(p_{Y_{k+1}}, p_{\lambda,\gamma}) \leq C_{1,q}(\lambda)\rho_\lambda^{k\gamma/q}. \tag{45}$$

On another hand, from (Renaud et al., 2025a, Theorem 1), there exists $C_{2,q}(\lambda) > 0$

$$W_q(p_{\lambda,\gamma}, \bar{p}_{\lambda,\gamma}) \leq C_{2,q}(\lambda)\left(L_f + \frac{1}{\lambda}\right)^{\frac{1}{q}}(G_1\gamma)^{\frac{1}{q}}$$

and from (Renaud et al., 2025a, Theorem 2), there exists $C_{3,q}(\lambda) > 0$:

$$W_q(\bar{p}_{\lambda,\gamma}, \pi_{\lambda,\gamma}) \leq C_{3,q}(\lambda)\gamma^{\frac{1}{2q}}.$$

Therefore,

$$W_q(p_{Y_{k+1}}, \pi_{\lambda,\gamma}) \leq C_{1,q}(\lambda)\rho_\lambda^{k\gamma/q} + C_{2,q}(\lambda)\left(L_f + \frac{1}{\lambda}\right)^{\frac{1}{q}} G_1^{\frac{1}{q}}\gamma^{\frac{1}{q}} + C_{3,q}(\lambda)\gamma^{\frac{1}{2q}}.$$

Note that we can absorb $\rho_\lambda := \rho_\lambda^{1/q}$.

Finally, we show the uniform bound (45) which boils down to a uniform bound for $p_{\lambda,\gamma}(V_{2q})$. We have

$$W_{2q}(p_{\lambda,\gamma}, \pi_{\lambda,\gamma}) \leq W_{2q}(p_{\lambda,\gamma}, \bar{p}_{\lambda,\gamma}) + W_{2q}(\bar{p}_{\lambda,\gamma}, \pi_{\lambda,\gamma}). \tag{46}$$

Similar to the arguments of Case 1, the RHS of (46) can be bounded uniformly by some $E(\lambda)$. By Hölder inequality,

$$\|x\|^{2q} \leq 2^{2q-1}(\|y\|^{2q} + \|x - y\|^{2q}).$$

Let $(X, Y)$ be the optimal coupling w.r.t. the cost $d(x, y) = \|x - y\|^{2q}$ of $p_{\lambda,\gamma}$ and $\pi_{\lambda,\gamma}$,

$$\begin{aligned}
\int \|x\|^{2q}dp_{\lambda,\gamma}(x) = \mathbb{E}\|X\|^{2q} \\
\leq 2^{2q-1}\left(\mathbb{E}\|Y\|^{2q} + \mathbb{E}\|X - Y\|^{2q}\right) \\
= 2^{2q-1}\left(\mathbb{E}\|Y\|^{2q} + W_{2q}(p_{\lambda,\gamma}, \pi_{\lambda,\gamma})^{2q}\right) \\
\leq 2^{2q-1}\left(\int \|y\|^{2q}d\pi_{\lambda,\gamma}(y) + E(\lambda)^{2q}\right) \\
\leq 2^{2q-1}\left(e^{\frac{G_1^2\gamma_0}{2}}\int \|y\|^{2q}d\pi_\lambda(y) + E(\lambda)^{2q}\right)
\end{aligned}$$

where the last inequality follows (43).

Lastly, for the sequence $\{X_k\}_k$

$$W_q(p_{X_{k+1}}, \nu_{\lambda,\gamma}) \leq W_q(p_{Y_{k+1}}, \pi_{\lambda,\gamma}).$$

**I.6. Bound $W_q(\pi_\lambda, \pi)$**

We follow the proof template in (Renaud et al., 2025a, Appendix G.3) where we additionally handle the part $-r^\lambda$.

Case 1: Assumption 4(i):

Since $r_i$ is $G_i$-Lipschitz ($i \in \{1, 2\}$), it holds $r_i(x) - r_i^\lambda(x) \leq \frac{\lambda G_i^2}{2}$ for all $x$. Combining with the fact that the Moreau envelope of a function is a lower bound of that function, we get

$$-\frac{\lambda G_2^2}{2} \leq r_1 - r_2 - (r_1^\lambda - r_2^\lambda) \leq \frac{\lambda G_1^2}{2}. \tag{47}$$

Let $\beta := \frac{\lambda G_2^2}{2}$. Given a function $\mathcal{V} : \mathbb{R}^d \to [1, \infty)$ (that induces the $\mathcal{V}$-norm), we write

$$\|\pi_\lambda - \pi\|_\mathcal{V} = \sup_{|\phi| \leq \mathcal{V}} \int \phi(x)(\pi_\lambda(x) - \pi(x))dx$$

$$\leq \sup_{|\phi| \leq \mathcal{V}} \int |\phi(x)||\pi_\lambda(x) - \pi(x)|dx$$

$$\leq \int \mathcal{V}(x)|\pi_\lambda(x) - \pi(x)|dx$$

$$= \int \mathcal{V}(x)\pi(x) \left| 1 - \frac{\pi_\lambda(x)}{\pi(x)} \right| dx$$

$$= \int \mathcal{V}(x)\pi(x) \left| 1 - e^{r_2^\lambda(x) - r_1^\lambda(x) - r_2(x) + r_1(x)} \frac{\int e^{-f-r_1+r_2}}{\int e^{-f-r_1^\lambda+r_2^\lambda}} \right| dx$$

$$= \int \mathcal{V}(x)\pi(x) \left| 1 - e^{r_2^\lambda(x) - r_1^\lambda(x) - r_2(x) + r_1(x) + \beta} \cdot e^{-\beta} \frac{\int e^{-f-r_1+r_2}}{\int e^{-f-r_1^\lambda+r_2^\lambda}} \right| dx$$

$$\leq \int \mathcal{V}(x)\pi(x) \left| 1 - e^{r_2^\lambda(x) - r_1^\lambda(x) - r_2(x) + r_1(x) + \beta} \right| dx$$

$$+ \int \mathcal{V}(x)\pi(x)e^{r_2^\lambda(x) - r_1^\lambda(x) - r_2(x) + r_1(x) + \beta} \left| 1 - e^{-\beta} \frac{\int e^{-f-r_1+r_2}}{\int e^{-f-r_1^\lambda+r_2^\lambda}} \right| dx.$$

Since

$$r_2^\lambda(x) - r_1^\lambda(x) - r_2(x) + r_1(x) + \beta \geq 0 \tag{48}$$

it follows $e^{r_2^\lambda(x) - r_1^\lambda(x) - r_2(x) + r_1(x) + \beta} \geq 1$ and

$$e^{-\beta} \frac{\int e^{-f-r_1+r_2}}{\int e^{-f-r_1^\lambda+r_2^\lambda}} \leq 1,$$

we have

$$\|\pi_\lambda - \pi\|_\mathcal{V} \leq \int \mathcal{V}(x)\pi(x) \left( e^{r_2^\lambda(x) - r_1^\lambda(x) - r_2(x) + r_1(x) + \beta} - 1 \right) dx$$

$$+ \int \mathcal{V}(x)\pi(x)e^{r_2^\lambda(x) - r_1^\lambda(x) - r_2(x) + r_1(x) + \beta} \left( 1 - e^{-\beta} \frac{\int e^{-f-r_1+r_2}}{\int e^{-f-r_1^\lambda+r_2^\lambda}} \right) dx$$

Now using the evaluation (47), we derive

$$\|\pi_\lambda - \pi\|_\mathcal{V} \leq \int \mathcal{V}(x)\pi(x) \left( e^{\frac{\lambda(G_1^2+G_2^2)}{2}} - 1 \right) dx$$

$$+ \int \mathcal{V}(x)\pi(x)e^{\frac{\lambda(G_1^2+G_2^2)}{2}} \left( 1 - e^{-\beta} \frac{\int e^{-f-r_1+r_2}}{\int e^{-f-r_1^\lambda+r_2^\lambda}} \right) dx$$

It also follows from (47) that

$$\frac{\int e^{-f-r_1+r_2}}{\int e^{-f-r_1^\lambda+r_2^\lambda}} \geq e^{-\frac{\lambda G_1^2}{2}}.$$

Therefore

$$\|\pi_\lambda - \pi\|_\mathcal{V} \leq 2 \left( e^{\frac{\lambda(G_1^2+G_2^2)}{2}} - 1 \right) \int \mathcal{V}(x)\pi(x)dx = O(\lambda).$$

It then follows from (Renaud et al., 2025a, Lemma 18) that (by using $\mathcal{V} = 1 + \| \cdot \|^q$)

$$W_q(\pi_\lambda, \pi) = O(\lambda^{1/q}).$$

Case 2: Assumption 4(ii):

It is a standard result that

$$r_2(x) - r_2^\lambda(x) \le \frac{\lambda}{2}\|\nabla r_2(x)\|^2, \quad \forall x \in \mathbb{R}^d.$$

Indeed, a short proof is as follows

$$
\begin{aligned}
r_2(x) - r_2^\lambda(x) &= r_2(x) - \inf_y \left\{ r_2(y) + \frac{1}{2\lambda}\|x - y\|^2 \right\} \\
&= \sup_y \left\{ r_2(x) - r_2(y) - \frac{1}{2\lambda}\|x - y\|^2 \right\} \\
&\le \sup_y \left\{ \langle \nabla r_2(x), x - y \rangle - \frac{1}{2\lambda}\|x - y\|^2 \right\} \\
&= \sup_y \left\{ -\frac{1}{2\lambda}\|y - x + \lambda \nabla r_2(x)\|^2 + \frac{\lambda}{2}\|\nabla r_2(x)\|^2 \right\} \\
&\le \frac{\lambda}{2}\|\nabla r_2(x)\|^2.
\end{aligned}
$$

Under Assumption 4(ii):

$$\|\nabla r_2(x)\|^2 \le 2(\|\nabla r_2(0)\|^2 + M^2\|x\|^{2\kappa}).$$

Therefore

$$r_2(x) - r_2^\lambda(x) \le \lambda(\|\nabla r_2(0)\|^2 + M^2\|x\|^{2\kappa})$$

and

$$|r_1(x) - r_2(x) - r_1^\lambda(x) + r_2^\lambda(x)| \le \frac{\lambda G_1^2}{2} + \lambda\|\nabla r_2(0)\|^2 + \lambda M^2\|x\|^{2\kappa}.$$

By applying

$$|e^t - 1| \le |t|e^{|t|}, \quad \forall t \in \mathbb{R},$$

we have

$$
\begin{aligned}
\|\pi_\lambda - \pi\|_{\mathcal{V}_q} &\leq \int \mathcal{V}_q(x)\pi(x)\left|1 - e^{r_2^\lambda(x)-r_1^\lambda(x)-r_2(x)+r_1(x)}\frac{\int e^{-f-r_1+r_2}}{\int e^{-f-r_1^\lambda+r_2^\lambda}}\right|dx \\
&\leq \int \mathcal{V}_q(x)\pi(x)\left|1 - e^{r_2^\lambda(x)-r_1^\lambda(x)-r_2(x)+r_1(x)}\right|dx \\
&\quad + \int \mathcal{V}_q(x)\pi(x)e^{r_2^\lambda(x)-r_1^\lambda(x)-r_2(x)+r_1(x)}dx\left|1 - \frac{\int e^{-f-r_1+r_2}}{\int e^{-f-r_1^\lambda+r_2^\lambda}}\right| \\
&\leq \int \mathcal{V}_q(x)\pi(x)|r_2^\lambda(x)-r_1^\lambda(x)-r_2(x)+r_1(x)|e^{|r_2^\lambda(x)-r_1^\lambda(x)-r_2(x)+r_1(x)|}dx \\
&\quad + e^{\lambda G_1^2/2}\pi(\mathcal{V}_q)\left|1 - \frac{\int e^{-f-r_1+r_2}}{\int e^{-f-r_1^\lambda+r_2^\lambda}}\right| \\
&\leq \underbrace{\int \mathcal{V}_q(x)\pi(x)\left(\frac{\lambda G_1^2}{2} + \lambda\|\nabla r_2(0)\|^2 + \lambda M^2\|x\|^{2\kappa}\right)e^{\frac{\lambda G_1^2}{2}+\lambda\|\nabla r_2(0)\|^2+\lambda M^2\|x\|^{2\kappa}}dx}_{I} \\
&\quad + \underbrace{e^{\lambda G_1^2/2}\pi(\mathcal{V}_q)\left|1 - \frac{\int e^{-f-r_1+r_2}}{\int e^{-f-r_1^\lambda+r_2^\lambda}}\right|}_{II}.
\end{aligned}
$$

Since $\pi$ has sub-Gaussian tails (Appendix A),

$$
\int \mathcal{V}_q(x)\|x\|^{2\kappa}e^{\lambda M^2\|x\|^{2\kappa}}\pi(x) < +\infty \tag{49}
$$

for all $\kappa \in (0,1)$. To see this, recall $\pi \propto e^{-V}$ and there exist $a, b > 0$:

$$
V(x) \geq a\|x\|^2 - b.
$$

Let $\bar{a} = a/(\lambda M^2\kappa)$. By Young's inequality

$$
\|x\|^{2\kappa} \leq \frac{\kappa\bar{a}}{2}\|x\|^2 + \left(\frac{2}{\bar{a}}\right)^{\frac{\kappa}{1-\kappa}}(1-\kappa).
$$

Therefore,

$$
V(x) - \lambda M^2\|x\|^{2\kappa} \geq \frac{a}{2}\|x\|^2 - c
$$

for some $c$. So $e^{-V(x)+\lambda M^2\|x\|^{2\kappa}}$ also has sub-Gaussian tails, which guarantees the finiteness of (49). As a result, $I = O(\lambda)$. Now that

$$
\int e^{-\lambda\left(\|\nabla r_2(0)\|^2+M^2\|x\|^{2\kappa}\right)}\pi(x)\,dx \leq \frac{\int e^{-f(x)-r_1^\lambda(x)-r_2^\lambda(x)}\,dx}{\int e^{-f(x)-r_1(x)+r_2(x)}\,dx} \leq e^{\lambda G_1^2/2}.
$$

Therefore,

$$\left|1 - \frac{\int e^{-f-r_1+r_2}}{\int e^{-f-r_1^\lambda+r_2^\lambda}}\right| \leq \left|1 - e^{-\lambda G_1^2/2}\right| + \left|1 - \frac{e^{\lambda\|\nabla r_2(0)\|^2}}{\int e^{-\lambda M^2\|x\|^{2\kappa}}\pi(x)dx}\right|$$

$$\leq \lambda \frac{G_1^2}{2}e^{\lambda G_1^2/2} + \frac{|1 - e^{\lambda\|\nabla r_2(0)\|^2}| + |1 - \int e^{-\lambda M^2\|x\|^{2\kappa}}\pi(x)dx|}{\int e^{-\lambda M^2\|x\|^{2\kappa}}\pi(x)dx}$$

$$\leq \lambda \frac{G_1^2}{2}e^{\lambda G_1^2/2} + \frac{\lambda\|\nabla r_2(0)\|^2 e^{\lambda\|\nabla r_2(0)\|^2} + \int |1 - e^{-\lambda M^2\|x\|^{2\kappa}}|\pi(x)dx}{\int e^{-\lambda M^2\|x\|^{2\kappa}}\pi(x)dx}$$

$$\leq \lambda \frac{G_1^2}{2}e^{\lambda G_1^2/2} + \frac{\lambda\|\nabla r_2(0)\|^2 e^{\lambda\|\nabla r_2(0)\|^2} + \lambda M^2 \int \|x\|^{2\kappa}e^{\lambda M^2\|x\|^{2\kappa}}\pi(x)dx}{\int e^{-\lambda M^2\|x\|^{2\kappa}}\pi(x)dx}$$

$$= O(\lambda)$$

thanks to $\pi$ having sub-Gaussian tails.

## I.7. On the dependence of the bounds on key parameters

For simplicity, we discuss the $W_1$-distant case.

Recall that the general drift $b_\lambda^\gamma$ is $L$ smooth and $(m^+, R)$ distant dissipative, where $L = \frac{2}{\lambda} + L_f$, $m^+ = \frac{\mu}{2}$, and $R = \max\left\{R_0, \frac{8G_1+4L_f\bar\gamma G_1+4G_2}{\mu}\right\}$ under Assumption 4(i) or $R = \max\left\{R_0, \frac{16G_1+8L_f\bar\gamma G_1}{\mu}, \left(\frac{4M}{\mu}\right)^{\frac{1}{1-\kappa}}\right\}$ under Assumption 4(ii), here $\bar\gamma = m^+/L^2$. We discuss *the difficult paradigm*: $LR^2 \gg 1$. The limit of interest is indeed: small $\lambda$ and $\mu$, large $L_f, R_0, G_1, G_2, M$.

The constant $\rho_\lambda$ in (34) comes from Corollary 2 of (De Bortoli & Durmus, 2019) and is given in the equation (10) therein (with $m = -L$):

$$-\log(\log(\rho_\lambda^{-1})) \simeq (LR^2/4) \sup_{\gamma\in(0,\bar\gamma]} \left\{\left(1 + \frac{\gamma L}{2}\right)\left(1 - \exp\left[-\frac{R^2 L(2+\gamma L)}{1+2\gamma L+\gamma^2 L^2}\right]\right)^{-1}\right\} \tag{50}$$

where $\bar\gamma = m^+/L^2$ and $\simeq$ denotes equality up to logarithmic factors. The supremum term is upper bounded at the limits of interest. Indeed, for $\gamma \leq \bar\gamma$:

$$\left(1 + \frac{\gamma L}{2}\right)\left(1 - \exp\left[-\frac{R^2 L(2+\gamma L)}{1+2\gamma L+\gamma^2 L^2}\right]\right)^{-1} \leq \left(1 + \frac{m^+}{2L}\right)\left(1 - \exp\left[-\frac{2R^2 L}{(1+\gamma L)^2}\right]\right)^{-1}$$

$$\leq \left(1 + \frac{m^+}{2L}\right)\left(1 - \exp\left[-\frac{2R^2 L}{(1+m^+/L)^2}\right]\right)^{-1}$$

$$= O(1).$$

Therefore, $-\log(\log(\rho_\lambda^{-1})) \simeq LR^2$.

By Lemma H.4, there exists $x^*$: $b_\lambda^\gamma(x^*) = 0$ and we can shift:

$$\hat b_\lambda^\gamma(x) := b_\lambda^\gamma(x + x^*), \tag{51}$$

to make $\hat b_\lambda^\gamma(0) = 0$. We denote the sequence corresponding to $\hat b_\lambda^\gamma$ as $\hat X_{k+1} = \hat X_k - \gamma\hat b_\lambda^\gamma(\hat X_k) + \sqrt{2\gamma}\hat Z_{k+1}$ and let $\hat Q$ be the Markov transition kernel corresponding to this sequence. Since $(\hat X_{k+1} + x^*) = (\hat X_k + x^*) - \gamma b_\lambda^\gamma(\hat X_k + x^*) + \sqrt{2\gamma}\hat Z_{k+1}$, the law of $X_k|(X_0 = x + x^*)$ and the law of $\hat X_k|(\hat X_0 = x) + x^*$ are the same. Since we choose $x^*$ by Lemma H.4, $\|x^*\| \leq \max\{R, (\sqrt{2}/m^+)\|b_\lambda^\gamma(0)\|\}$. Moreover, we can bound $\|b_\lambda^\gamma(0)\|$ by $O(G_1 + G_2 + \|\nabla f(0)\|)$ if Assumption 4(i) or $O(G_1 + \|\nabla f(0)\| + M^{1/(1-\kappa)} + \|\nabla r_2(0)\|)$ if Assumption 4(ii), so $\|x^*\|$ is bounded uniformly w.r.t. small $\lambda$ and $\gamma$.

Applying De Bortoli & Durmus (2019, Thm. 13) (also see the discussion right after Theorem 13) to $b_\lambda^\gamma$: let $\mathcal{V}(x,y) := 1 + \|x-y\|/R$, $\mathbf{c}(x,y) = 1(x \neq y)\mathcal{V}(x,y)$ and $W_\mathbf{c}$ the Wasserstein distance associated with the cost $\mathbf{c}$, it holds

$$W_\mathbf{c}(\delta_x Q^k, \delta_y Q^k) = W_\mathbf{c}(\delta_{x-x^*}\hat Q^k, \delta_{y-x^*}\hat Q^k) \leq \left(D_{\bar\gamma,1,a} + D_{\bar\gamma,2,a} + C_{\bar\gamma,a}\right)\rho^{k\gamma/4}\mathbf{c}(x,y)$$

for all $\gamma \in (0, \bar{\gamma}]$, where $\bar{\gamma} = m^+/L^2$, $\rho$ is the same as $\rho_\lambda$ in (34) and (50) (up to a constant power), and

$$D_{\bar{\gamma},1,a} = 1 + 4A \log^{-1}\left(\frac{1}{\hat{\lambda}}\right)/\hat{\lambda}^{\bar{\gamma}}$$

$$D_{\bar{\gamma},2,a} = D_{\bar{\gamma},1,a}A\hat{\lambda}^{-(1+\bar{\gamma})\ell}(1+\bar{\gamma})\ell$$

$$C_{\bar{\gamma},a} = \frac{8A}{\log(1/\rho)\rho^{\bar{\gamma}}}$$

$$\hat{\lambda} = \exp\left[-\frac{1}{2}\left(m^+ - \frac{\bar{\gamma}L^2}{2}\right)\right]$$

$$A = m^+ + L$$

$$\ell = \lceil R^2 \rceil$$

We simplify: $\hat{\lambda} = \exp\left[-(1/4)m^+\right]$.

$\underline{D_{\bar{\gamma},1,a}}$:

$$D_{\bar{\gamma},1,a} = 1 + \frac{16(m^+ + L)}{m^+ \exp\left[-\frac{(m^+)^2}{4L^2}\right]} = O(L/m^+).$$

$\underline{D_{\bar{\gamma},2,a}}$:

$$D_{\bar{\gamma},2,a} = D_{\bar{\gamma},1,a}(m^+ + L)\exp\left(\frac{m^+}{4}(1+\bar{\gamma})\ell\right)(1+\bar{\gamma})\ell$$

$$= O\left(\frac{L^2\ell}{m^+}\exp(m^+\ell)\right) = O\left(\frac{L^2R^2}{m^+}\exp(m^+R^2)\right).$$

$\underline{C_{\bar{\gamma},a}}$:

Since $\rho \uparrow 1$ in the hard regime,

$$C_{\bar{\gamma},a} \sim \frac{8(m^+ + L)}{\log(1/\rho)}.$$

Using $\log(\log^{-1}(\rho^{-1})) \simeq LR^2$,

$$C_{\bar{\gamma},a} = \exp(\tilde{O}(LR^2)).$$

Now with

$$\Psi(\gamma, \ell, t) = 2\Phi\left(-\frac{t}{2\Xi_{\ell\lceil 1/\gamma\rceil}^{1/2}(\kappa)}\right)$$

$$\Xi_n(\kappa) = \gamma\sum_{k=1}^{n}(1 + \gamma\kappa(\gamma))^{-k}$$

$$\kappa(\gamma) = 2L + L^2\gamma$$

where $\Phi$ is the CDF of $\mathcal{N}(0,1)$, applying Corollary 14 and Proposition 9 in (De Bortoli & Durmus, 2019), we can go from $W_{\mathbf{c}}$ to $W_1$:

$$\frac{1}{R}W_1(\delta_{x-x^*}\hat{Q}^k, \delta_{y-x^*}\hat{Q}^k) \leq \left[-\mathbf{a}(D_{\bar{\gamma},1,a} + D_{\bar{\gamma},2,a} + C_{\bar{\gamma},a})/\rho^{1/4} + \frac{D_{\bar{\gamma},1,a}}{R}\exp[\kappa(1+\bar{\gamma})]/\rho^{(1+\bar{\gamma})/4}\right]\rho^{k\gamma/4}\|x-y\|$$

where $\mathbf{a} = \inf_{\gamma \in (0,\bar{\gamma}]} \mathbf{\Psi}'(\gamma, 1, 0) \geq -(\pi \inf_{\gamma \in (0,\bar{\gamma}]} \Xi_{\lceil 1/\gamma \rceil}(\kappa))^{-1/2}$. Moreover,

$$
\begin{aligned}
\Xi_{\lceil 1/\gamma \rceil}(\kappa) &= \gamma \sum_{k=1}^{\lceil 1/\gamma \rceil} (1 + \gamma \kappa(\gamma))^{-k} \\
&= \frac{1 - (L\gamma + 1)^{-2\lceil 1/\gamma \rceil}}{L(2 + L\gamma)} \\
&\geq \frac{1 - (L\gamma + 1)^{-2\lceil 1/\gamma \rceil}}{L(2 + L\bar{\gamma})} \\
&\geq \frac{1 - \exp\left(-\frac{2}{\gamma} \log(1 + \gamma L)\right)}{L(2 + L\bar{\gamma})} \\
&\geq \frac{1 - \exp\left(-\frac{2L}{1+\gamma L}\right)}{L(2 + L\bar{\gamma})} \\
&\geq \frac{1 - \exp\left(-\frac{2L}{1+\bar{\gamma} L}\right)}{L(2 + L\bar{\gamma})}.
\end{aligned}
$$

Therefore

$$
-\mathbf{a} \leq \pi^{-1/2} \left( \frac{m^+ + 2L}{1 - \exp\left(-\frac{2L^2}{m^+ + L}\right)} \right)^{1/2} = O(L^{1/2}).
$$

And we get

$$
W_1(\delta_x Q^k, \delta_y Q^k) = W_1(\delta_{x-x^*} \hat{Q}^k, \delta_{y-x^*} \hat{Q}^k) \leq e^{\tilde{O}(LR^2)} \rho^{k\gamma/4} \|x - y\|.
$$

And we obtain the constant $D(\lambda)$ in (34) as $D(\lambda) = e^{\tilde{O}(LR^2)}$.

Next, $C_2(\lambda)$ in (36) is given as (see Renaud et al. (2025a, Appendix F.5))

$$
C_2(\lambda) = \frac{2}{1 - \rho^{1-\bar{\gamma}}} \hat{D}(1 + 2M_2)(2M_4 + 2M_4^2)^{1/2} \tag{52}
$$

where $\rho$ is just $\rho_\lambda$ up to a constant power, $\hat{D} \propto \hat{E}\sqrt{M_2}$, $\hat{E} = E_{\bar{\gamma},1}^{1/2}$ given in Corollary 2 in (De Bortoli & Durmus, 2019), and $M_2, M_4$ can be retrieved from Lemma 16 and Lemma 17 in (Laumont et al., 2022). Since $\|\mu - \nu\|_{TV} \leq W_{\mathbf{c}}(\mu, \nu)$, $E_{\bar{\gamma},1}$ is given by

$$
E_{\bar{\gamma},1} = D_{\bar{\gamma},1,a} + D_{\bar{\gamma},2,a} + C_{\bar{\gamma},a} = e^{\tilde{O}(LR^2)}.
$$

We now give estimates for $M_{2\varpi}$ for $\varpi \in \mathbb{N}$. Recall that $M_{2\varpi}$ comes from the inequality: $Q^k V_{2\varpi}(x) \leq M_{2\varpi} V_{2\varpi}(x)$ for all $k$ and $x$. Here $V_{2\varpi}(x) = 1 + \|x\|^{2\varpi}$. To obtain the form of $M_{2\varpi}$, we first need Lemma 16 in (Laumont et al., 2022), which is stated as in the following lemma.

**Lemma I.1.** *Let $b$ be a drift that is $L$-Lipschitz, $(m^+, R)$-distant dissipative, and $b(0) = 0$. Let $R$ be the Markov transition kernel of $X_{k+1} = X_k - \gamma b(X_k) + \sqrt{2\gamma} Z_{k+1}$. Let $\bar{\gamma} = m^+/L^2$, then: for any $\varpi \in \mathbb{N}^*$, $\gamma \in (0, \bar{\gamma}]$, $b$ satisfies the Foster–Lyapunov drift condition:*

$$
R V_{2\bar{\varpi}}(x) \leq \check{\eta}^{\gamma/2} V_{2\varpi}(x) + \check{C}\gamma,
$$

*where $\check{\eta} = \exp(-m^+ \varpi/2)$ and $\check{C} = O\left( \frac{L^{8\varpi^2} R^{4\varpi^2}}{(m^+)^{4\varpi^2 + 2\varpi - 1}} \right)$.*

*Proof.* We restate the proof of (Laumont et al., 2022) to track the dependence of $\check{\eta}$ and $\check{C}$ on the drift's parameters.

By separating two cases, $\|x\| \le R$ and $\|x\| > R$, we can show that:

$$\langle b(x), x \rangle \ge m^+ \|x\|^2 - (m^+ + L)R^2 = m^+ \|x\|^2 - \hat{d}, \quad \forall x \in \mathbb{R}^d.$$

Let $T_\gamma(x) = x - \gamma b(x)$. Consider $\|x\|^2 \ge 4\hat{d}/m^+$,

$$
\begin{aligned}
\|T_\gamma(x)\| &= \left( \|x\|^2 - 2\gamma \langle b(x), x \rangle + \gamma^2 \|b(x)\|^2 \right)^{1/2} \\
&\le \left[ \|x\|^2 - 2\gamma \left( m^+ \|x\|^2 - \hat{d} \right) + \gamma^2 L^2 \|x\|^2 \right]^{1/2} \\
&\le \left[ (1 - \gamma m^+) \|x\|^2 + (\gamma m^+/2)\|x\|^2 \right]^{1/2} \\
&\le e^{-\gamma m^+/4} \|x\|.
\end{aligned}
$$

Consider the case $\|x\|^2 < 4\hat{d}/m^+$:

$$
\begin{aligned}
\|T_\gamma(x)\| &\le (1 + \gamma L)\|x\| \\
&\le e^{-\gamma m^+/4}\|x\| + e^{\gamma L}\|x\| - e^{-\gamma m^+/4}\|x\| \\
&\le e^{-\gamma m^+/4}\|x\| + e^{\gamma L}(\gamma L + \gamma m^+/4)\|x\| \\
&\le e^{-\gamma m^+/4}\|x\| + 2\gamma e^{m^+/L}(L + m^+/4)\sqrt{\frac{\hat{d}}{m^+}}
\end{aligned}
$$

Combine two cases,

$$\|T_\gamma(x)\| \le \hat{\eta}^\gamma \|x\| + \gamma \hat{c}$$

where $\hat{\eta} = \exp(-m^+/4)$ and $\hat{c} = O(L^2 R/m^+)$.

Therefore, for $k \in \mathbb{N}$,

$$
\begin{aligned}
\|T_\gamma(x)\|^k &\le \hat{\eta}^{k\gamma}\|x\|^k + \gamma 2^k \max\{\hat{c}, 1\}^k \max\{\bar{\gamma}, 1\}^{k-1} \left( 1 + \|x\|^{k-1} \right) \\
&\le \tilde{\eta}_k^\gamma \|x\|^k + \tilde{c}_k \gamma (1 + \|x\|^{k-1}),
\end{aligned}
$$

where $\tilde{\eta}_k = \hat{\eta}^k = \exp(-m^+ k/4)$ and $\tilde{c}_k = O(\hat{c}^k)$. Now,

$$
\begin{aligned}
\int_{\mathbb{R}^d} (1 + \|y\|^{2\varpi}) R(x, dy) \le & 1 + \tilde{\eta}_{2\varpi}^\gamma \|x\|^{2\varpi} + \tilde{c}_{2\varpi} \gamma \{1 + \|x\|^{2\varpi - 1}\} \\
& + \gamma 2^{3\varpi/2} 2^{2\varpi} \max(\bar{\gamma}, 1)^{2\varpi} \sup_{k \in \{1, \dots, \varpi\}} \{(1 + \tilde{c}_k \bar{\gamma}) \mathbb{E}[\|Z\|^k]\}(1 + \|x\|^{2\varpi - 1}) \\
\le & 1 + \check{\eta}^\gamma \|x\|^{2\varpi} + \gamma \check{c}(1 + \|x\|^{2\varpi - 1}),
\end{aligned}
\tag{53}
$$

where $\check{\eta} = \tilde{\eta}_{2\varpi} = \exp(-m^+ \varpi/2)$ and $\check{c} = O(\tilde{c}_{2\varpi}) = O(\hat{c}^{2\varpi})$. We then evaluate

$$
\begin{aligned}
& 1 + \check{\eta}^\gamma \|x\|^{2\varpi} + \gamma \check{c}(1 + \|x\|^{2\varpi - 1}) \\
&= \check{\eta}^{\gamma/2} V_{2\varpi}(x) + \gamma \check{c}(1 + \|x\|^{2\varpi - 1}) + (1 - \check{\eta}^{\gamma/2}) + (\check{\eta}^\gamma - \check{\eta}^{\gamma/2})\|x\|^{2\varpi} \\
&\le \check{\eta}^{\gamma/2} V_{2\varpi}(x) + \gamma \check{c}(1 + \|x\|^{2\varpi - 1}) + \frac{1}{2}\gamma \log\left(\frac{1}{\check{\eta}}\right) - \log\left(\frac{1}{\check{\eta}}\right) \gamma \frac{\check{\eta}^{\gamma/2}}{2} \|x\|^{2\varpi}.
\end{aligned}
\tag{54}
$$

As an elementary evaluation: for $s \ge 0, p \ge 2$

$$
as^{p-1} - bs^p \le \frac{a^p}{b^{p-1}} \left[ \underbrace{\left(\frac{p-1}{p}\right)^{p-1} - \left(\frac{p-1}{p}\right)^p}_{:=\varphi(p)} \right].
$$

We have

$$1 + \check{\eta}^\gamma \|x\|^{2\varpi} + \gamma \check{c}(1 + \|x\|^{2\varpi-1})$$

$$\leq \check{\eta}^{\gamma/2} V_{2\varpi}(x) + \gamma \left[ \check{c} + \frac{1}{2} \log\left(\frac{1}{\check{\eta}}\right) + \check{c}\|x\|^{2\varpi-1} - \log\left(\frac{1}{\check{\eta}}\right) \frac{\check{\eta}^{\gamma/2}}{2} \|x\|^{2\varpi} \right]$$

$$\leq \check{\eta}^{\gamma/2} V_{2\varpi}(x) + \gamma \left[ \check{c} + \frac{1}{2} \log\left(\frac{1}{\check{\eta}}\right) + \frac{\check{c}^{2\varpi}}{\left(\log\left(\frac{1}{\check{\eta}}\right) \frac{\check{\eta}^{\gamma/2}}{2}\right)^{2\varpi-1}} \varphi(2\varpi) \right] \tag{55}$$

$$= \check{\eta}^{\gamma/2} V_{2\varpi}(x) + \gamma \left[ \check{c} + \frac{1}{2} \frac{m^+ \varpi}{2} + \frac{\check{c}^{2\varpi}}{\left(\frac{m^+ \varpi}{2} \frac{\check{\eta}^{\gamma/2}}{2}\right)^{2\varpi-1}} \varphi(2\varpi) \right] \tag{56}$$

From (53), (54), and (55),

$$RV_{2\varpi}(x) \leq \check{\eta}^{\gamma/2} V_{2\varpi}(x) + \gamma \check{C}$$

where $\check{C} = O(\check{c}^{2\varpi}/(m^+)^{2\varpi-1}) = O(\hat{c}^{4\varpi^2}/(m^+)^{2\varpi-1})$ and $\check{\eta} = \exp(-m^+\varpi/2)$. $\qquad\square$

With the two-stepsize technique as in (39), we can apply Lemma I.1 to $\hat{b}_\lambda^\gamma$ we can apply Lemma 17 in (Laumont et al., 2022):

$$\hat{Q}^k V_{2\varpi}(x) \leq \left(1 + \check{C}\left(\bar{\gamma} + \log\left(\frac{1}{\check{\eta}^{1/2}}\right)\right)\right) V_{2\varpi}(x) = O\left(\check{C}\right) V_{2\varpi}(x)$$

for all $k, x$. Now we have

$$Q^k V_{2\varpi}(x) = 1 + \mathbb{E}\left[\|X_k\|^{2\varpi} | X_0 = x\right]$$

$$= 1 + \mathbb{E}\left[\|\hat{X}_k + x^*\|^{2\varpi} | \hat{X}_0 = x - x^*\right]$$

$$\leq 1 + 2^{2\varpi-1}\left(\mathbb{E}(\|\hat{X}_k\|^{2\varpi} | \hat{X}_0 = x - x^*) + \|x^*\|^{2\varpi}\right)$$

$$= 1 + 2^{2\varpi-1}(\|x^*\|^{2\varpi} - 1) + 2^{2\varpi-1}\hat{Q}^k V_{2\varpi}(x - x^*)$$

$$\leq 1 + 2^{2\varpi-1}(\|x^*\|^{2\varpi} - 1) + 2^{2\varpi-1}O(\check{C})V_{2\varpi}(x - x^*)$$

$$= O(\check{C}(1 + \|x^*\|^{2\varpi}))V_{2\varpi}(x).$$

Now with the uniform bound of $\|x^*\|$, we obtain

$$Q^k V_{2\varpi}(x) \leq \begin{cases} O\left(\check{C}\left(\max\{R, \sqrt{2}/m^+\}(G_1 + G_2 + \|\nabla f(0)\|)\right)^{2\varpi}\right) V_{2\varpi}(x) & \text{if Assumption 4(i)} \\ O\left(\check{C}\left(\max\{R, \sqrt{2}/m^+\}(G_1 + \|\nabla f(0)\| + M^{1/(1-\kappa)} + \|\nabla r_2(0)\|)\right)^{2\varpi}\right) V_{2\varpi}(x) & \text{if Assumption 4(ii)} \end{cases}$$

and $M_{2\varpi}$ are then bounded accordingly: $O\left(\check{C}\left(\max\{R, \sqrt{2}/m^+\}(G_1 + G_2 + \|\nabla f(0)\|)\right)^{2\varpi}\right)$ if Assumption 4(i), and $O\left(\check{C}\left(\max\{R, \sqrt{2}/m^+\}(G_1 + \|\nabla f(0)\| + M^{1/(1-\kappa)} + \|\nabla r_2(0)\|)\right)^{2\varpi}\right)$ if Assumption 4(ii). We observe that these constants depend polynomially on the parameters, i.e., $\text{poly}(\Theta)$ where $\Theta = (R, 1/m^+, L_f, G_1, G_2, \|\nabla f(0)\|)$ if Assumption 4(i) and $\Theta = (R, 1/m^+, L_f, G_1, M, \|\nabla f(0)\|, \|\nabla r_2(0)\|)$ if Assumption 4(ii).

Now back to the expression of $C_2(\lambda)$ in (52), we have:

$$\frac{1}{1 - \rho^{1-\bar{\gamma}}} \sim \frac{1}{(1 - \bar{\gamma}) \log(1/\rho)} = e^{\tilde{O}(LR^2)}$$

$$\hat{D} = O(\hat{E}\sqrt{M_2}) = \text{poly}(\Theta)e^{\tilde{O}(LR^2)}$$

implying $C_2(\lambda) = \mathrm{poly}(\Theta)e^{\tilde{O}(LR^2)}$. Here we do not absorb the poly term into $\exp^{\tilde{O}}$ because the poly term also has some extra parameters like $\|\nabla f(0)\|, \|\nabla r_2(0)\|$. If we keep these extra parameters fixed, we can then absorb.

Similarly, $C_3(\lambda) = \mathrm{poly}(\Theta)e^{\tilde{O}(LR^2)}$. As a consequence, $C_1(\lambda) = O(D(\lambda)(C_2(\lambda) + C_3(\lambda))) = \mathrm{poly}(\Theta)e^{\tilde{O}(LR^2)}$. Therefore, $A_\lambda = \mathrm{poly}(\Theta)e^{\tilde{O}(LR^2)}$, $B_\lambda = \mathrm{poly}(\Theta)e^{\tilde{O}(LR^2)}$ (Theorem 4.4) and also $B'_\lambda = \mathrm{poly}(\Theta)e^{\tilde{O}(LR^2)}$ (Theorem 4.6). Hence, the key parameters $\lambda, L_f, R_0, \mu, G_1, G_2, M$ enter the bound through $L, R$ and $\Theta$.

Finally, tracing from the proof in Section I.6, we obtain the dependence of $C$ in Theorem 4.6 as $C = O(G_1^2 + G_2^2)$ if Assumption 4(i) and $O(G_1^2 + M^2 + \|\nabla r_2(0)\|^2)$ if Assumption 4(ii).

## I.8. Analysis of DC-LA-S

We recall DC-LA-S

$$Y_{k+1} = X_k - \gamma \nabla f(X_k) + \gamma \nabla r_2(X_k) + \sqrt{2\gamma} Z_{k+1}$$
$$X_{k+1} = \mathrm{Prox}_{\gamma r_1^\lambda}(Y_{k+1}).$$

Similarly to Section 4.1, $Y_{k+1}$ is formulated as

$$Y_{k+1} = Y_k - \gamma b_\lambda^\gamma(Y_k) + \sqrt{2\gamma} Z_{k+1} \tag{57}$$

where the drift is

$$b_\lambda^\gamma(y) := \nabla r_1^\lambda(\mathrm{Prox}_{\gamma r_1^\lambda}(y)) + \nabla f(\mathrm{Prox}_{\gamma r_1^\lambda}(y)) - \nabla r_2(\mathrm{Prox}_{\gamma r_1^\lambda}(y)). \tag{58}$$

$b_\lambda^\gamma$ is $(L_f + \frac{1}{\lambda} + L_{r_2})$-smooth. Similarly to the evaluations in I.3, for $\gamma \leq \gamma_0$, we have

$$\langle b_\lambda^\gamma(x) - b_\lambda^\gamma(y), x - y \rangle \geq \mu \|x - y\|^2 - 2G_1(2 + L_f \gamma_0 + L_{r_2})\|x - y\|$$

for $\|x - y\| \geq R_0$. Therefore, $b_\lambda^\gamma$ is $(\frac{\mu}{2}, R)$-distant dissipative, where

$$R := \max\left\{ R_0, \frac{4G_1(2 + L_f \gamma_0 + L_{r_2})}{\mu} \right\}.$$

We next define another drift

$$\bar{b}_\lambda^\gamma(y) = \nabla r_1^\lambda(\mathrm{Prox}_{\gamma r_1^\lambda}(y)) + \nabla f(y) - \nabla r_2(y), \tag{59}$$

and the corresponding general ULA:

$$\bar{Y}_{k+1} = \bar{Y}_k - \gamma \bar{b}_\lambda^\gamma(\bar{Y}_k) + \sqrt{2\gamma} \bar{Z}_{k+1}. \tag{60}$$

We also denote

$$\pi_{\lambda,\gamma}(x) \propto \exp(-f(x) - (r_1^\lambda)^\gamma(x) + r_2(x)),$$

it follows that $-\nabla \log \pi_{\lambda,\gamma} = \bar{b}_\lambda^\gamma$.

Similarly, we can show that: for all $\gamma > 0$, $\bar{b}_\lambda^\gamma$ is $(\frac{1}{\lambda} + L_f + L_{r_2})$-smooth and is $(\frac{\mu}{2}, \bar{R})$-distant dissipative, where

$$\bar{R} = \max\left\{ R_0, \frac{8G_1}{\mu} \right\}.$$

Let $Q, \bar{Q}$ be the transition kernels of $\{Y_k\}_k$ and $\{\bar{Y}_k\}_k$, respectively. According to Lemma H.3, $Q, \bar{Q}$ admit invariant distributions called $p_{\lambda,\gamma}, \bar{p}_{\lambda,\gamma} \in \mathcal{P}_1(\mathbb{R}^d)$. Furthermore, if $q \geq 2$, Lemma H.3 guaranties $p_{\lambda,\gamma}, \bar{p}_{\lambda,\gamma} \in \mathcal{P}_{2q}(\mathbb{R}^d)$.

Similarly to the arguments in Section I.5, there exist $C_{1,q}(\lambda), C_{2,q}(\lambda), C_{3,q}(\lambda) > 0$ and $\rho_{q,\lambda} \in (0,1)$ such that

$$W_q(p_{Y_{k+1}}, \pi_{\lambda,\gamma}) \leq C_{1,q}(\lambda)\rho_{q,\lambda}^{k\gamma/q} + C_{2,q}(\lambda)(L_f + L_{r_2})^{\frac{1}{q}} G_1^{\frac{1}{q}}\gamma^{\frac{1}{q}} + C_{3,q}(\lambda)\gamma^{\frac{1}{2q}}.$$

Finally, we can apply (Renaud et al., 2025a, Proposition 1) two times to get the following

$$W_q(\pi_{\lambda,\gamma}, \pi_\lambda) = O(\gamma^{1/q})$$
$$W_q(\pi_\lambda, \pi) = O(\lambda^{1/q}).$$

## J. Additional experimental results and details

### J.1. Standard proximal operators

It is well-known that the proximal operators of $\ell_1$ and $\ell_2$ are given by

$$\text{Prox}_{\gamma\ell_1}(x) = \text{sign}(x) \odot \max(|x| - \gamma, 0),$$
$$\text{Prox}_{\gamma\ell_2}(x) = \max\left(1 - \frac{\gamma}{\|x\|_2}, 0\right)x.$$

Recently, Lou & Yan (2018) showed that the proximal operator of $\ell_1 - \ell_2$, or more generally, $\ell_1 - \epsilon\ell_2$ with $\epsilon > 0$, is given in closed form as

$$\text{Prox}_{\gamma(\ell_1 - \epsilon\ell_2)}(x) \ni \begin{cases} 0, & \text{if } \|x\|_\infty \leq (1-\epsilon)\gamma, \\ \left(1 + \frac{\epsilon\gamma}{\|\text{Prox}_{\gamma\ell_1}(x)\|_2}\right)\text{Prox}_{\gamma\ell_1}(x), & \text{if } \|x\|_\infty > \gamma, \\ \text{sign}(x_{i^\star})(\|x\|_\infty + (\epsilon - 1)\gamma)e_{i^\star}, & \text{if } (1-\epsilon)\gamma < \|x\|_\infty \leq \gamma, \text{ where } i^\star = \arg\max_i |x_i|. \end{cases}$$

Note that, in the above formulation, whenever $\text{prox}_{\gamma(\ell_1 - \epsilon\ell_2)}(x)$ is set-valued, we use an arbitrary but fixed single-valued selection (any element of the proximal set), which suffices for implementation.

### J.2. $\ell_1 - \ell_2$ prior

**Histograms of samples from multiple chains** With the same setups as in Subsection 5.1, we do the experiments with other covariance matrices, including the identity $\begin{bmatrix} 1 & 0 \\ 0 & 1 \end{bmatrix}$ and $\begin{bmatrix} 1 & -0.8 \\ -0.8 & 2 \end{bmatrix}$. Figures 6 and 7 show the histograms of DC-LA, PSGLA, Moreau ULA and ULA. Figures 8 and 9 show the binned KL divergence between the histograms of samples generated by each sampling algorithm and the binned target distribution. In these cases, DC-LA achieves the lowest binned KL divergence.

**Histograms of samples from a single chain** We also report histograms based on samples obtained from a single Markov chain of length 10,000, discarding the first 500 samples as burn-in. Although our theoretical results apply to multiple chains using the final sample from each, it is common practice to run a single long chain and obtain samples directly from it. Figures 10 and 11 show the histograms of single chains of four samplers. As expected, the fidelity is lower than when running multiple chains. Still, DC-LA achieves a good compromise between ULA/Moreau ULA and PSGLA, although all algorithms struggle to efficiently explore every mode.

**Ablation experiment on $(\lambda, \gamma)$** We study the sensitivity of the performance of DC-LA with respect to the smoothing parameter $\lambda$ and the step size $\gamma$. To this end, we sweep $\lambda \in \{10^{-4}, 3 \times 10^{-4}, 10^{-3}, 3 \times 10^{-3}, 10^{-2}, 3 \times 10^{-2}\}$ and $\gamma \in \{10^{-4}, 3 \times 10^{-4}, 10^{-3}, 3 \times 10^{-3}, 10^{-2}\}$. For each grid point $(\lambda, \gamma)$, we run 3000 DC-LA chains of length 1000 and keep the last sample of each chain. Figure 12 shows the performance of DC-LA when $(\lambda, \gamma)$ varies. We observe a broad region of stable performance for intermediate values of both parameters. The consistency across all 12 subplots suggests that while the absolute error scale changes (the color bar ranges), the optimal "sweet spot" for hyperparameters remains relatively stable, indicating that DC-LA is robust and does not require drastic retuning when the underlying data distribution (mean or correlation) shifts. In this experiment, we see that setting $\gamma$ on the order of $\lambda$ (but a factor smaller) provides a favorable balance, yielding stable behavior while still allowing the chain to mix quickly.

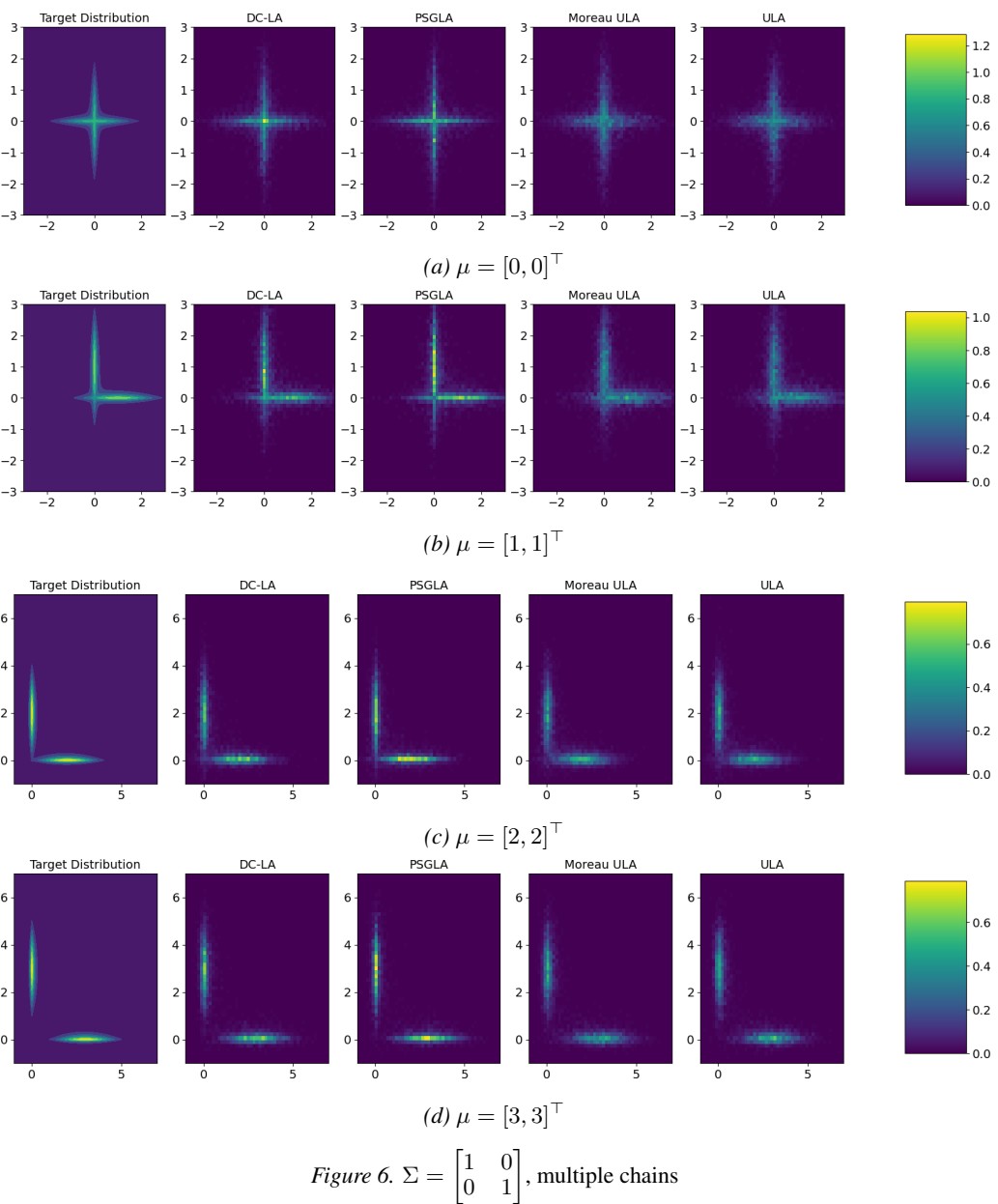

*(a)* $\mu = [0, 0]^\top$

*(b)* $\mu = [1, 1]^\top$

*(c)* $\mu = [2, 2]^\top$

*(d)* $\mu = [3, 3]^\top$

*Figure 6.* $\Sigma = \begin{bmatrix} 1 & 0 \\ 0 & 1 \end{bmatrix}$, multiple chains

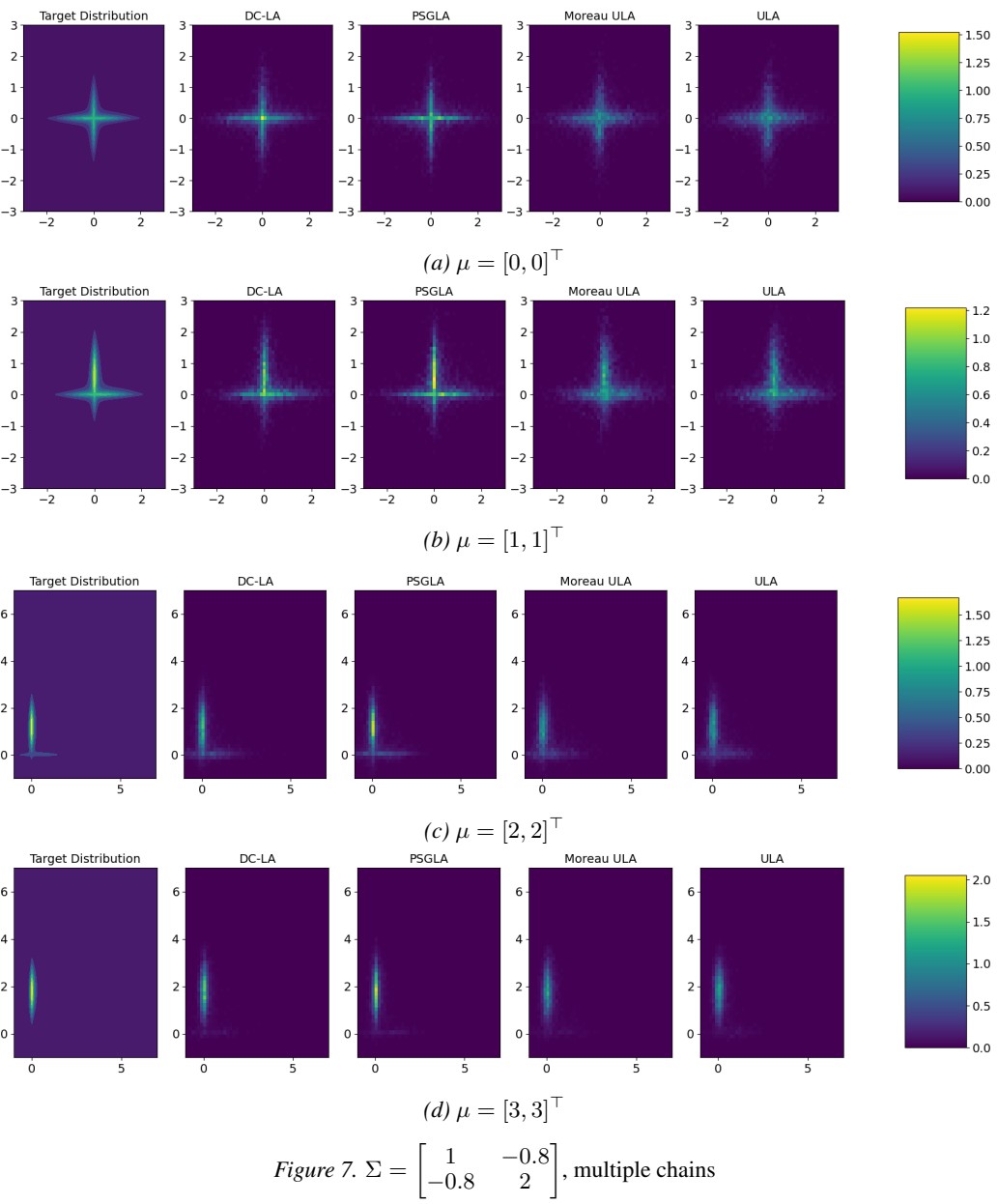

(a) $\mu = [0,0]^\top$

(b) $\mu = [1,1]^\top$

(c) $\mu = [2,2]^\top$

(d) $\mu = [3,3]^\top$

*Figure 7.* $\Sigma = \begin{bmatrix} 1 & -0.8 \\ -0.8 & 2 \end{bmatrix}$, multiple chains

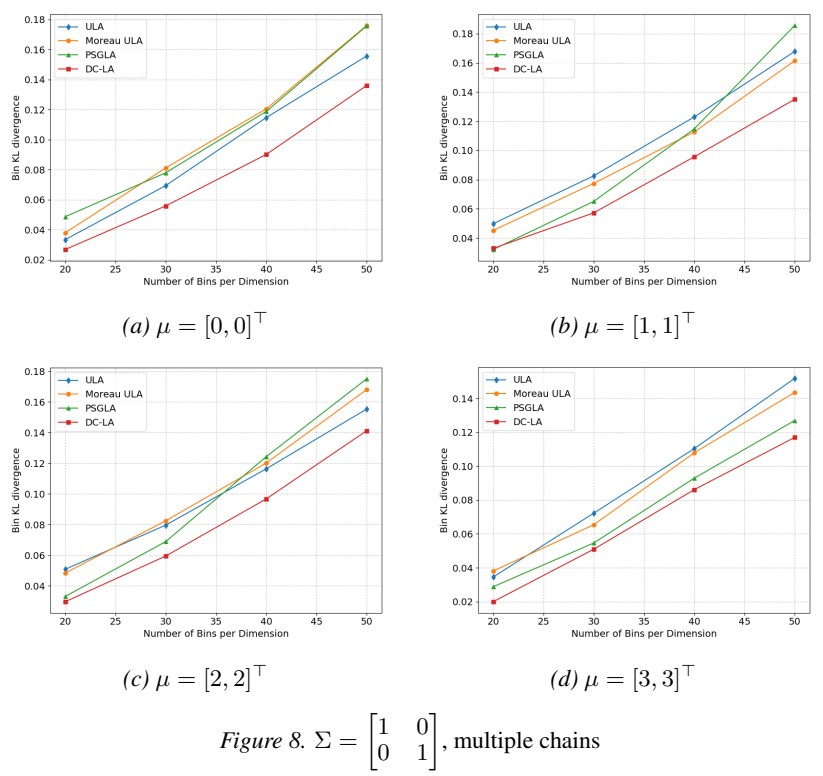

*(a)* $\mu = [0,0]^\top$

*(b)* $\mu = [1,1]^\top$

*(c)* $\mu = [2,2]^\top$

*(d)* $\mu = [3,3]^\top$

*Figure 8.* $\Sigma = \begin{bmatrix} 1 & 0 \\ 0 & 1 \end{bmatrix}$, multiple chains

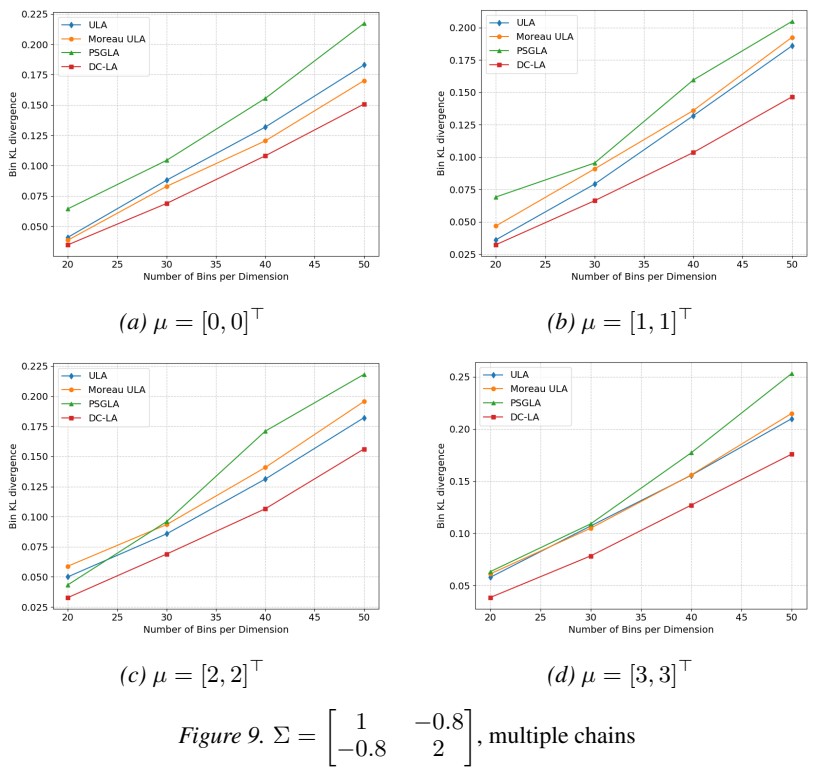

*(a)* $\mu = [0,0]^\top$

*(b)* $\mu = [1,1]^\top$

*(c)* $\mu = [2,2]^\top$

*(d)* $\mu = [3,3]^\top$

*Figure 9.* $\Sigma = \begin{bmatrix} 1 & -0.8 \\ -0.8 & 2 \end{bmatrix}$, multiple chains

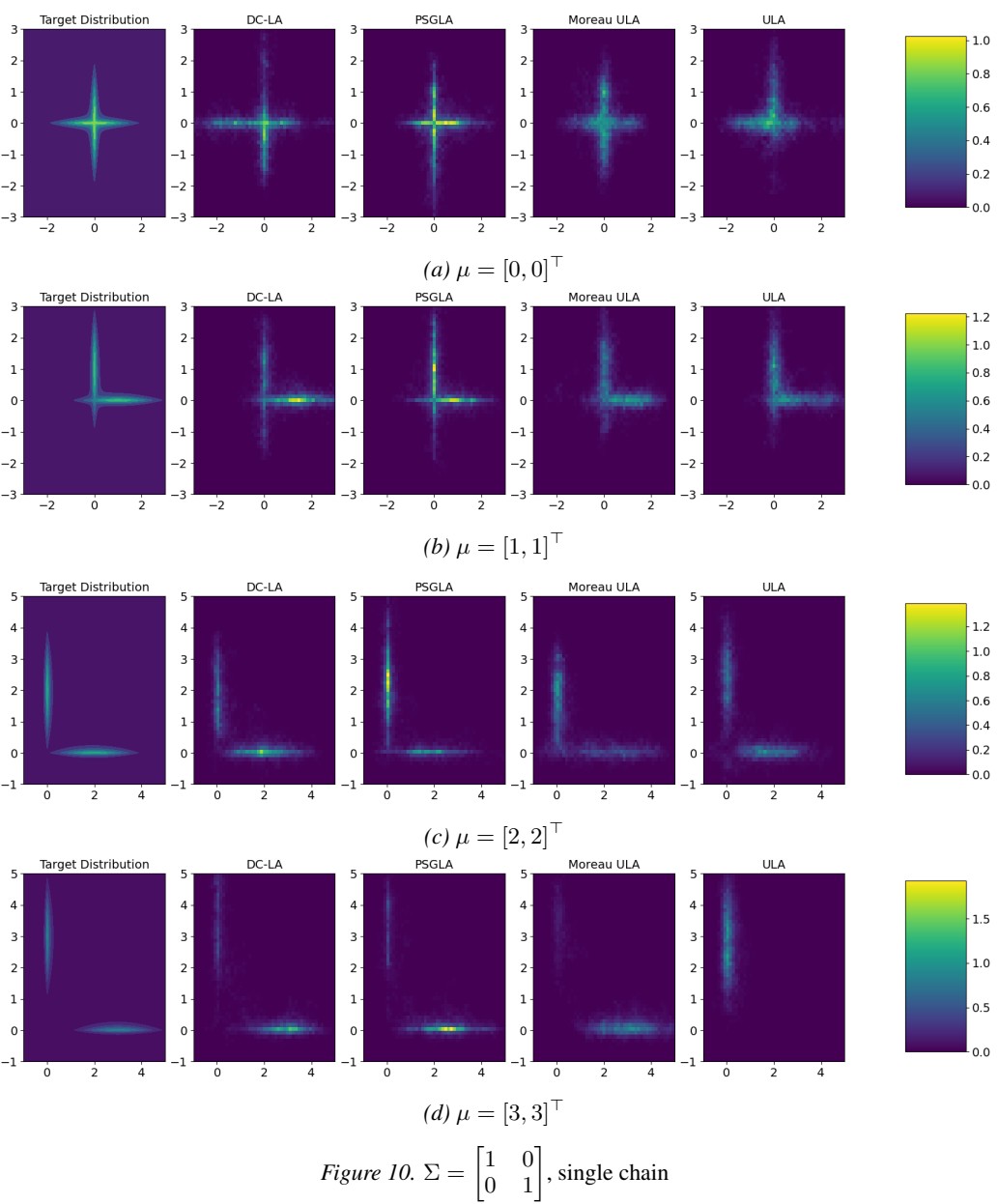

*(a)* $\mu = [0,0]^\top$

*(b)* $\mu = [1,1]^\top$

*(c)* $\mu = [2,2]^\top$

*(d)* $\mu = [3,3]^\top$

*Figure 10.* $\Sigma = \begin{bmatrix} 1 & 0 \\ 0 & 1 \end{bmatrix}$, single chain

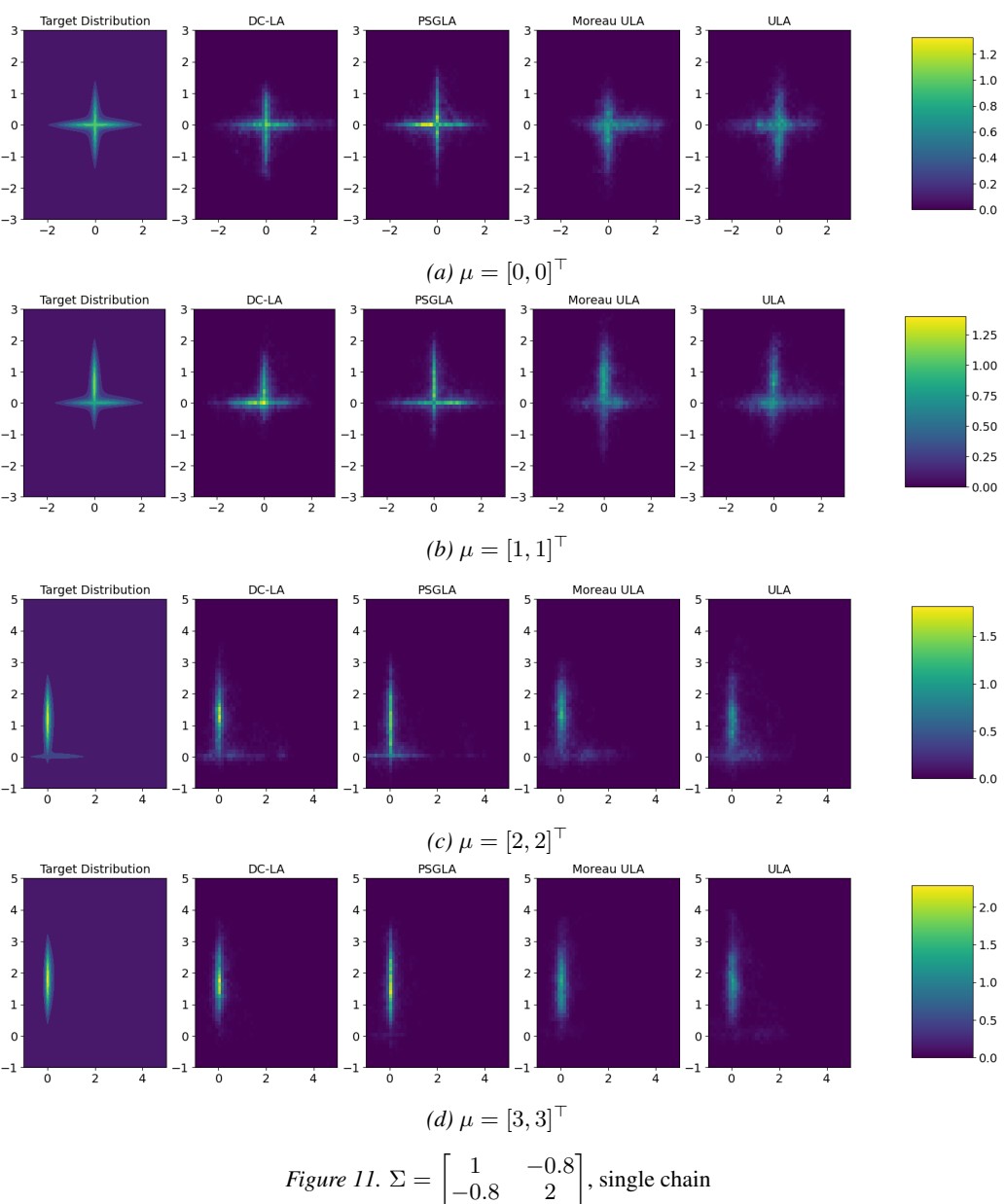

*(a)* $\mu = [0, 0]^\top$

*(b)* $\mu = [1, 1]^\top$

*(c)* $\mu = [2, 2]^\top$

*(d)* $\mu = [3, 3]^\top$

*Figure 11.* $\Sigma = \begin{bmatrix} 1 & -0.8 \\ -0.8 & 2 \end{bmatrix}$, single chain

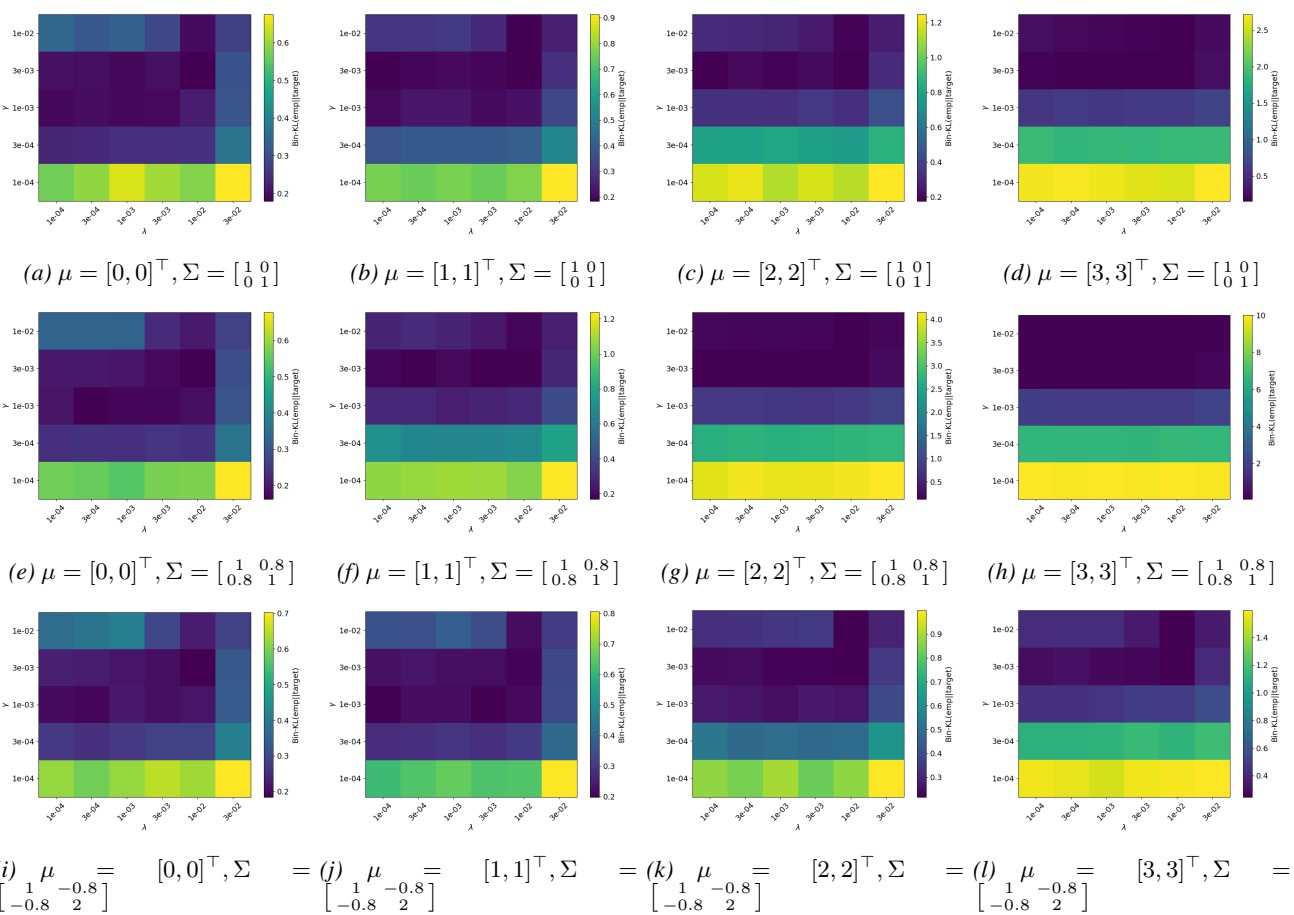

*(a)* $\mu = [0,0]^\top, \Sigma = \left[\begin{smallmatrix} 1 & 0 \\ 0 & 1 \end{smallmatrix}\right]$    *(b)* $\mu = [1,1]^\top, \Sigma = \left[\begin{smallmatrix} 1 & 0 \\ 0 & 1 \end{smallmatrix}\right]$    *(c)* $\mu = [2,2]^\top, \Sigma = \left[\begin{smallmatrix} 1 & 0 \\ 0 & 1 \end{smallmatrix}\right]$    *(d)* $\mu = [3,3]^\top, \Sigma = \left[\begin{smallmatrix} 1 & 0 \\ 0 & 1 \end{smallmatrix}\right]$

*(e)* $\mu = [0,0]^\top, \Sigma = \left[\begin{smallmatrix} 1 & 0.8 \\ 0.8 & 1 \end{smallmatrix}\right]$    *(f)* $\mu = [1,1]^\top, \Sigma = \left[\begin{smallmatrix} 1 & 0.8 \\ 0.8 & 1 \end{smallmatrix}\right]$    *(g)* $\mu = [2,2]^\top, \Sigma = \left[\begin{smallmatrix} 1 & 0.8 \\ 0.8 & 1 \end{smallmatrix}\right]$    *(h)* $\mu = [3,3]^\top, \Sigma = \left[\begin{smallmatrix} 1 & 0.8 \\ 0.8 & 1 \end{smallmatrix}\right]$

*(i)* $\mu = [0,0]^\top, \Sigma = \left[\begin{smallmatrix} 1 & -0.8 \\ -0.8 & 2 \end{smallmatrix}\right]$   *(j)* $\mu = [1,1]^\top, \Sigma = \left[\begin{smallmatrix} 1 & -0.8 \\ -0.8 & 2 \end{smallmatrix}\right]$   *(k)* $\mu = [2,2]^\top, \Sigma = \left[\begin{smallmatrix} 1 & -0.8 \\ -0.8 & 2 \end{smallmatrix}\right]$   *(l)* $\mu = [3,3]^\top, \Sigma = \left[\begin{smallmatrix} 1 & -0.8 \\ -0.8 & 2 \end{smallmatrix}\right]$

*Figure 12.* Ablation experiment analyzing the sensitivity of DC-LA performance to the hyperparameters $(\lambda, \gamma)$

## J.3. Computed Tomography with DCINNs prior

We show in Figures 13, 14, 15, and 16 some other results with the same settings as in Section 5.2.

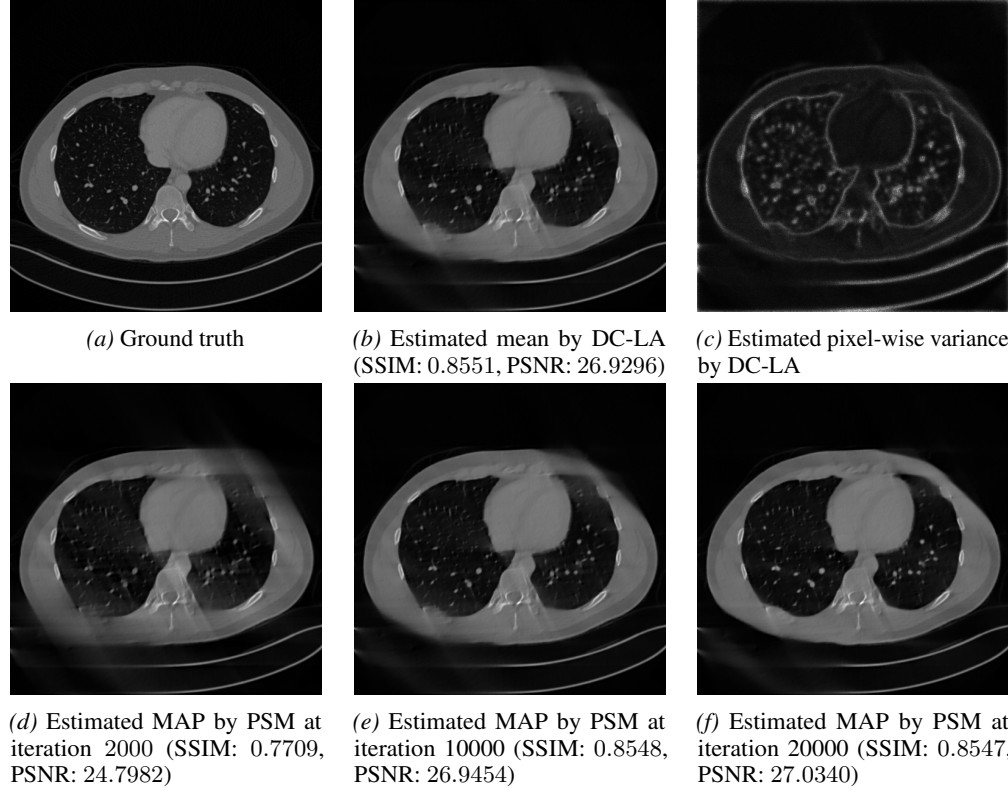

*(a)* Ground truth

*(b)* Estimated mean by DC-LA (SSIM: 0.8551, PSNR: 26.9296)

*(c)* Estimated pixel-wise variance by DC-LA

*(d)* Estimated MAP by PSM at iteration 2000 (SSIM: 0.7709, PSNR: 24.7982)

*(e)* Estimated MAP by PSM at iteration 10000 (SSIM: 0.8548, PSNR: 26.9454)

*(f)* Estimated MAP by PSM at iteration 20000 (SSIM: 0.8547, PSNR: 27.0340)

*Figure 13.* Scan: L333_FD_1_1.CT.0002.0010.2015.12.22.20.18.21.515343.358517203

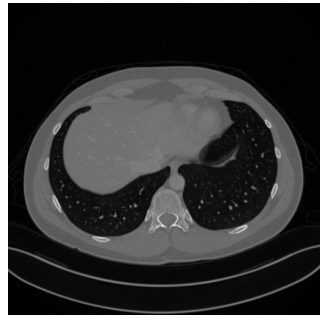
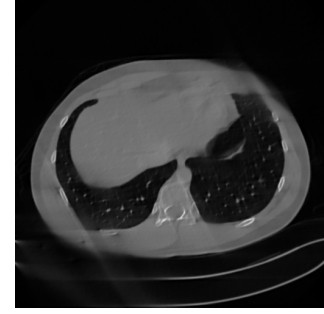
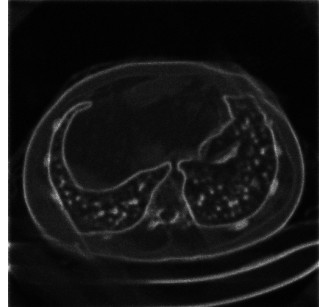

*(a)* Ground truth

*(b)* Estimated mean by DC-LA (SSIM: 0.8702, PSNR: 27.4151)

*(c)* Estimated pixel-wise variance by DC-LA

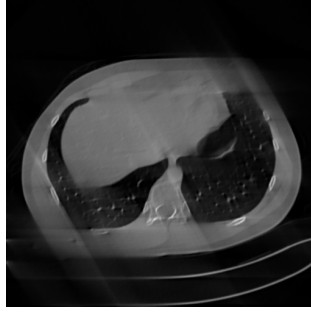
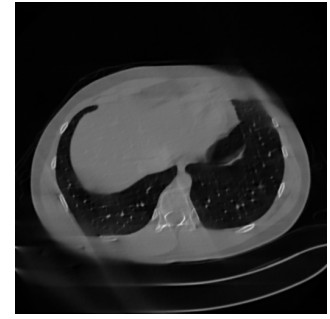
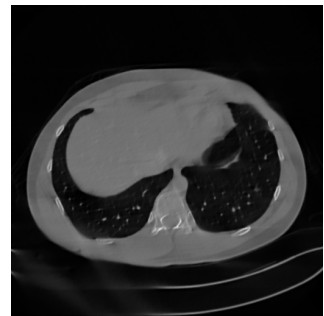

*(d)* Estimated MAP by PSM at iteration 2000 (SSIM: 0.7972, PSNR: 25.1140)

*(e)* Estimated MAP by PSM at iteration 10000 (SSIM: 0.8702, PSNR: 27.4418)

*(f)* Estimated MAP by PSM at iteration 20000 (SSIM: 0.8712, PSNR: 27.7555)

*Figure 14.* Scan: L333_FD_1_1.CT.0002.0050.2015.12.22.20.18.21.515343.358518163

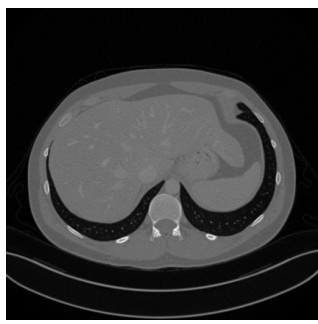
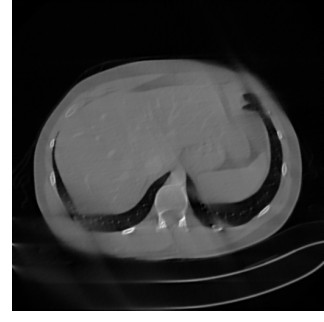
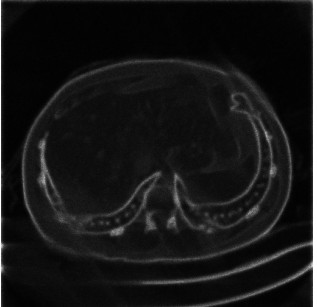

*(a)* Ground truth

*(b)* Estimated mean by DC-LA (SSIM: 0.8666 , PSNR: 26.9199)

*(c)* Estimated pixel-wise variance by DC-LA

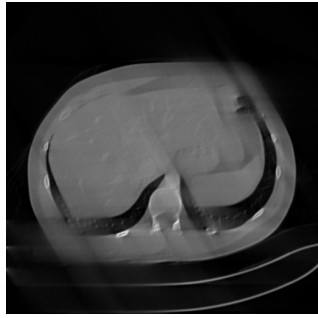
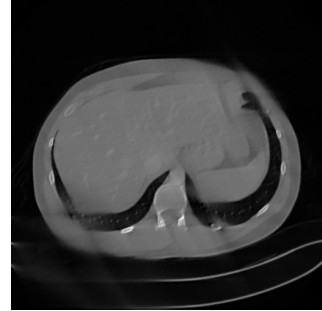
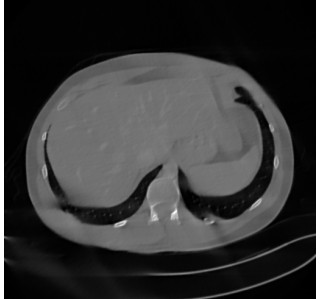

*(d)* Estimated MAP by PSM at iteration 2000 (SSIM: 0.7997, PSNR: 25.1084)

*(e)* Estimated MAP by PSM at iteration 10000 (SSIM: 0.8664, PSNR: 26.9521)

*(f)* Estimated MAP by PSM at iteration 20000 (SSIM: 0.8641, PSNR: 27.2395)

*Figure 15.* Scan: L333_FD_1_1.CT.0002.0080.2015.12.22.20.18.21.515343.358518883

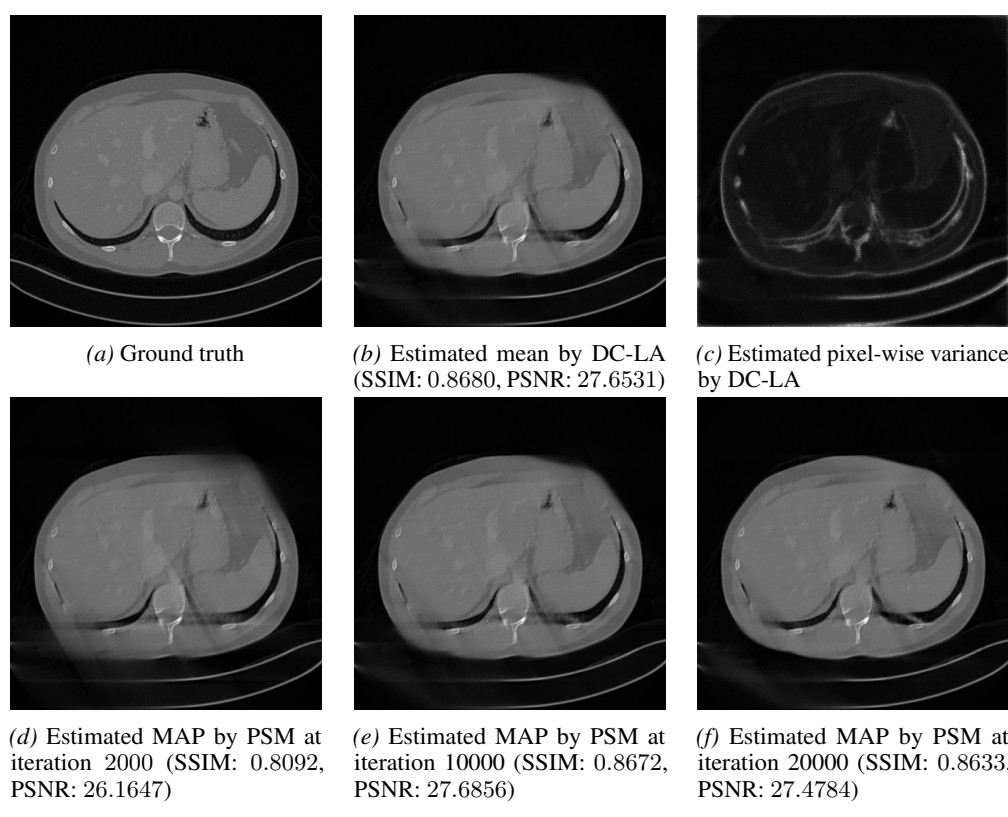

*(a)* Ground truth

*(b)* Estimated mean by DC-LA (SSIM: 0.8680, PSNR: 27.6531)

*(c)* Estimated pixel-wise variance by DC-LA

*(d)* Estimated MAP by PSM at iteration 2000 (SSIM: 0.8092, PSNR: 26.1647)

*(e)* Estimated MAP by PSM at iteration 10000 (SSIM: 0.8672, PSNR: 27.6856)

*(f)* Estimated MAP by PSM at iteration 20000 (SSIM: 0.8633, PSNR: 27.4784)

*Figure 16.* Scan: L333_FD_1_1.CT.0002.0100.2015.12.22.20.18.21.515343.358519363

