# OpenReview forum: "DC-LA: Difference-of-Convex Langevin Algorithm"
_ICML.cc/2026/Conference — ICML 2026 regular_

### Official Review · Reviewer_62PB · 2026-02-22

**Soundness:** 2
**Presentation:** 3
**Significance:** 3
**Originality:** 3
**Overall Recommendation:** 4
**Confidence:** 5

**Summary:**

This paper proposes a new method for sampling from Gibbs distributions of the form $\pi(x) \propto e^{-V(x)}$ with possibly non-convex and non-differentiable potentials $V$ of the form $V = f+r_1-r_2$ with $f$ L-smooth, $r_1$, $r_2$ convex, $r_1$ Lipschitz and $r_2$ Lipschitz, Hölder smooth or as well L-smooth. The proposed algorithm performs a gradient step with respect to $f$ and the Moreau envelope $r^\lambda_2$ of $r_2$ and afterward a prox-step with on the Moreau envelope $r^\lambda_1$ of $r_1$.

The framework of a non-smooth part of the potential of such a \emph{difference of convex} form $r_1-r_2$ allows to treat even non-weakly convex potentials $V(x)$.

The authors a theoretical analysis of ergodicity and convergence of the methods, consider multiple cases for $r_i$ and show numerical experiments in a synthetic setting as well as for CT imaging.

**Compliance With Llm Reviewing Policy:**

Affirmed.

**Final Justification:**

Soundness: The reviewers have addressed a technical error I found in the proof of the main result, thus, the paper should now be correct to the best of my knowledge.

# Summary
The paper is technically correct and provides theoretical guarantees for Langevin sampling for difference-of-convex (DC) potentials, thus, making sampling amenable for a new class of problems, for which Langevin sampling had not been possible previously.

The main reasons why I only suggest "weak accept" is that in my opinion the algorithm was presented in a somewhat obscure way, in particular, as suggested in my review, the different handling of the two Moreau envelopes seems rather odd to me. Even-though everything is correct, I think similar results could have been achieved with better presentation. In particular, I would argue that the main contribution/idea is to assume the DC structure and apply Moreau envelopes to the two parts of the DC separately instead of the entire non-smooth part. This idea, which could have been communicated very easily, seemed to be hidden under the algorithm where one functional is double Moreau enveloped.

**Key Questions For Authors:**

1. In the Proof of thm 4.4, line 1375 the application of the referenced thm is not correct. The referenced result bounds the error between the stationary distribution of the Markov chain with step size $\gamma$ and the target density---which is independent of $\gamma$---that is reached for $\gamma=0$. This is not the correct case within the manuscript, since in the manuscript the target distribution is itself dependent on $\gamma$. That is, the target in (14) is explicitly dependent on the step size. In order to apply the referenced result, one should instead use two different $\gamma_i$, $i=1,2$, one denoting the step size of the algo and one denoting the second Moreau parameter in (14) and afterward argue wether one can choose $\gamma_1 = \gamma_2$ without losing any convergence results.

2. Also in relation to 1.: The derivation and presentation of DC-LA is somewhat peculiar. It seems very unnatural, to consider the prox of the Moreau envelope. Since a proximal step is equivalent to a gradient step on the Moreau envelope, the prox of the Moreau envelope is equal to the gradient step on the Moreau envelope of the Moreau envelope. The Moreau envelope of the Moreau envelope is, however, simply the Moreau envelope of the sum of the two Moreau parameters. Therefore, DC-LA is effectively performing gradient steps on the Moreau envelope with two properties: (i) One envelope is with $\gamma$ one with $\gamma + \lambda$. (ii) The second gradient step (i.e. the prox one in DC-LA) is applied after the other updates. All in all, I do not really see the advantage in taking the double Moreau envelope of $r_1$ and performing the $r_1$-update after the other updates. Performing all steps at the same time would moreover closely resemble the update in (12) which is anyway used for the convergence analysis and it would be used to solve the issue in 1.

**Limitations:**

yes

**Strengths And Weaknesses:**

Soundness:
While most parts of the paper are technically sound (for minor issues see below) I encountered a technical error that should be corrected. The details are in the questions part below. This error is mostly relevant for my rating.

Presentation:
The presentation of the paper is good. The authors clearly state the goal of extending Langevin sampling to cases where the potential contains a difference of convex functions and derive the Moreau based algorithm. A potential improvement could be to motivate/elaborate why a proximal step is used on the Moreau envelope of $r_1$ instead of a gradient step (see also questions below). I found this a little unusual and confusing as a prox on the Moreau envelope is the same as a gradient step on the Moreau envelope with larger Moreau parameter. So why not directly use a Moreau gradient step? This almost seems to make the algorithm more complicated for no particular reason.

Significance:
The paper is reasonably significant. It extends Langevin sampling to potentials which can be non-weakly convex and non-differentiable, a use-case for which to the best of my knowledge previously sampling using Langevin dynamics was not feasible.

Originality:
The paper is also original. Albeit most of the techniques have been investigated in the literature in other contexts (Langevin sampling, proximal Langevin, Moreau envelope approximation, difference of convex) and the proofs are based on a single source (Renaud, 2025) to a large extend, I think the combination of techniques and application to the specific use-case is original. An additional contribution is the extensive analysis/discussion of DC priors.

Some suggestions/errors regarding mostly smaller issues:
* line 36-37, left: replace "under" with "of"
* l 81, right: You do not show in the paper that uncertainty quantification is "reliable". You simply show posterior variance plots. Moreover, assuming that the sampler samples correctly the reliabilty of variance estimates is rather a matter of the quality of the prior. I would reformulate this.
* l 97, right: Langevin sampling for weakly convex (and non-differentiable!) potentials had been analyzed in a less general way already before (Renaud, 2025) in (Habring et al, Diffusion at Absolute Zero: Langevin Sampling Using Successive Moreau Envelopes, 2026, SIAM Imaging) (the latter work was published earlier on arxiv). Also subgradient and proximal schemes for non-smooth but convex potentials have been analyzed in (Habring et al., Subgradient Langevin Methods for Sampling from Nonsmooth Potentials). Maybe consider citing these works.
* l 158 left: Regarding Gibbs sampling, consider citing (Kuric et al., The Gaussian latent machine: Efficient prior and posterior sampling for inverse problems)
* l 152 right: "Monreau" should be "Moreau"
* l 378, 379: The sentence is gramatically incorrect.
* l 435-437: Please mention that error to the ground truth is not a meaningful metric to evaluate sampling accuracy. The error is heavily influenced by the quality of the prior as a model for imaging. An exact sampler used with a poor prior has to lead to poor PSNR.
* In the conclusion: I think the statement that uncertainty quantification is overlooked in CT imaging is not entirely correct. Approaches based on posterior sampling usually include variance estimates (cf. Zach et al., Computed Tomography Reconstruction Using Generative Energy-Based Priors). In (Narnhofer et al, Posterior-variance–based error quantification for inverse problems in imaging) the variance estimates are also rigorously linked to uncertainty.
* l 631-647: This is a very complicated argument. It would have been sufficient to argue that g is locally bounded (because convex) and $tx$ is bounded for $t<t_0$
* l 656: You need Lipschitzness of $r_1^\lambda$, thus, boundedness of $\nabla r_1^\lambda$, not of $\nabla (r_1^\lambda)^\gamma$, I think.
* l 705: Please state that the elements are ordered "in descending order" by magnitude.
* l 842: I belive this argument relies on convexity of $Prox(\mathbb{R}^d)$. In your paper $Prox(\mathbb{R}^d)=\mathbb{R}^d$, so it is not a problem, but I think this should be noted.
* l 895, 896: Please reformulate a little. I was confused about the statement that $\bar{u}$ is convex because I immediately thought it is not. Only later I understood that you meant it would be convex under the premiss of the proof by contradiction. So maybe reformulate to "it would follow that $\bar{u}$ would be convex" or something similar.
* line 992: remove the last "such that"
* line 995: Add maybe "by Taylor expansion"
* The figures, especially page 39, are not well-formatted (very small fonts).

---

> ### Author Rebuttal · Authors · 2026-03-30
>
> We kindly thank the reviewer for the careful reading, insightful observations on our proofs, and thoughtful and positive comments on our work.
>
> > 1. The application of the referenced thm is not correct.
>
> Thank you for pointing this out. We clarify below how Theorem 2 in [Renaud2025] can be rigorously applied in our setting. In brief, we adopt the reviewer’s recommended two-stepsize procedure and then show that these step sizes can be chosen to be equal, thereby recovering our original formulation. Consequently, the statement in our paper remains valid.
>
> **Theorem 2 [Renaud2025].**
> Let $\pi \propto e^{-V}$ and $b = \nabla V$. Suppose that $b$ is $L$-Lipschitz and $(m,R)$-distant dissipative. Then, for $\gamma \in (0,\frac{m}{L^2}]$, the Markov Chain $X_{k+1} = X_k - \gamma b(X_k) + \sqrt{2\gamma}Z_{k+1}$ has an invariant law $\pi_\gamma$ and for $q \in \mathbb{N}^\star$, there exists
> $D_q\geq 0$ such that $\forall \gamma \in (0, \gamma_0]$, we have $W_q(\pi_\gamma, \pi) \leq D_q \gamma^{\frac{1}{2q}}$.
>
> **Remark.** The constant $D_q$ is **independent** of $\gamma$ and only depends on the drift $b$ via its three parameters: $L,m, R$.
>
>
> Now, we recall the Markov chain (13) in the manuscript **(Here we drop the \bar notation for display, as OpenReview markdown struggles with equations with \bar; should not be confusing to the chain (11) in our manuscript.)**
>
> $Y_{k+1} = Y_k - \gamma b_{\lambda}^{\gamma}(Y_k) + \sqrt{2\gamma} Z_{k+1} (*)$
>
> where $b_{\lambda}^{\gamma}(y):= \nabla r_1^{\lambda}(prox_{\gamma r_1^{\lambda}}(y)) + \nabla f(y)-\nabla r_2^{\lambda}(y)$. We recall $\pi_{\lambda,\gamma}(y) \propto \exp\left(-f(y)-(r_1^{\lambda})^{\gamma}(y) + r_2^{\lambda}(y) \right): = \exp(-V_{\lambda}^{\gamma}(y))$.
>
> Note $\nabla V_{\lambda}^{\gamma} = b_{\lambda}^{\gamma}$ and: for all $\gamma>0$, $b_{\lambda}^{\gamma}$ is $L$-Lipschitz and $(m,R)$-distant dissipative where $L,m,R$ do **not** depend on $\gamma \in (0,\gamma_0]$ ($\gamma_0$ is fixed), see lines 265-274, right column.
>
> Now fix $\gamma \in (0,m/L^2]$ and consider the Markov chain: $Y_{k+1} = Y_k - \epsilon b_{\lambda}^{\gamma}(Y_k) + \sqrt{2\epsilon} Z_{k+1} (**)$.
>
> Since $b_{\lambda}^{\gamma}$ is $L$-Lipschitz and $(m,R)$-distant dissipative, applying the above Theorem 2, for any $\epsilon \in (0,m/L^2]$, the chain (**) has an invariant distribution $p_{\lambda,\gamma,\epsilon}$ and there exists $D_q$ that is independent to both $\epsilon$ and $\gamma$ (since $D_q$ only depends on $b_{\lambda}^{\gamma}$ via its parameters $L,m,R$ and these parameters are independent to $\gamma$) such that: $W_q(p_{\lambda,\gamma,\epsilon},\pi_{\lambda,\gamma}) \leq D_q \epsilon^{\frac{1}{2q}}$. As this inequality holds for all $\epsilon \in (0,m/L^2]$, we can pick $\epsilon = \gamma$, leading to $W_q(p_{\lambda,\gamma,\gamma},\pi_{\lambda,\gamma}) \leq D_q \gamma^{\frac{1}{2q}}$. We simply denote $p_{\lambda,\gamma,\gamma}$ as $p_{\lambda,\gamma}$.
>
> As (**) is just (*) when $\epsilon=\gamma$, proof finished.
>
> > 2.  presentation of DC-LA
>
> We agree that the algebraic form of DC-LA may resemble two gradient steps (on Moreau envelopes); However, these steps serve different roles: exploration and dampening. See next.
>
> > advantage in taking the double Moreau envelope of $r_1$
>
> The first Moreau envelop of level $\lambda$ is crucial for our analysis. Without it, the Lipschitz constant of the drift $b_{\lambda}^{\gamma}$ in Lemma 4.2 can be as bad as $O(1/\gamma)$, e.g., take $r_1 = \Vert \cdot \Vert_1$. As a consequence, stability results in [Renaud2025] cannot be applied because they require a uniform upper bound of the Lipschitz constant of $b_{\lambda}^{\gamma}$ for all small $\gamma$. The second Moreau envelope of level $\gamma$ appears from the backward step by reformulation: $X_{k+1} = Y_{k+1} - \gamma \nabla r_1^{\lambda + \gamma}(Y_{k+1})$. This step is not just a gradient step, it is a contraction. First, the Moreau level $\lambda+\gamma$ is bigger than the stepsize $\gamma$, so this step is non-expansive by design (built-in stability when gamma is not small enough). Second, it decouples the roles of $\lambda$ and $\gamma$. Treating $\lambda+\gamma$ as a single parameter is possible, but imposes additional constraints and complicates the analysis.
>
> > Performing all steps at the same time
>
> If performing all (forward) steps of the same time, the scheme becomes the ULA applied to the smoothed target (we call it *Moreau-ULA* in our manuscript): $X_{k+1} = X_k-\gamma \nabla f(X_k) + \gamma \nabla r_2^{\lambda}(X_k) -\gamma \nabla r_1^{\lambda}(X_k) +\sqrt{2\gamma}Z_{k+1}$.
>
> **Order matters:** DC-LA's backward stabilizes the noisy $Y_{k+1}$, while Moreau-ULA has no noise absrobing machenism.
>
> Empirically (Sect. 5.1), Moreau-ULA produces blurry histograms, while DC-LA exhibits a clear stabilization effect.
>
> Other suggestions and smaller issues: thank you, we agree with those and will revise the manuscript accordingly with additional discussions on the mentioned works.

---

> > ### Author Rebuttal · Reviewer_62PB · 2026-04-01
> >
> > I updated my score based on the response and, in particular, the corrected proof.
> >
> > 1. Thank you for revising the proof. It looks fine to me now.
> >
> > > We agree that the algebraic form of DC-LA may resemble two gradient steps (on Moreau envelopes);
> >
> > To be clear, it does not "resemble", it "is".
> >
> > > 2. presentation of DC-LA
> >
> > I think there are some misunderstandings here.
> > * I agree that the first Moreau parameter is necessary to ensure Lipschitzness as the Lipschitz constant scales potentially with $1/\lambda$ (not $1/\gamma$ I think. I assume this was a typo by the authors). However, this does not lead to a necessity of applying yet another Moreau envelope. It merely implies that one should not use the Moreau parameter $\lambda$ as the step size otherwise, when the step size goes to zero, also the Lipschitz constant blows up. But this means all the results would still work if one fixes $\lambda$ of the first Moreau envelope and afterward chooses the step size $\gamma$ sufficiently small (leading to the Moreau-ULA)
> > * What I meant with "performing all steps at the same time" is the following update
> > \begin{equation}
> > (1)\quad X_{k+1} = X_k - \gamma \nabla r_1^{\lambda + \gamma}(X_k) - \gamma\nabla f(X_k) + \gamma \nabla r_2^{\lambda}(X_k) + \sqrt{2\gamma} Z_k
> > \end{equation}
> > which is exactly your sequence $\overline{Y}\_k$ and which is equivalent to
> > \begin{equation}
> > (2) \quad X\_{k+1} = \mathrm{prox}\_{\gamma r_1^\lambda}(X_k)- \nabla f(X_k) + \gamma \nabla r_2^{\lambda}(X_k) + \sqrt{2\gamma} Z_k.
> > \end{equation}
> > Note that your form of DC-LA is exactly equivalent to
> > \begin{equation}
> > (3) \quad
> > \begin{cases}
> > Y_{k+1} &= X_k - \gamma\nabla f(X_k) + \gamma \nabla r_2^{\lambda}(X_k) + \sqrt{2\gamma} Z_k.\newline
> > X_{k+1} &= Y_{k+1}- \gamma \nabla r_1^{\lambda + \gamma}(Y_{k+1}).
> > \end{cases}
> > \end{equation}
> > But I think I am ok with the exposition. I just think the proofs would have been simpler and maybe the presentation clearer with the above one.
> > Regarding the comment that order matters. I think this should be interpreted carefully and the authors should not attribute this effect incorrectly to the very specific double Moreau envelope update rule. Every method which adds Gaussian noise in the last step of the update rule will lead to blurry/noisy samples as the samples are distributed as a convolution with a Gaussian. The sharp samples could probably also be obtained from Moreau ULA if one would write it as
> > \begin{equation}
> > (4) \quad
> > \begin{cases}
> > \quad Y_{k+1} &= X_k - \gamma \nabla r_1^{\lambda}(X_k) - \gamma\nabla f(X_k) + \gamma \nabla r_2^{\lambda}(X_k)\newline
> > X\_{k+1} &= Y_{k+1}+ \sqrt{2\gamma} Z_k
> > \end{cases}
> > \end{equation}
> > and use as samples $Y_{k}$ instead of $X_k$.

---

> > > ### Author Response · Authors · 2026-04-02
> > >
> > > We thank the reviewer for the increased score and for the prompt, thoughtful feedback.
> > >
> > > We agree with most of the points raised.
> > >
> > > > But I think I am ok with the exposition. I just think the proofs would have been simpler and maybe the presentation clearer with the above one.
> > >
> > > We are glad that the exposition is fine. To add to the discussion:
> > >
> > > It is right that your proposed equation (1) is equivalent to our sequence $\{\bar{Y}_k\}$ **(again, we drop \bar for displaying)**:
> > >
> > > $Y_{k+1} = prox_{\gamma r_1^{\lambda}}(Y_k) - \gamma \nabla f(Y_k)+\gamma \nabla r_2^{\lambda}(Y_k)+ \sqrt{2\gamma} Z_k.$
> > >
> > > In our work, $\{\bar{Y}_k\}$ was introduced to facilitate the analysis, but not the main sequence. As a byproduct of the proof, we could establish the convergence of $\{\bar{Y}_k\}$ as well, and therefore, this sequence is also a valid sampler with quantifiable behavior.
> > >
> > > In practice, the sequence $\bar{Y}_k$ itself should have a blurry histogram since the noise is added at the end. This relates to the last point of the reviewer: we can deconvolve the noise by considering another auxiliary sequence
> > >
> > > $U_k = prox_{\gamma r_1^{\lambda}}(Y_k) -  \gamma \nabla f(Y_k) + \gamma \nabla r_2^{\lambda}(Y_k)$,
> > >
> > > $Y_{k+1} = U_k + \sqrt{2\gamma} Z_k$.
> > >
> > > This helps with the noise part, but it is not enough. For example, consider the case where $r_2 = 0$, $r_1$ is close to an indicator of a set to impose a soft constraint, we would want our sequence of interest to land close to the constraint set, while the term $-\gamma \nabla f(\bar{Y}_k)$ in the formulation of $U_k$ is constraint-unaware (although we choose $\gamma$ to be small to reduce its effect, the term is still constraint-unaware by nature). Hence, the ideal auxiliary sequence in this case should be
> > >
> > > $X_k = prox_{\gamma r_1^{\lambda}}(\bar{Y}_k)$: it is both noise-deconvolved and constraint-aware.
> > >
> > > The scheme is then written as:
> > >
> > > $Y_{k+1} = X_k - \gamma \nabla f(Y_k) + \gamma \nabla r_2^{\lambda}(Y_k) + \sqrt{2\gamma} Z_k,$
> > >
> > > $X_{k+1} = prox_{\gamma r_1^{\lambda}}(Y_{k+1})$
> > >
> > >
> > > In comparison, DC-LA is given as:
> > >
> > > $Y_{k+1} = X_k - \gamma \nabla f(X_k) + \gamma \nabla r_2^{\lambda}(X_k) + \sqrt{2\gamma} Z_k$
> > >
> > > $X_{k+1} = prox_{\gamma r_1^{\lambda}}(Y_{k+1})$,
> > >
> > > The only difference is that DC-LA evaluates gradients at $X_k$, i.e., it does not introduce a lag in the update of $Y_{k+1}$. We view this distinction primarily as an algorithmic design choice.
> > >
> > > > I agree that the first Moreau parameter is necessary to ensure Lipschitzness...
> > >
> > > Yes, we mean that without the smoothing parameter $\lambda$, the Lipschitz constant would blow up with rate $1/\gamma$ as the step size goes to 0.
> > >
> > > The second envelope arises from the proximal backward step, with smoothing level $\gamma$, which counterbalances the stepsize $\gamma$ used in the forward step. Altogether, this leads to the term $\nabla r^{\lambda + \gamma}$ in the second update of DC-LA.
> > >
> > > If one were to use only $\nabla r^{\lambda}$ in the backward:
> > >
> > > $X_{k+1} = Y_{k+1} - \gamma \nabla r_1^{\lambda}(Y_{k+1})$
> > >
> > > By choosing $\gamma < \lambda$, the update remains non-expansive. In this case, it can be written as the proximal operator
> > >
> > > $X_{k+1} = prox_{\gamma r_1^{\lambda - \gamma}}(Y_{k+1})$.
> > >
> > > The convergence analysis is then expected to follow along similar lines. Therefore, it is plausible that this approach can be used, although additional assumptions may be required.
> > >
> > > In hindsight, it is not necessary to use the same smoothing parameter $\lambda$ for both $r_1$ and $r_2$. One could instead introduce distinct parameters $\lambda_i$ for each $r_i$, which may provide a useful connection to the reviewer’s proposed approach.
> > >
> > > We will include a remark based on the above discussion.

---

### Official Review · Reviewer_jhfD · 2026-03-10

**Soundness:** 3
**Presentation:** 3
**Significance:** 3
**Originality:** 2
**Overall Recommendation:** 4
**Confidence:** 4

**Summary:**

The paper proposes a proximal method for sampling from distributions with regularizer given by a difference-of-convex functions, without necessarily ensuring smoothness of the regularizer. The core idea is to first apply smoothing, before combining a proximal step with a standard Euler step. Guarantees are obtained in $W_q$ distances, and some experimental validation is also performed.

**Compliance With Llm Reviewing Policy:**

Affirmed.

**Final Justification:**

My assessment of this paper was broadly positive, although in my opinion it addressed a somewhat peripheral problem in sampling/optimization, and also did not significantly advance the theoretical techniques in the field. The rebuttal clarified some of these points, but did not ultimately change my assessment of the paper's impact. I appreciate the authors' time and effort, but I believe my current score fairly reflects the importance of the paper.

**Key Questions For Authors:**

Can some interpretable bounds be given for e.g., the quadratic case, or dissipative at infinity case, and some standard regularizers?

DC-LA requires extra gradient evaluations compared to ULA. Is this accounted for? What about comparisons against other benchmarks (such as with a midpoint or something)? Are these likely to change the story?

Is the choice of lambda likely to matter much in practice?

**Limitations:**

Yes

**Strengths And Weaknesses:**

**Strengths**

The proof is rigorous and the results match the rates expected in this regime.

The algorithm is simple and intuitive for users to implement.

The experiments conducted show a clear gain in performance compared to competitors. I also appreciate the CT experiment as it illustrates some bona fide applicability of this scheme.

**Weaknesses**

The paper spends some time justifying why difference-of-convex regularizers may be helpful, but the applications still seem somewhat niche. I do think the paper is trying its best here, however, and this is just a consequence of DC regularizers being a .

The analysis is not particularly unexpected; apart from the smoothing step, I believe the guarantees here are relatively routine for PGSLD, and analogous to those seen in the sampling literature (and analogous therefore to proximal methods in optimization as well).

The dependence on key parameters in the final guarantees is a bit difficult to parse, and some illustrative examples might be helpful to witness some of the dependencies (dimension, $\lambda$, etc.)

The rates seem to scale exponentially in the smoothing constant; see Appendix I.6. This is not ideal although it seems to be fundamental to this approach.

---

> ### Author Rebuttal · Authors · 2026-03-30
>
> We kindly thank the reviewer for the careful reading and the positive and thoughtful remarks on our work. We reply to the questions as follows.
>
> > The dependence on key parameters in the final guarantees is a bit difficult to parse
>
> and
>
> > Can some interpretable bounds be given for e.g., the quadratic case, or dissipative at infinity case, and some standard regularizers?
>
> Thank you for your comment. We partially address these dependencies as follows. First, the new parameter introduced in our work is the smoothing parameter $\lambda$ and we discussed the dependencies of $A_{\lambda}, B_{\lambda}, \rho_{\lambda}$ on $\lambda$ in Appendix I.6.
>
> The full depedence of $\rho_{\lambda}$ on other parameters is: $\log(\log^{-1}(\rho_{\lambda}^{-1}))$ behaves like $L R^2$ where $L$ is the Lipschitz smoothness constant and $R$ is the radius in the distant dissipativity condition of $\bar{b}_{\lambda}^{\gamma}$ (see equation (10) in [Bortoli2020]). Consider Assumption 4(i) for simplicity, these constants are given by (see Lemmas 4.2 and 4.3) $L = (2/\lambda + L_f)$ and $R = \max \\{R_0, (8 G_1 + 4 L_f \gamma_0 G_1 + 4 G_2)/\mu \\}$. We can see that the $1/\lambda$ term mentioned in Appendix I.6 comes from $L$. On the other hand, other paramters like $G_1,G_2$ enter the formula via $R$.
>
> These parameters may or may not depend on the dimension $d$. For example, for $\ell_1 - \ell_{\sigma_q}$ prior, $G_1 = G_2 = \sqrt{d}$, while with $\ell_1-\ell_2$ prior, $G_1 = \sqrt{d}, G_2 = 1$. For quadratic likelihood $f(x) = (1/2)\Vert Ax-b\Vert^2$, $L_f = \Vert A^{\top} A \Vert_{op}$. If $A$ is normalized, $L_f = O(1)$, if not normalized, it can scale with $d$. We also note that $R_0$ can also potentially scale with the dimension. To see this, consider the separable potential of the form $U(x) = U_1(x_1) + U_2(x_2) + \ldots + U_d(x_d)$ where $x=[x_1,x_2,\ldots,x_d]^{\top}$. Assume that each $U_i$ is $(m,r)$-distant dissipative in 1D, then -- in the worst-case -- $U$ is $(m,r \sqrt{d})$-distant dissipative.
>
> > DC-LA requires extra gradient evaluations compared to ULA. Is this accounted for?
>
> We do not expect DC-LA to incur a higher per-iteration oracle cost than ULA. In particular, if a proximal evaluation is viewed as comparable in cost to a gradient evaluation (at least for elementary convex functions $r_1$ and $r_2$), then both methods require a similar number of first-order oracle calls per iteration.
>
> For ULA, each iteration involves evaluating $\nabla f$, $\partial r_1$ and $\partial r_2$. Each iteration of DC-LA (see the fully unrolled form in Appendix C) requires evaluating $\nabla f$, $prox_{\lambda r_2}$, and $prox_{(\lambda + \gamma) r_1}$. Therefore, the per-iteration computational cost is comparable, with DC-LA replacing subgradient evaluations by proximal operations. On another hand, if we consider the ULA applied to the smoothed target (we called it Moreau-ULA in Section 5.1), the oracle cost is identical: Moreau-ULA requires $\nabla f, prox_{\lambda r_1}, prox_{\lambda r_2}$ per iteration.
>
> > What about comparisons against other benchmarks (such as with a midpoint or something)? Are these likely to change the story?
>
> We believe the overall conclusions would remain the same. Our work primarily addresses the challenge of sampling from nonsmooth log-DC target distributions with theoretical guarantees, rather than improving discretization accuracy for smooth problems. While midpoint or higher-order schemes may reduce discretization error in smooth settings, they do not directly handle the nonsmooth DC structure considered in our setting.
>
> Moreover, such schemes typically assume smoothness and rely on higher-order derivatives, whereas DC-LA is specifically designed to exploit the forward--backward structure and proximal steps to manage nonsmooth components.
>
> > Is the choice of lambda likely to matter much in practice?
>
> Yes, $\lambda$ does matter, but DC-LA is not overly sensitive within a reasonable range of $\lambda$. In our ablation experiments (Appendix J.2), across different target distributions, we observe a broad range of $\lambda$ values for which DC-LA performs well, with no significant degradation when $\lambda$ varies within this region.
>
> **References**
>
> [Bortoli2020] De Bortoli, Valentin, and Alain Durmus. "Convergence of diffusions and their discretizations: from continuous to discrete processes and back." arXiv preprint arXiv:1904.09808 (2019).

---

> > ### Author Rebuttal · Reviewer_jhfD · 2026-04-04
> >
> > I thank the authors for responding to my concerns. I think this problem is interesting but, as I said in my original review, probably not groundbreaking in the context of the field of sampling/stochastic optimization. I opt to maintain my score.

---

> > > ### Author Response · Authors · 2026-04-07
> > >
> > > Thank you once again for your careful review and for engaging with our responses. We are glad that the concerns have been adequately addressed.
> > >
> > > We also hope to highlight the potential impact of our work. Our framework enables practical Langevin-type sampling with DC priors (possibly **non-weakly-convex**). We note that DC priors are important and have been widely used throughout the literature (e.g., [1–5]), demonstrating clear advantages (expressivity, tractability) over other types of priors; However, a principled sampler is still lacking (note that the PGSLA sampler and other Langevin-type samplers cannot address this problem since the potential can be non-weakly-convex). Our work addresses this need. We therefore expect some downstream implications of our scheme in this area.
> > >
> > > [1] Zhang, T. (2010). Analysis of multi-stage convex relaxation for sparse regularization. Journal of Machine Learning Research, 11(3).
> > >
> > > [2] Yin, P., Lou, Y., He, Q., & Xin, J. (2015). Minimization of $\ell_{1-2}$ for compressed sensing. SIAM Journal on Scientific Computing, 37(1), A536-A563.
> > >
> > > [3] Lou, Y., Zeng, T., Osher, S., & Xin, J. (2015). A weighted difference of anisotropic and isotropic total variation model for image processing. SIAM Journal on Imaging Sciences, 8(3), 1798-1823.
> > >
> > > [4] Zhang, S., & Xin, J. (2018). Minimization of transformed $\ell_1$ penalty: theory, difference of convex function algorithm, and robust application in compressed sensing. Mathematical Programming, 169(1), 307-336.
> > >
> > > [5] Zhang, Y., & Leong, O. (2025). Learning difference-of-convex regularizers for inverse problems: A flexible framework with theoretical guarantees. arXiv preprint arXiv:2502.00240.

---

### Official Review · Reviewer_iCKg · 2026-03-12

**Soundness:** 3
**Presentation:** 3
**Significance:** 3
**Originality:** 2
**Overall Recommendation:** 4
**Confidence:** 3

**Summary:**

This submission concerns the problem of sampling from a target distribution of the form $\pi\propto \exp(-V)$ for $V=f+r$, where $f$ has $L$-Lipschitz continuous gradient and $r$ can be expressed as the difference of two real-valued convex functions $r_1,r_2$. This choice of $V$ fits within the framework of Bayesian inference where $r$ is treated as a regularizer coming from a prior. In order to sample from $\pi$, it is proposed to modify the standard unadjusted Langevin algorithm (ULA), $X_{k+1}=X_k-\gamma\nabla V(X_k)+\sqrt{2\gamma}Z_{k+1}$ for differentiable potentials to the following update,  $X_{k+1}=\mathrm{prox}\_{\gamma r_1^{\lambda}}(X_k-\gamma\nabla f(X_k)+\gamma \nabla r_2^{\lambda}(X_k)+\sqrt{2\gamma}Z_{k+1})$, where $r_1^{\lambda}$ and $r_2^{\lambda}$ are the Moreau envelopes of $r_1$ and $r_2$ and $\mathrm{prox}$ is the proximal operator. This decomposition is inspired by previous literature on the problem of minimizing differences of convex functions. It is shown using the properties of the Moreau envelope that $X_{k+1}$ can be computed by evaluating $\nabla f$ and the proximal operators of $r_1$ and $r_2$.


The convergence of this scheme is then analyzed under the assumptions that (1) $V$ is distant dissipative, (2) $r_1$ is $G_1$-Lipschitz continuous, and (3) $r_2$  is either $G_2$-Lipschitz continuous or it is differentiable with H\"older continuous derivative. It is noted that, under assumptions (2) and (3), assumption (1) holds if and only if $f$ itself is distant dissipative. With this, it is shown that the iterations of this method prior and post prox correspond to general ULA updates with particular drift functions which are Lipschitz continuous and distant dissipative. With this, the results of [1] can be applied to obtain rates of convergence for the distribution of the iterates to the distribution of the true sampling distribution. If $r_2$ is smooth, a simplified iteration is also proposed and an analogous convergence result is proved.

The paper concludes with some numerical experiments to study the performance of the proposed approach. First, a synthetic experiment is performed with the $\ell_1-\ell_2$ prior. It is shown that DC-LA outperforms other methods which do not account for the difference of convex structure in the prior. Next, an experiment on real life tomography with priors based on differences of input convex neural networks is performed. In this context, the proposed method performs reasonably compared to a maximum a posteriori estimate based on direct optimization.

1. Renaud, M., De Bortoli, V., Leclaire, A., & Papadakis, N. (2025). From stability of Langevin diffusion to convergence of proximal MCMC for non-log-concave sampling. NeurIPS 2025.

**Compliance With Llm Reviewing Policy:**

Affirmed.

**Final Justification:**

The rebuttal adequately addressed the questions that I had.
I have maintained my initial score, as while the submission does include some novel ideas and proof techniques, I agree with the other reviewers that the work is a bit incremental in nature; this is addressed, partially, by a more clear discussion of what regularizers can be analyzed with these new results.
Beyond this, I find the paper to be clearly and soundly written.

**Key Questions For Authors:**

My main question for the authors regards the technical contributions of the work. It would be helpful if the authors can address in the text what new technical ideas are required to generalize the paper of Renaud et al. Additionally, it would be helpful if the authors can comment on if they believe the assumptions are sharp and to expand on the concluding remarks to further emphasize future directions stemming from this line of work.

**Limitations:**

yes

**Strengths And Weaknesses:**

The submission is technically sound, the various results are supported by proofs which are included in the appendix. The various assumptions are clearly discussed and appear reasonable for the setting considered.

The presentation of the paper is reasonable. The comparison with prior work is adequate and contextualizes the contributions of this submission well. The code used to generate the examples is also available in the supplementary materials. I would recommend that the authors directly state which regularizers fit within the proposed framework (i.e. satisfy all of the requisite assumptions) directly within the main text. At present that discussion is a bit scattered.
There are only a few typos which should be corrected.

1. p. 1 line 27, column 2: "very ill-posed in terms of differentiability" is not a very clear description. One could just say non-differentiable here.
2. p. 3 line 135, column 1: "leveraging on" -> "leveraging".
3. p. 4 line 212, column 2: "This kind difference-of-Moreau-envelope" -> "This kind of difference-of-Moreau-envelope".

In my opinion, the problem studied in this submission is of interest, given that priors associated with difference of convex functions have been proposed in the literature previously. Since the assumptions in this work are not particularly strong, I believe the results may be useful for practitioners. I believe, however, that a more clear discussion of future directions for this line of work are warranted.

The paper appears to leverage the results of [1] for the analysis. As noted in the text, using a difference of Moreau envelopes has also been considered before although not in the context of sampling. To my understanding, the main technical insight corresponds to showing that the difference of convex Langevin algorithm can be analyzed from the perspective of a general unadjusted Langevin algorithm with a drift.

1. Renaud, M., De Bortoli, V., Leclaire, A., & Papadakis, N. (2025). From stability of Langevin diffusion to convergence of proximal MCMC for non-log-concave sampling. NeurIPS 2025.

---

> ### Author Rebuttal · Authors · 2026-03-30
>
> We kindly thank the reviewer for the careful reading and the positive and constructive comments on our work. We reply the the questions/remarks as follows.
>
> >  I would recommend that the authors directly state which regularizers fit within the proposed framework...
>
> Thank you for the suggestion, we will make these points clearer in the revised manuscript.
>
> > There are only a few typos which should be corrected.
>
> Thank you, we will revise accordingly.
>
> > My main question for the authors regards the technical contributions of the work. It would be helpful if the authors can address in the text what new technical ideas are required to generalize the paper of Renaud et al.
>
> Our work being able to tackle nonsmooth DC priors is primarily thanks to the "correct" DC splitting and the appropriate DC Moreau smoothing technique, rather than by introducing an entirely new analytical framework compared to [Renaud2025]. In particular, we highlight that the stability result established in [Renaud2025] remains a fundamental component of the overall analysis.
>
> That said, our analysis introduces several novel contributions. First, we develop a new strategy to establish distant dissipativity for general drifts arising from the algorithm. A crucial step in proving convergence of the algorithm is to show that its algorithmic drift $b$ (recall that the algorithm can be reformulated as a ULA scheme with a general drift) satisfies a distant dissipativity condition: there exist constants $m', R' > 0$ such that $\langle b(x) - b(y), x - y \rangle \geq m' \|x - y\|^2 \quad \text{for all } \|x - y\| \geq R'$.
>
> Under either the Hölder smoothness or Lipschitz continuity of $r_2$ and the Lipschitz continuity of $r_1$, we were able to bound some key terms appearing in $\langle b(x) - b(y), x - y \rangle$ (see the proof of Lemma 4.3). In particular, we obtained
> \begin{align*}
> \langle b(x) - b(y), x - y \rangle \geq \frac{\mu}{2} \|x - y\|^2 - \text{lower-order terms in } \|x - y\|.
> \end{align*}
> Here $\mu>0$ is the dissipative constant of the target distribution.
>
> For sufficiently large $\|x - y\|$, the quadratic term dominates, yielding distant dissipativity.
>
> In contrast, [Renaud2025] follows a different route, leading to a coefficient in front of the quadratic term that is not guaranteed to be positive (see their Appendix G.1, end of page 52). As a result, they require a strong additional Assumption 3 to enforce positivity.
>
> Second, our use of two separate Moreau envelopes introduces additional technical challenges. In particular, we need to eventually control the discrepancy between the smoothed distribution and the target distribution, which we address in Appendix I.7. This control is possible thanks to the sub-Gaussian tail of target distribution (as a consequence of distant dissipativity -- Appendix A) and the lower-than-linear growth of the gradient of $r_2$.
>
> > if the authors can comment on if they believe the assumptions are sharp...
>
> We are not entirely sure if they are sharp but they seem very hard to further relax. When attempting to relax these assumptions individually, we failed to proceed further -- not just inconvenient, but fundamentally difficult. So they are essential to our analysis.
>
> First, we believe that the distant dissipativity condition on $V$ is fundamental if one aims to establish convergence to the target distribution $\pi$. Relaxing this condition would likely require re-establishing the stability result of [Renaud2025], which is a nontrivial task.
>
> Second, the Hölder smoothness or Lipschitz continuity assumptions on $r_1$ and $r_2$ arise primarily from the proofs, where they are used to control several critical terms. For example, if we relax the Lipschitz continuity of $r_1$ the control on terms I and II in the proof of Lemma 4.3 (page 23) is not possible. Also, the Lipschitz continuity or Hölder smoothness of $r_2$ is essential for controlling term III and for the final stage of bounding $W_q(\pi_{\lambda},\pi)$. Still, we believe these conditions can be relaxed to growth control of the subgradients (e.g., Hölder smoothness implies growth control on the subgradients and the vice versa does not necessarily hold), but this would require substantially more refined analysis and new techniques.
>
> > to expand on the concluding remarks to further emphasize future directions stemming from this line of work.
>
> We will expand the concluding remarks based on the above answers.
>
> **References**
>
> [Renaud2025] Renaud, M., De Bortoli, V., Leclaire, A., & Papadakis, N. (2025). From stability of Langevin diffusion to convergence of proximal MCMC for non-log-concave sampling. NeurIPS 2025.

---

> > ### Author Rebuttal · Reviewer_iCKg · 2026-03-31
> >
> > I thank the authors for their answers to the questions I raised in the review.
> > All of the questions I initially raised have been adequately addressed and I will maintain more current score.

---

> > > ### Author Response · Authors · 2026-04-07
> > >
> > > Thank you once again for your encouraging feedback and for taking the time to review our manuscript. We are pleased that our responses have adequately addressed your concerns.

---

### Decision · Program_Chairs · 2026-04-30

**Decision:**

Accept (regular)

**Comment:**

This paper studies the task of sampling from a distribution of the form $\exp(-f-r)/Z$ where the data fidelity term $f$ is Lipschitz and where the regularizer $r = r_1 - r_2$ is expressed as the difference of convex (DC) function.  The sampling method is a modification of the standard unadjusted Langevin algorithm that combines a proximal operator on the convex part of $r$, and a gradient step on the concave part of $r$.  The authors study convergence of this scheme under certain regularity assumptions on the functions $f+r$, $r_1$, $r_2$ etc.  The authors conduct numerical experiments to compare their method with methods that do not take into account the DC structure, and on a real-life tomography example.

The general consensus among the reviewers is that the contribution is sufficiently interesting and of importance, and the results are sound and relevant to the proposed methodology.  The paper is backed by a reasonable set of numerical experiments.  I am happy to recommend acceptance.

The reviewers do raise a number of useful suggestions that will strengthen the paper, including (i) deriving interpretable bounds, and (ii) a more in-depth discussion about extra gradient evaluations compared to ULA (this was briefly treated in the rebuttal as "we do not expect" but now is a good opportunity to address such concerns precisely).  I would urge the authors to take the suggestions raised by the reviewer seriously.